# A quantitative map of nuclear pore assembly reveals two distinct mechanisms

Shotaro Otsuka[1,3 ✉], Jeremy O. B. Tempkin[2], Wanlu Zhang[1], Antonio Z. Politi[1,4], Arina Rybina[1], M. Julius Hossain[1,5], Moritz Kueblbeck[1], Andrea Callegari[1], Birgit Koch[1,6], Natalia Rosalia Morero[1], Andrej Sali[2] & Jan Ellenberg[1 ✉]

Understanding how the nuclear pore complex (NPC) is assembled is of fundamental importance to grasp the mechanisms behind its essential function and understand its role during the evolution of eukaryotes[1–4]. There are at least two NPC assembly pathways—one during the exit from mitosis and one during nuclear growth in interphase—but we currently lack a quantitative map of these events. Here we use fluorescence correlation spectroscopy calibrated live imaging of endogenously fluorescently tagged nucleoporins to map the changes in the composition and stoichiometry of seven major modules of the human NPC during its assembly in single dividing cells. This systematic quantitative map reveals that the two assembly pathways have distinct molecular mechanisms, in which the order of addition of two large structural components, the central ring complex and nuclear filaments are inverted. The dynamic stoichiometry data was integrated to create a spatiotemporal model of the NPC assembly pathway and predict the structures of postmitotic NPC assembly intermediates.

The nuclear pore complex (NPC) is the largest non-polymeric protein complex in eukaryotic cells. It spans the double membrane of the nucleus—the nuclear envelope—and mediates macromolecular transport between the nucleus and the cytoplasm. To achieve this essential function, the NPC forms an octameric proteinaceous channel composed of multiples of 8 of more than 30 different nucleoporins (Nups) that form 6–8 protein modules, the NPC subcomplexes[1,2]. Therefore, more than 500 individual proteins have to come together to assemble one nuclear pore, which has the mass of tens of ribosomes. NPCs are thought to represent a key step in the evolution of endomembrane compartmentalization that allowed ancestral eukaryotes to separate their genome from the cytoplasm[3,4].

In proliferating cells that undergo open mitosis, there are two main pathways by which NPCs assemble. During nuclear assembly after mitosis, NPCs form together with nuclear membranes to rapidly build new nuclei in the daughter cells—this process is called postmitotic NPC assembly. During nuclear growth in interphase, NPCs continue to assemble continuously for homeostasis—this is referred to as interphase assembly. Research over the past decade has revealed that postmitotic and interphase NPC assembly possess distinct kinetic, molecular and structural features[5–12], suggesting that two fundamentally different mechanisms build the same protein complex. In postmitotic assembly, several thousand NPCs assemble within a few minutes during sealing of the initially fenestrated nuclear membranes, whereas interphase NPC assembly occurs more sporadically, requires about one hour, and involves a new discontinuity in the double membrane barrier of the nuclear envelope. Studies using molecular depletions

have shown that the Nup ELYS is required for postmitotic assembly but appears dispensable for interphase assembly[5], whereas the membrane curvature-sensing domain of Nup133[6], Pom121 and Sun1[7,8], the import of Nup153 into the nucleus[9], and Torsins[10] seem to be required only for interphase assembly. Recent studies correlating real-time imaging with 3D electron microscopy have revealed that postmitotic NPC assembly proceeds by radial dilation of small membrane openings[11], whereas in interphase, assembly induces an asymmetric inside-out fusion of the inner and outer nuclear membranes[12].

However, how several hundred proteins self-organize to form the NPC channel via these two distinct assembly pathways has remained largely unknown. It is technically challenging to locate the transient and rare assembly events, which has prevented investigation of the structure of assembly intermediates by either cryo-electron microscopy (cryo-EM) tomography or super-resolution microscopy. In addition, the large number of building blocks and their cooperativity often leads to complex nonlinear kinetics that can only be interpreted mechanistically using computational modelling of the structures formed during assembly. Although we have some information about the dynamic addition of Nups after mitosis[13,14], only sparse dynamic data is available for interphase assembly[15,16]. Notably, these earlier studies could not distinguish between postmitotic and interphase assemblies, whose co-occurrence in different regions of the nucleus was only discovered later; moreover, these studies provided only qualitative descriptions, as they relied on ectopic expression of fluorescently tagged Nups. Kinetic data about NPC assembly that can distinguish between the postmitotic and interphase pathways is required, including the copy

[1]Cell Biology and Biophysics Unit, European Molecular Biology Laboratory, Heidelberg, Germany. [2]Department of Bioengineering and Therapeutic Sciences, Department of Pharmaceutical Chemistry, Quantitative Biosciences Institute, University of California, San Francisco, San Francisco, CA, USA. [3]Present address: Max Perutz Labs, University of Vienna and the Medical University of Vienna, Vienna Biocenter (VBC), Vienna, Austria. [4]Present address: Max Planck Institute for Multidisciplinary Sciences, Göttingen, Germany. [5]Present address: Centre for Cancer Immunology, Faculty of Medicine, University of Southampton, Southampton, UK. [6]Present address: Max Planck Institute for Medical Research, Heidelberg, Germany. ✉e-mail: shotaro.otsuka@univie.ac.at; jan.ellenberg@embl.de

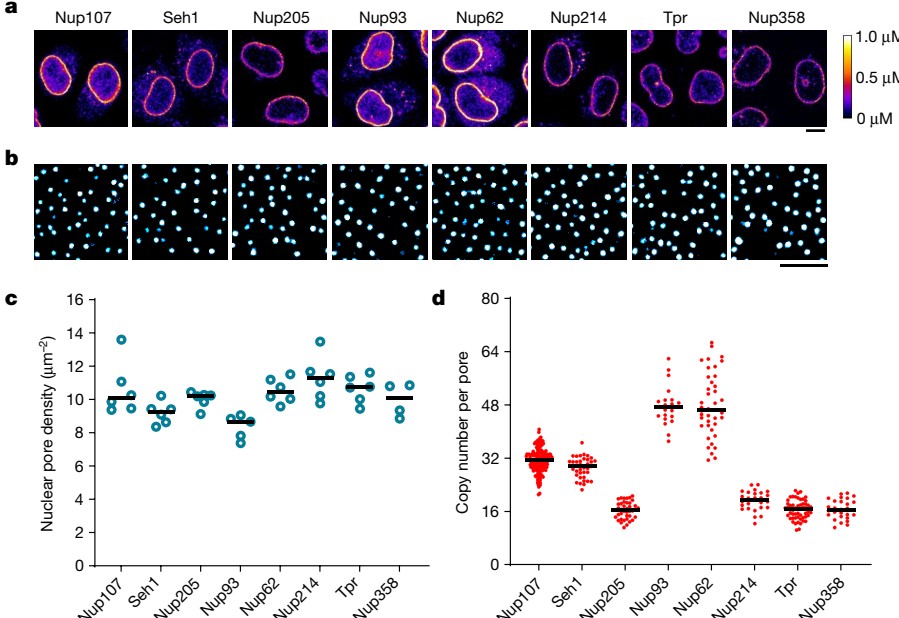

**Fig. 1 | Quantitative imaging of GFP-knock-in Nup cell lines. a**, Confocal microscopy of genome-edited HeLa cells with homozygous expression of mEGFP-tagged Nups. Fluorescence intensity was converted to protein concentration (colour bar) using FCS-calibrated imaging[22]. Images were filtered with a median filter (kernel size: 0.25 × 0.25 μm) for presentation purposes. Scale bar, 10 μm. **b,c**, Stimulated emission depletion (STED) microscopy of genome-edited cells stained with Nup62 antibody. Cells were imaged (**b**) and the density of nuclear pores was quantified (**c**). $n$ = 6 (Nup107), 6 (Seh1), 6 (Nup205), 5 (Nup93), 6 (Nup62), 6 (Nup214), 6 (Tpr) and 4 (Nup358) cells. Scale bar, 1 μm. **d**, Calculated copy number of Nups per nuclear pore. $n$ = 241 (Nup107), 37 (Seh1), 41 (Nup205), 20 (Nup93), 41 (Nup62), 26 (Nup214), 55 (Tpr) and 28 (Nup358) cells. The horizontal line represents the median.

numbers of Nups that assemble into forming NPCs over time. Such data would enable modelling of the assembly process and enable us to understand the two assembly mechanisms.

## Quantitative imaging of ten Nups

To quantitatively analyse the changes in concentration of Nups at the nuclear envelope during exit from mitosis and nuclear growth in G1, we genome-edited HeLa cells, homozygously tagging the endogenous genes encoding different Nups with mEGFP or mCherry. Building on previous work[11,12,17,18], we created a set of ten endogenously tagged Nups that systematically represent the major building blocks of the fully assembled pore, including the nuclear filament protein Tpr, the nuclear Nup153, the Y-complex members Nup107 and Seh1, the central ring complex members Nup93 and Nup205, the central channel protein Nup62, the transmembrane protein Pom121, as well as the cytoplasmic filament proteins Nup214 and Nup358. Homozygous tagging was verified by careful quality control of the genome-edited monoclonal cell lines[19], ensuring that the tagged subunit was localized to the NPC and that cell viability and mitotic progression were normal (Figs. 1 and 2). Tagged Nup153, Pom121 and Nup358 were expressed at subphysiological levels (Extended Data Figs. 1 and 2; details in Discussion). The fusion proteins are likely to be functional, given that most Nups show strong phenotypes upon knockout or depletion[20].

To characterize the fluorescently tagged Nups, we first performed super-resolution (STED) microscopy to determine the NPC density in fully grown nuclei of the knock-in cell lines, showing that homozygous tagging had little effect on NPC density, which was within a 15% range between all cell lines, with an average of 10.1 NPCs per μm² (Fig. 1b,c), in good agreement with our previous estimates from electron microscopy studies on HeLa cells[12]. We then used fluorescence correlation spectroscopy (FCS) calibrated confocal microscopy[21,22] to determine the concentration and total number of Nups at the nuclear envelope in living cells (Supplementary Table 1; details in Methods). Using the measured NPC density and the Nup concentration at

the nuclear envelope, we were able to calculate the average copy number of each Nup per NPC (Fig. 1d). These data showed that the investigated Nups were present in 16, 32, or 48 copies per pore on average, as expected from the eight-fold symmetry of the complex and are in good overall agreement with previous estimates from mass spectrometry[23].

## Dynamic change of Nup numbers

We then used our cell line resource to quantitatively image the Nups during both postmitotic and interphase NPC assembly, from metaphase until the end of the rapid nuclear growth phase in G1, 2 h after anaphase onset. To this end, we performed systematic FCS-calibrated 3D confocal time-lapse microscopy[17] (Fig. 2). Using the single-molecule fluctuation calibration, the 4D imaging data could be converted into maps of subcellular protein concentration (Fig. 2). Counterstaining live nuclei with silicon–rhodamine (SiR)–DNA[24] enabled computational image segmentation to measure the soluble cytoplasmic pool and the nuclear envelope-associated pool over time[17]. Temporal alignment to anaphase onset then enabled us to compare the dynamic association of all Nups with the nuclear envelope over time (Fig. 2). Overall, the investigated Nups are present in 250,000 to 1,200,000 copies per human metaphase cell. After mitosis, this building material is split between the daughter cells with little detectable new protein synthesis in the first hour after anaphase onset (Extended Data Fig. 3). Notably, 34–53% of the soluble pool present in the cytoplasm in metaphase was rapidly re-localized to the nuclear envelope during the first hour after exit from mitosis (43, 38, 47, 37, 41, 44, 53 and 34% for Nup107, Seh1, Nup205, Nup93, Nup62, Nup214, Tpr and Nup358, respectively) (Extended Data Fig. 3), indicating that NPC assembly initially relies almost entirely on the pool of building blocks inherited from the mother cell to form the first 4,000 to 5,000 NPCs[11,12]. We confirmed that the observed GFP-tagged Nups are indeed recruited specifically to NPCs and not to the nuclear envelope surface in general by imaging of single NPCs using STED microscopy (Extended Data Fig. 4a,b).

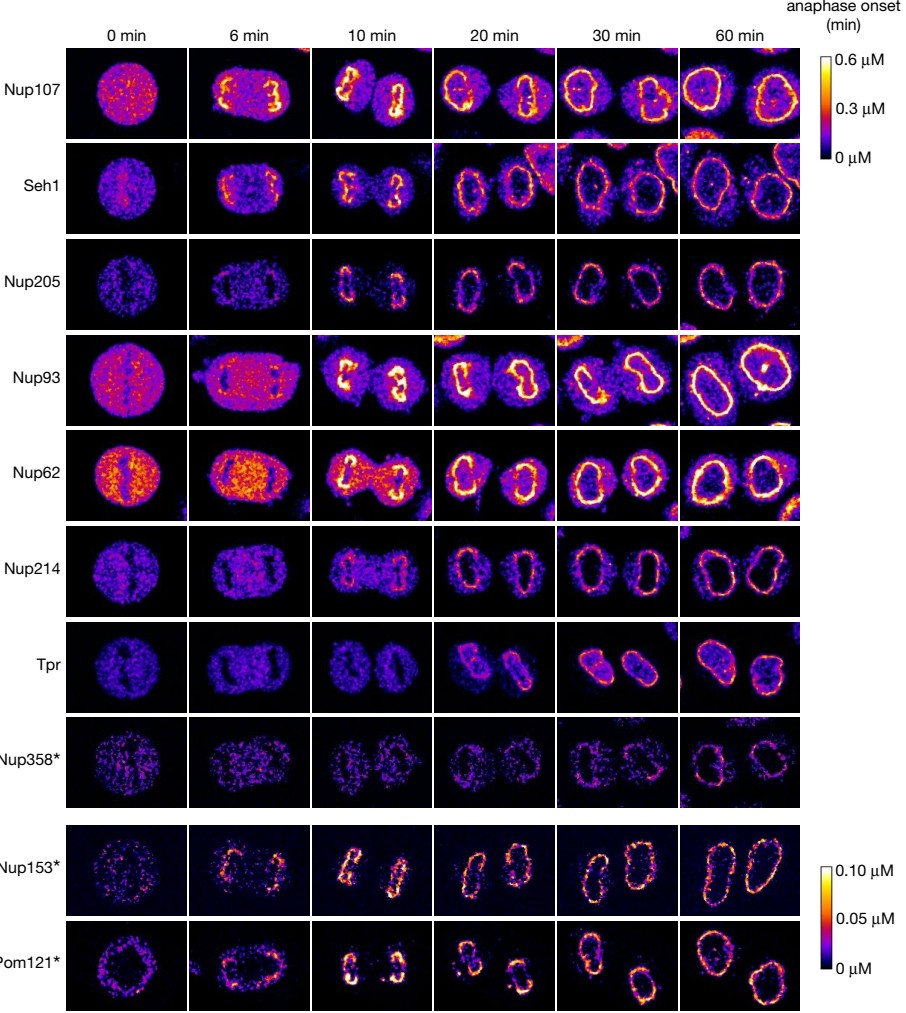

**Fig. 2 | Dynamic concentration maps of Nups after anaphase onset.** HeLa cells whose Nups are endogenously tagged with mEGFP or mCherry were imaged every 30 s by 3D confocal microscopy. Single confocal sections are shown. Images were calibrated by FCS to convert fluorescence intensities into cellular protein concentration (colour bars). Nup153 is heterozygously tagged and Nup358 and Pom121 are subphysiologically expressed (Extended Data Figs. 1 and 2). The bottom colour bar is for Nup153 and Pom121; the top colour bar is for the rest of the Nups. Images were filtered with a median filter (kernel size: 0.25 × 0.25 μm). Scale bar, 10 μm.

## Quantification of assembly kinetics

Postmitotic and interphase NPC assembly can be observed in the same living cell in different regions of the nuclear envelope within the first 2 h after mitosis[12]. Whereas postmitotic assembly dominates the peripheral 'non-core' regions of the nuclear envelope, the central 'inner-core' area is populated with NPCs only after exit from mitosis[25], when dense spindle microtubules have been removed from the DNA surface[26]. Using computational segmentation and assignment of the inner-core and non-core regions (Extended Data Fig. 5a; details in Methods, modified from ref. [12]), we measured the changes in concentration of the ten Nups in these two regions separately. A two-component model of a fast (postmitotic) and a slow (interphase) assembly process fits the experimental data well, enabling us to kinetically unmix the two assembly processes for each Nup (Extended Data Figs. 5b–d and 6). In this way, we were able to perform an integrated analysis of the real-time kinetics of absolute concentration changes of Nups in all major NPC modules during the two assembly processes at the nuclear envelope (Fig. 3a). This analysis immediately revealed that the overall duration of the two processes is very different, with postmitotic assembly essentially complete 15 min after anaphase onset, whereas interphase assembly only reaches a

plateau after 100 min, consistent with our previous estimates based on live-cell imaging[13,16] and correlative electron microscopy[11,12]. Both processes reached the same final ratios between the different Nups and thus presumably formed identical NPCs. However, the temporal orders in which components were added were distinct, including— for example—an earlier assembly of Pom121 relative to the Y-shaped complex during interphase assembly, consistent with our previous observations[16].

To comprehensively investigate the molecular differences between the two assembly processes, we relied on the constant NPC density and changes in nuclear surface area[12] to convert the nuclear envelope concentrations of all investigated Nups into their average copy number per assembling NPC over time (Fig. 3a). This result in turn enabled us to estimate changes in subunit stoichiometry of the complex during its assembly in living cells. To facilitate the comparative analysis of the assembly kinetics between the two pathways, in which ten components assemble with different speeds and orders, we first reduced the dimensionality of the kinetic data. To this end, we assigned a single characteristic time point of assembly to each Nup by sigmoidal fitting of its full kinetic signature (Extended Data Fig. 7a). Plotting the copy number against the median assembly time

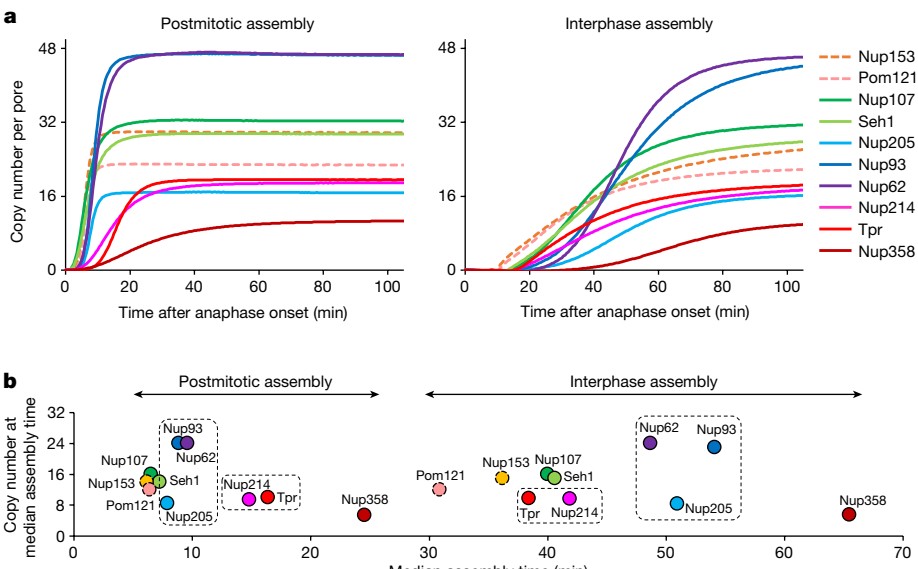

**Fig. 3 | The molecular assembly order and maturation kinetics are distinct for postmitotic and interphase assembly. a**, The average copy number per nuclear pore computed from mathematical modelling for postmitotic (left) and interphase (right) assembly (details in Methods and Extended Data Figs. 5 and 6). For Nup153 and Pom121 (dashed lines), the absolute amount was estimated using the copy number determined from a previous study[23] (32 for Nup153 and 16 for Pom121). **b**, The average copy number of individual Nups per nuclear pore are plotted along their median assembly time in postmitotic and interphase assembly pathways. Boxes highlight the Nups that show marked differences in their order of assembly between the two pathways.

provides an overview of the major molecular differences between the two assembly pathways (Fig. 3b).

Whereas the Y-complex, Pom121 and Nup153 form a core of the first modules that assemble almost simultaneously within one minute in postmitotic assembly, these components are stretched out into a clear temporal order of first Nup153, second Pom121 and third the Y-complex (notably with its two investigated subunits also assembling simultaneously in interphase) over more than ten minutes in interphase assembly. The end of assembly is marked for both processes by the addition of the large cytoplasmic filament protein Nup358. The major difference is thus not in initiation or termination of assembly, but rather in the middle of the two assembly pathways. During postmitotic assembly, the Y-complex is rapidly combined with components of the central ring, building the inner core of the pore within just three minutes before addition of either cytoplasmic or nuclear filament proteins, which follow later. By contrast, during interphase assembly, the Y-complex is first combined with the nuclear filament protein Tpr and the base of the cytoplasmic filament Nup214, and the central ring complex is added later. This observation clearly shows that postmitotic and interphase NPC assembly proceed with different speeds and follow different molecular mechanisms, with inverted molecular orders between the central ring and nuclear filaments (Fig. 3b and Extended Data Fig. 7b,c).

To validate whether Tpr indeed assembles earlier in the interphase assembly pathway, we used immunostaining of assembling NPCs with antibodies against Tpr and a later-assembling Nup, Nup62, and visualized them by 3D STED super-resolution microscopy. This single-NPC imaging demonstrated that a large number of assembling NPCs in the inner-core region in early G1 contain Tpr but not Nup62, whereas most of the assembled NPCs in the non-core region of the same cells contain both Tpr and Nup62 (Fig. 4a,c), confirming the surprisingly early recruitment of Tpr in the interphase assembly pathway. In addition to cells in early G1, we also examined cells later in interphase during S and G2 phase. Consistently, we found that a substantial number of NPCs also contain Tpr but not Nup62 (Fig. 4b,c), indicating that interphase assembly at later cell cycle stages follows the same order.

## Integrative modelling of postmitotic assembly

To obtain a more comprehensive mechanistic view, we computed a spatiotemporal model of the macromolecular assembly pathway on the basis of our dynamic multimolecular stoichiometric data in combination with the available ultrastructural data about NPC assembly[11] and the partial pseudoatomic model of the mature human NPC[27,28]. We modelled only the postmitotic assembly, as it showed the kinetic hallmarks of a sequential process and is known to proceed by dilating an existing membrane pore with a smoothly growing proteinaceous density[11]. We focused on the Nups contained in the structural model of the human NPC, including Nup107 and Seh1 for the Y-complex as well as Nup93, Nup205 and Nup62 for the central ring or channel complex[27,28]. We constrained their copy numbers with our stoichiometry data for the time points for which correlative electron tomography data are available[11], enabling us to use the membrane shapes and associated protein densities from electron tomography to model the Nup configurations at these time points. To compute a spatiotemporal model of the assembly pathway, we generalized our integrative modelling method for determining static structures of macromolecular assemblies[29,30]. In outline (see Methods for details), we first computed ensembles of Nup configurations at discrete time points, connected pairs of these configurations at adjacent time points into assembly trajectories, and finally ranked the alternative trajectories by fit to our data.

We focused on the best fitting macromolecular assembly pathway that accounted for over 85% of the posterior model likelihood (the second-scoring model accounted for only 14.5%; Fig. 5 and Extended Data Fig. 8). This trajectory starts with the formation of a single nuclear ring, composed of eight Y-complexes, concomitantly with an initial accumulation of the FG-repeat protein Nup62 and the inner-ring complexes (the Nup93–Nup188–Nup155 and Nup205–Nup93–Nup155 complexes and Nup155) in the centre of the membrane hole (Fig. 5, 5 and 6 min). We experimentally validated that Nup155 and Nup188 assemble as the model predicted (Extended Data Figs. 9 and 10 and Supplementary Fig. 1). The cytoplasmic Y-complex is then added on the cytoplasmic side, before the second set of the Y-complex ring assembles on the nuclear side, again one eight-membered ring after another (Fig. 5, 8 and

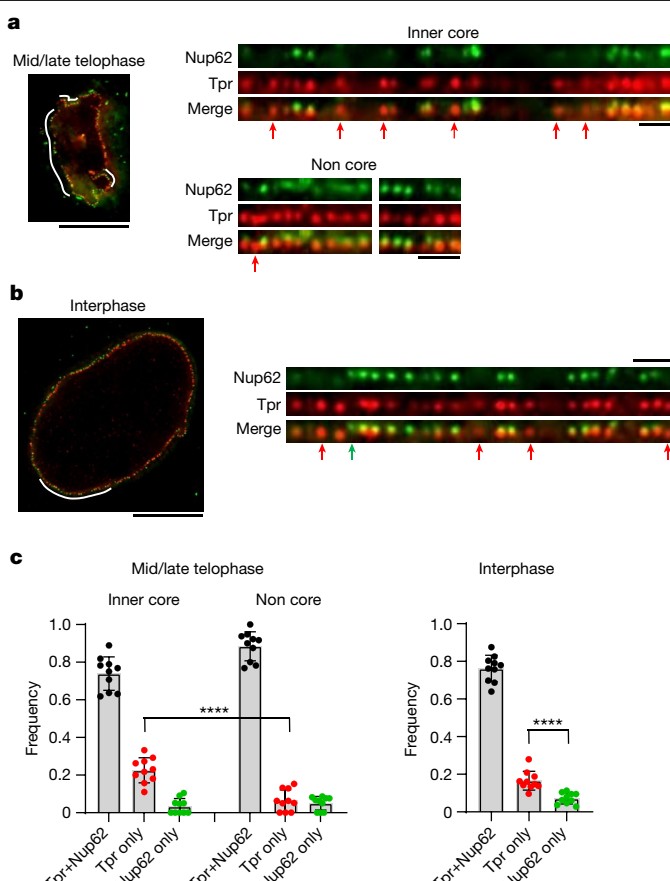

**Fig. 4 | Observation of single nuclear pores by super-resolution microscopy confirms the recruitment of Tpr precedes Nup62 in the interphase assembly pathway. a**,**b**, Three-dimensional STED imaging of Nup62–mEGFP genome-edited cells stained with a GFP nanobody and an anti-Tpr antibody at mid/late telophase (**a**) and interphase (**b**). Scale bars: left, 10 μm; right, 1 μm. The regions of the nuclear envelope indicated by white lines (left) are flattened and displayed (right). Nuclear pores that contain only Tpr or Nup62 are indicated by red and green arrows, respectively. **c**, The frequency of nuclear pores that contain both Tpr and Nup62, only Tpr or only Nup62 at each cell cycle stage. Data are from 10 cells for each stage. Data are mean ± s.d. ****$P = 0.000017$ (mid/late telophase) and ****$P = 0.000080$ (interphase), unpaired two-tailed $t$-test.

10 min). In the centre of the pore, Nup62 dilates from an amorphous mass into a small ring and associates with inner-ring Nups to form the central ring complex. This 'nuclear-ring-first' assembly mechanism is consistent with the observation of an eight-fold symmetric protein density on the inner nuclear membrane at early stages of assembly[11].

## Discussion

Our data revealed that the two NPC assembly pathways are markedly different. Although it had previously been shown that postmitotic and interphase assembly pathways have different molecular requirements[5–12], very little was known about the interphase pathway owing to its rare and sporadic nature. Although relative kinetics were available for some Nups[15,16], how key Nups composing cytoplasmic filaments, central channel, and nuclear basket, or multiple subunits of the Y-shaped and inner-ring complexes assemble in interphase had not been studied. Here we provide measurements on subunit composition and stoichiometry for both postmitotic and interphase assembly for ten Nups that represent all major building blocks of the NPC. Our data thus systematically reveal the main molecular differences between the two assembly pathways. During postmitotic assembly, the Y-complex is rapidly combined with components of the central ring, building the inner—and at this time already transport competent[13]—core of the pore before the addition of either cytoplasmic or nuclear filament proteins. By contrast, during interphase assembly, the Y-complex is first combined with the nuclear filament protein Tpr and the base of the cytoplasmic filament Nup214, whereas the central ring complex is added much later.

The molecular pathway that we observed for interphase assembly makes new predictions about its unique, inside-out evaginating, mechanism. The cell first builds the two nuclear Y-rings and accumulates the material for their cytoplasmic counterparts and then combines them with the nuclear filament proteins, all preceding membrane fusion[12]. This mechanism suggests that the cytoplasmic Y-rings, including the base of the cytoplasmic filament Nup214, are already 'prebuilt' within the inner membrane evagination, where the small available volume would predict that they must be present in a very different structure than in the fully mature pore after fusion. Surprisingly, the central ring complex—which is a core structural element between the nuclear and cytoplasmic Y-rings in the mature pore, is added later—suggesting that during interphase assembly, nuclear transport control is added after the membrane fusion step. The unexpectedly early presence of Tpr suggests—although it does not prove—a potential role of Tpr in the initiation of interphase assembly. This long coiled-coil multimeric protein component could be involved in bending the inner nuclear membrane to create the invagination, potentially providing force by unfolding or restructuring a coiled-coil bundle[1,2]. Conversely, considering previous reports that Tpr together with the kinase ERK is required to control the spacing between NPCs[31], Tpr-based signalling might also be important in assembly site selection to prevent simultaneous assembly of multiple NPCs in close proximity with each other, which would probably result in abnormal distortions of the inner nuclear membrane.

Our data on interphase assembly in human cells furthermore revealed striking differences with budding yeast, which undergoes a closed mitosis and lacks the postmitotic pathway[32]: (1) the Tpr homologues Mlp1/2 assemble late in yeast; (2) human cells exhibit a synchronous assembly of the Y-complex components, whereas in yeast, the stem (for example, Nup107) and the head (for example, Seh1) of the Y-complex do not assemble simultaneously; and (3) the central ring components assemble after the Y-complex components in human cells, whereas they assemble together in yeast[32]. These differences could be owing to the fundamental differences in structure (such as the presence of a nuclear lamina) and/or cell cycle remodelling of the nuclear envelope (closed versus open mitosis) between human and yeast cells.

Our dynamic stoichiometry data, combined with previous electron tomography data on membrane topology and protein volume, enabled us to predict the structures of the NPC assembly intermediates using integrative spatiotemporal modelling. We focused on the topologically simpler postmitotic assembly. The resulting model enables multiple new mechanistic predictions regarding postmitotic assembly that will help guide future work. For example, the model suggests that the central ring complex might be needed to prevent sealing of holes in the endoplasmic reticulum after mitosis, and that the hydrophobic FG-repeats of central channel Nups might have a role in the dilation of the small membrane hole into the larger, NPC-sized channel. Our pathway model was constrained by the fully assembled NPC structure[27,28] containing 16 copies of Nup205 and 32 copies of Nup62; thus, it does not address how additional copies of Nup205 and Nup62 that have been reported in recent structural studies and a proximity-dependent biotin identification study[27,33–35] may be assembled.

Our combined experimental and computational approach for determining macromolecular assembly pathways is likely to benefit from additional experimental data provided by emerging methods, such as 3D super-resolution microscopy for imaging the molecular architecture of the NPC intermediates[36,37] and dynamic super-resolution

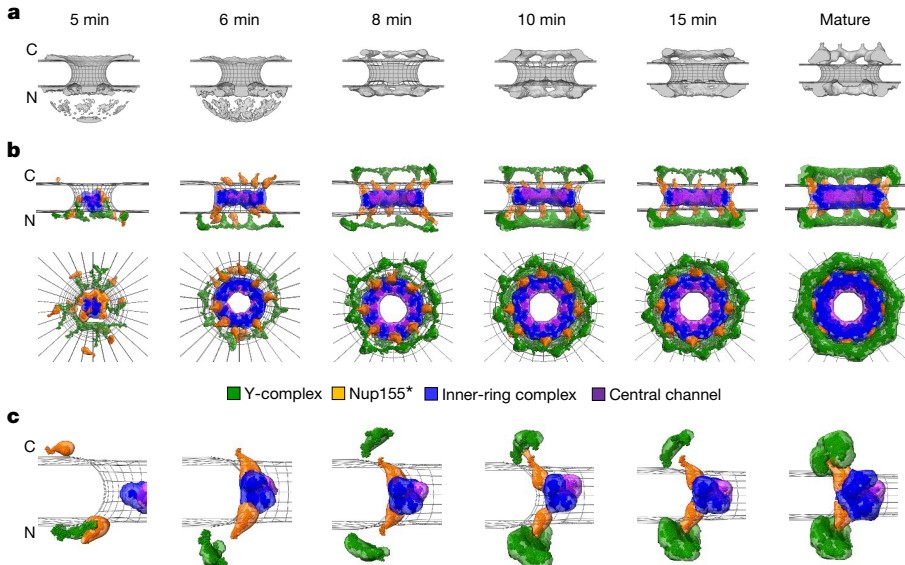

**Fig. 5 | Integrative model of the postmitotic NPC assembly pathway.**
**a**, Protein density (grey) overlaid with the nuclear envelope surface (wireframe model, grey) at each time point[12] used for integrative modelling. C, cytoplasm; N, nucleoplasm. The dome-like density in the nucleoplasmic side at 5 and 6 min is the noise from electron tomography. **b**, The best-scoring model of the postmitotic assembly pathway (top and side views). The uncertainty of each Nup localization is indicated by the density of the corresponding colour. The Y-complex is shown in green; an isolated fraction of Nup155 that is not forming a complex with Nup93, Nup188 and Nup205 is shown in orange; Nup93–Nup188–Nup155 and Nup205–Nup93–Nup155 complexes are in blue;

and the Nup62–Nup58–Nup54 complex is in purple. **c**, The enlargement of one of the spokes. The model has a much higher score than all the other lower-scoring pathways (Extended Data Fig. 8). The pseudoatomic model of the native NPC structure[27,28] we imposed as the endpoint of the assembly pathway does not include the Y-complex-bound fraction of Nup205 or the Nup214-bound fraction of Nup62, and thus contains only 16 of the 40 copies of Nup205[33,34] and 32 of the 48 copies of Nup62[35] in the fully mature NPC. How the remaining 24 copies of Nup205 and 16 copies of Nup62 are assembled remains elusive.

methods for mapping structural changes in real time[38]. We expect that this additional information will enable higher resolution modelling of even more complex assembly mechanisms, such as the interphase NPC assembly that involves much more substantial changes in membrane topology and protein conformation.

The need for better methods to determine dynamic and variable copy numbers of Nups reliably at the level of single protein complexes under physiological conditions is further highlighted by the fact that our live-cell-based approach resulted in copy number estimates that differ from estimates derived by other approaches for four out of the ten investigated Nups. The main methods used in the field, including quantitative mass spectrometry from cell populations[23], fitting of single protein structures into highly averaged cryo-EM densities[33,34], as well as calibrated imaging of fluorescently tagged knock-in proteins in single living cells as used in our study, currently all have limitations in this regard. For example, cryo-EM based averaging often selects 'complete' NPCs, to obtain a 'fully occupied' average NPC model, whereas live-cell imaging data average NPC stoichiometry from all pores in one cell and thus include pore-to-pore variability, which can result in lower estimates. By contrast, whereas homozygous genome editing of a fluorescent tag ensures close to complete labelling and biological function of the fusion protein, it can still affect the expression level, which could lead to a different occupancy within the NPC.

Four out of the ten investigated Nups are relevant to discuss in the light of these methodological limitations—Nup153, Pom121, Nup205, and Nup358. For the first two, our estimates were much lower than expected from the structural model of the fully assembled state that we imposed as the end-state of our pathway model (Fig. 2 and Extended Data Figs. 1 and 2). We therefore regarded them as effectively expressed at subphysiological levels and normalized their stoichiometry to the expected number of copies in the mature pore for comparison with other Nups (Fig. 3a). For Nup205 and Nup358, our copy number estimates were 40–50% lower than expected, even though for Nup205 our

cell lines express physiological levels of the fusion protein, whereas for Nup358 this correlates to a reduced level of expression (Extended Data Fig. 2). More work is needed to reliably measure the dynamic and variable composition of the NPC, especially during its assembly, and incorporate these data into modelling.

Beyond the assembly mechanism, the two pathways that we have mapped here begin to shed light on the role of the NPC during the evolution of the endomembrane system in eukaryotes[3,4]. We speculate that the modern NPC combines ancient membrane bending (for example, coiled-coil filaments, such as Tpr) and membrane hole plugging or dilation (for example, central ring complex and FG-repeat proteins) modules that were potentially previously used separately for extruding the cell surface or keeping transport channels open in the endomembranes around the genome. The key evolutionary innovation might lie in combining and controlling these activities, potentially by the eight-membered ring architecture of the nuclear Y-complex. In future, comparing NPC assembly pathways in different species on the eukaryotic evolutionary tree might help us to understand how the assembly of complex modern protein machines reflects their evolutionary origins.

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

## Methods

### Cell culture

Wild-type HeLa Kyoto cells (RRID:CVCL_1922) were a gift from S. Narumiya; their genome was sequenced previously[39]. Cells were grown in high glucose Dulbecco's Modified Eagle's Medium (DMEM) containing 4.5 g l$^{-1}$ D-glucose (Sigma Aldrich) supplemented with 10% fetal calf serum (FCS), 2 mM L-glutamine, 1 mM sodium pyruvate and 100 μg ml$^{-1}$ penicillin and streptomycin at 37 °C and 5% $CO_2$. Cells tested negative for mycoplasma contamination by PCR every 2 or 3 months.

### Genome editing

Monomeric enhanced GFP (mEGFP) and mCherry were inserted into the genome using zinc finger nucleases, CRISPR–Cas9 nickases, or Alt-R S.p. HiFi Cas9 Nuclease V3. The following six cell lines had been generated and published previously: Nup62–mEGFP[18], mEGFP–Nup107[12], mEGFP–Nup205[11], mEGFP–Nup214, mEGFP–Nup358 (also called RanBP2) and Tpr–mEGFP[17]. The following five cell lines were generated in this study: mEGFP–Seh1, Nup93–mEGFP, mEGFP–Nup153, Nup188–mEGFP and Pom121–mCherry. The guide RNA (gRNA) sequences used to generate these cell lines are summarized in Supplementary Table 2. For the Nup93–mEGFP cell line, CRISPR–Cas9 nickases and the donor plasmid were transfected by electroporation (Neon Transfection System, Thermo Fisher Scientific) instead of a polymer-mediated transfection reagent.

### Western blot

Cells were lysed on ice in RIPA buffer (50 mM Tris-HCl, 150 mM NaCl, 1.0% Triton X-100, 1% sodium deoxycholate, 0.1% SDS, 2 mM EDTA, pH 7.5), supplemented with complete EDTA-free protease inhibitor cocktail (Sigma Aldrich), PhosSTOP (Sigma Aldrich), and 0.1 mM phenylmethylsulfonyl fluoride (Sigma Aldrich). The cell lysates were snap-frozen in liquid nitrogen and quickly thawed at 37 °C two times. The lysates were then centrifuged at 16,000g at 4 °C for 10 min and the supernatant was used for immunoblot analysis. Protein concentration was quantitated using the Pierce BCA Protein Assay Kit (Thermo Fisher Scientific). For Nup188 (Extended Data Fig. 10b), the cell lysates were run on NuPAGE 4–12% Bis-Tris Gels (Novex Life Technologies) and transferred onto PVDF membrane using the Bio-Rad transfer system. After blocking with 4% milk solution (nonfat milk powder in PBS + 0.05% Tween 20), the proteins were labelled with anti-Nup188 (A302–322A, Bethyl Laboratories, 1:5,000) and anti-γ-tubulin (T5192, Sigma Aldrich, 1:1,000) antibodies. Subsequently anti-rabbit IgG horseradish peroxidase (HRP)-conjugated secondary antibody (W4011, Promega) was used to detect the protein of interest with chemiluminescence reaction. For Nup205 and Nup358 (Extended Data Fig. 2), lysates were run on NuPAGE 3–8% Tris-acetate gels (Novex Life Technologies), and anti-Nup205 (ab157090, Abcam, 1:1,000), anti-Nup358 (HPA023960, The Human Protein Atlas, 1:1,000), and anti-vinculin (ab219649, Abcam, 1:1,500) antibodies were used to detect the proteins. Simple western assays were also performed in a Jess instrument (ProteinSimple) using 66–440 kDa capillary cartridges in accordance with the provider's instructions. The anti-Nup205, Nu358, and γ-tubulin antibodies were used at 1:50 dilution. For Nup205, mouse anti-γ-tubulin antibody (T5326, Sigma) was used instead of the rabbit anti-γ-tubulin antibody to adjust the signal intensity. The secondary antibodies used were the ones provided by the company at a ready-to-use dilution (goat anti-mouse HPR-conjugated secondary antibody, 040-655, Bio-Techne; goat anti-rabbit HPR-conjugated secondary antibody, 040-656, Bio-Techne).

### FCS-calibrated live-cell imaging and estimation of Nup copy numbers per NPC

Live-cell imaging was performed using wild-type cells, wild-type cells transfected with mEGFP using Fugene6 (Promega), mEGFP–Nup107 genome-edited cells, or cells of another mEGFP–Nup genome-edited cell line. Cells were seeded in 8-well Lab-Tek Chambered Coverglass (Thermo Fisher Scientific). On the day of live-cell imaging, DMEM was replaced with imaging medium: $CO_2$-independent medium without phenol red (Invitrogen) containing 20% FCS, 2 mM L-glutamine, and 100 μg ml$^{-1}$ penicillin and streptomycin. The imaging medium was supplemented with 50 nM SiR (Hoechst) to stain DNA[24]. Cells were incubated inside the microscope-body-enclosing incubator at 37 °C for at least 30 min before imaging. For Nup188–mEGFP genome-edited cells, the following medium was also used instead of the imaging medium: 30 mM HEPES pH 7.4 containing 9.3 g l$^{-1}$ minimum essential medium Eagle (Sigma Aldrich), 10% FCS, 1% MEM non-essential amino acids (Thermo Fisher Scientific, Gibco), and 100 μg ml$^{-1}$ penicillin and streptomycin.

Calibrated imaging using FCS was carried out as described[22], using Fluctuation Analyzer 4G 150223 (https://www-ellenberg.embl.de/resources/data-analysis), FCSFitM v0.8 (https://git.embl.de/grp-ellenberg/FCSAnalyze), FCSCalibration v0.4.2 (https://git.embl.de/grp-ellenberg/FCSAnalyze), RStudio 1.1.383, R 3.4.1, and Python v3.6.8. In brief, the confocal volume was determined by performing FCS using a dye with known diffusion coefficient and concentration (Alexa Fluor 488 NHS ester; Thermo Fisher Scientific for mEGFP). To convert fluorescence intensity to the concentration, FCS was performed in cells that transiently express mEGFP alone, and a calibration curve was obtained by plotting the fluorescence intensity against concentration. The background fluorescence signal was measured in cells not expressing fluorescent proteins and subtracted.

To measure the concentration of Nups, mEGFP–Nup genome-edited cells in interphase were imaged in 3D using a confocal microscope (LSM780; Carl Zeiss, Oberkochen, Germany) and a 40× 1.2 NA C-Apochromat water immersion objective (Carl Zeiss) at 37 °C in a microscope-body-enclosing incubator, under the following conditions: 21 optical sections, section thickness of 2.0 μm, z-stacks every 1.0 μm, and xy pixel size of 0.25 μm. When the nuclear envelope is not perpendicular to the confocal plane of the 3D stacks, the fluorescence intensity at the nuclear envelope is non-isotropic in the point-spread function (PSF), which results in underestimation of the signal. To avoid such underestimation, a single plane was selected that contains the largest nuclear area in which the nuclear envelope is perpendicular to the imaging plane and thus isotropic in the PSF. The fluorescence intensity of Nups was quantified on this single plane using the nuclear envelope mask with the width of three pixels (0.75 μm) that was generated from a SiR–DNA channel. Background fluorescence intensity was measured in wild-type cells without expressing any fluorescent proteins and subtracted. The Nup fluorescence intensity on the nuclear envelope was converted to the concentration using the calibration curve generated by FCS above. The number of Nups per μm$^2$ was calculated from the concentration and then divided by the NPC density per μm$^2$ measured by STED microscopy. This absolute quantification of Nup copy number with FCS calibration was done using 47 mEGFP–Nup107 genome-edited cells in interphase. For other mEGFP-Nups genome-edited cells, their Nup fluorescent intensities on the nuclear envelope were directly compared with the ones of mEGFP–Nup107 genome-edited cells on the same 8-well Lab-Tek Chambered Coverglass, and then their concentrations were determined using the intensity ratios to the mean intensity of mEGFP–Nup107 without using a FCS calibration curve. For Pom121–mCherry, the copy number was quantified independently by performing FCS using Alexa Fluor 568 NHS ester (Thermo Fisher Scientific) to measure the confocal volume and using the cells that transiently express mCherry alone to convert fluorescence intensity to the concentration.

### Measurement of nuclear pore density by STED microscopy

Cells were fixed with 2.4% formaldehyde (Electron Microscopy Sciences) in PBS for 10 min, extracted with 0.4% Triton X-100 (Sigma Aldrich) in PBS for 5 min, and blocked with 5% normal goat serum (Life

Technologies) in PBS for 10 min at room temperature. Subsequently, the cells were incubated overnight at 4 °C with a mouse anti-Nup62 (610497; BD Biosciences, 1.25 µg ml⁻¹) antibody, and then with an Abberior Star Red-conjugated anti-mouse IgG (Abberior, 0.5 µg ml⁻¹) for 30 min at room temperature. For Extended Data Fig. 4b, anti-GFP (Roche, 1:200) and anti-Elys (The Human Protein Atlas, 0.5 µg ml⁻¹) antibodies, Star Red-conjugated anti-rabbit IgG (Abberior, 0.5 µg ml⁻¹) and Star 580-conjugated anti-mouse IgG (Abberior, 0.5 µg ml⁻¹) were used. After multiple washes in PBS, cells were mounted in Vectashield (H-1500, Vector Laboratories). Super-resolution imaging was performed on a Leica SP8 3X STED microscope as described[12]. Images were taken with a final optical pixel size of 20 nm, z-stacks of every 250 nm, and optical section thickness of 550 nm. Images were filtered with a Gaussian filter (kernel size: 0.5 × 0.5 pixels) for presentation purposes. The shrinkage of the nucleus caused by formaldehyde fixation and/ or Vectashield mounting was quantified by comparing the volume of the nuclei of live cells with the ones of fixed cells. The shrinkage was 9.1 ± 2.6% (mean ± s.e.m., n = 36 cells). The NPC density was corrected for the nuclear shrinkage for the calculation of Nup copy number per NPC in Fig. 1d.

To quantify NPC density, the raw STED data were processed in ImageJ[40] with a mean filter (kernel size: 2 × 2 pixels) and a sliding paraboloid (radius: 5 pixels) for background subtraction. Detection of central peak positions for individual NPCs was carried out with the plugin TrackMate[41], using DoG detector and adjusting the detection threshold as the spot diameter size. The resulting 3D NPC coordinates were used to visualize and determine flat and curved regions of the nucleus. Using this map, circular and ellipsoidal regions of interest (ROIs) could then be selected in the flatter parts containing central NPC positions within the z-depth of approx. 500 nm, which corresponds to 2–3 microscopic slices in the images. The remaining signal outside the ROIs, as in curved regions or cytoplasmic structures were discarded from further analysis. NPC densities were calculated for each cell separately by dividing the number of NPCs within the selected ROIs by the corresponding ROI areas. For each cell line, the values were combined to calculate the mean and median NPC density values.

### 3D STED microscopy for visualizing single nuclear pore assembly intermediates

Cells were coated on 18 × 18 mm no. 1.5 square coverslips and synchronized by double thymidine arrest. After 10 h release from the second thymidine treatment, cells were fixed with 2.4% formaldehyde in PBS for 15 min, extracted with 0.25% Triton X-100 (Sigma Aldrich) in PBS for 15 min, and blocked with 2% bovine serum albumin (A2153; Sigma Aldrich) in PBS for 30 min at room temperature. Subsequently, the cells were incubated overnight at 4 °C with rabbit anti-Tpr (HPA019661; The Human Protein Atlas, 1:100) and a GFP nanobody (FluoTag-X4 anti-GFP nanobody Abberior Star 635P; N0304-Ab635P-L; NanoTag Biotechnologies, 1:50), and then with an Alexa Fluor 594 goat anti-rabbit IgG (A-11037; Life Technologies, 1:250) for 30 min at room temperature. After multiple washes in PBS, cells were mounted in ProLong Gold Antifade Mountant (P10144; Invitrogen). Super-resolution imaging was performed on an Abberior STED/RESOLFT Expert Line microscope. Samples were imaged with an UPlan-S Apochromat 100× 1.4 NA oil-immersion objective on an IX83 stand (Olympus). Stimulated depletion was performed using a 775 nm pulsed laser (40 MHz) in combination with 594 and 640 nm pulsed excitation lasers in line switching mode. Fluorescence signal was detected using two separate Avalanche photo diodes with bandpass filters of 605–625 and 650–720 nm. The images were taken with a final optical pixel size of 35 nm, z-stacks of every 200 nm, and optical section thickness of 1,000 nm. For presentation purposes, images were filtered with PureDenoise plugin[42] in ImageJ, and lines with width of 175 nm were drawn along the nuclear envelope, flattened and shown in the figures.

### Quantification of Nup copy number in the cytoplasm and the nucleoplasm and in non-core and core regions of the nuclear envelope

Mitotic cells were imaged and monitored from anaphase onset for 2 h in 3D by confocal microscopy. The microscopy setup and the imaging conditions are described above. Time-lapse imaging for mEGFP-tagged Nups was performed every 30 s. Photobleaching was negligible and thus not corrected. Time-lapse imaging for Pom121–mCherry was carried out every 60 s, and photobleaching was corrected by measuring a fluorescence signal decay in a neighbouring cell in the same field of view. Visualization of the chromosome surface in 3D was done in the Amira software package[43].

To measure Nup accumulation on the nuclear envelope, single planes were selected that contain the largest nuclear area at individual time points to avoid underestimation of the signal as mentioned earlier. The Nup intensity was quantified on the nuclear envelope mask with the width of 0.75 µm that was generated from a SiR–DNA signal at each time point. Except for Nup107 and Seh1, the Nup signal in the cytoplasm and nucleoplasm was measured and used as background. For Nup107 and Seh1, only the cytoplasmic signal was used as background because of their localization at kinetochores. These background values were quantified at individual time points and subtracted from the Nup intensities on the nuclear envelope. The measured Nup intensity was converted into the concentration and then multiplied with nuclear surface area to calculate the total number of the Nups on the nuclear envelope. For Nup153 and Pom121, we did not convert the fluorescence intensity to the concentration as the cell lines were not fully validated to be homozygously tagged.

The Nup copy number was also calculated in the cytoplasm and the nucleoplasm during the first 2 h after anaphase onset. The cytoplasm mask was created by subtracting a mask of the nucleus generated from a SiR–DNA signal from a mask of the whole cell generated from a mEGFP–Nup signal. The mask for the nucleoplasm was created by eroding three pixels of the nuclear mask generated from a SiR–DNA signal. The Nup fluorescence intensity was quantified on these cytoplasmic and nucleoplasmic masks over time and then converted into the concentration. To calculate the total number of Nups in the cytoplasm and the nucleoplasm, the measured concentration was multiplied by the volume of the respective compartments. Cytosol volume at metaphase was quantified using the cytosolic signal of the Nups. The cytosolic GFP signal of some of the Nups (for example, Nup205, Nup214, Tpr and Nup358) were too low to precisely segment the cytosol in the z-slices close to the glass surface due to the relatively high background fluorescence signal. Therefore, we measured the cell volume using the brightest Nup62–mEGFP cell line (4,900 ± 330 µm³ (mean ± s.d. from 8 cells)), and for the other Nup cell lines, we measured the cytosol area on a middle z-slice plane and calculated the ratios to the area for the Nup62 cell line (the ratios were 0.97, 0.97, 1.09, 1.14, 1.03, 1.07 and 1.02 for Nup107, Seh1, Nup205, Nup93, Nup214 and Nup358, respectively). From the measured volume of the Nup62 cell and the ratios of cytosol area to the Nup62 cell, we estimated the volume for the other cell lines (4,670, 4,660, 5,550, 6,030, 5,110, 5,420 and 5,080 µm³ for Nup107, Seh1, Nup205, Nup93, Nup214 and Nup358 lines, respectively). For the cytoplasm volume after anaphase, we used the data that were measured previously in histone H2b–mCherry-expressing HeLa cells using fluorescently labelled dextran[17]. Assuming that the cell volume changes in the same degree during mitosis exit as the H2b–mCherry cell, we calculated the cytoplasmic volume of the Nup cell lines using the ratio to the volume at metaphase (5,300 µm³ for the H2b–mCherry cell[17]). The nucleoplasmic volume was quantified in each mEGFP–Nup knock-in cell line using a SiR–DNA signal at every time points as described previously[12].

Core regions were predicted on the nuclear envelope based on a previously described protocol using the core marker Lap-2α[12]. In brief, nuclear volume was segmented using SiR–DNA fluorescence signals

that were processed with a 3D Gaussian filter and a multi-level thresholding. Nuclear volume was then divided into inner and outer volumes using the cutting plane that was constructed from the largest eigenvector and the second one orthogonal to the first vector of the pixel coordinates of the nuclear volume. Surface area of each nucleus was calculated and utilized to adjust the size of the inner and outer-core regions at individual time points. The previously defined criteria for being core and non-core regions[12] was applied. The position of inner and outer core was determined with respect to the intersection point of the largest eigenvector on the nuclear surface. The core region prediction was done in MATLAB (MathWorks).

## Mathematical modelling for the nuclear pore assembly kinetics

Previous EM data showed that within 2 h after the onset of anaphase, postmitotic assembly is the dominant process in the non-core region, whereas slower interphase assembly predominates in the core region[12]. Assuming that this relation is also reflected in the live-cell Nup dynamics, we derived a mathematical model. We assumed that the observed total fluorescence intensity in the non-core, $n(t)$, and core region, $c(t)$, at time $t$ after anaphase onset is a linear combination of the postmitotic, $pm(t)$, and interphase assembly, $ip(t)$, according to

$$n(t) = f_n \, pm(t) + (1 - f_n) \, ip(t) \tag{1}$$

$$c(t) = f_c \, pm(t) + (1 - f_c) \, ip(t). \tag{2}$$

The fraction of postmitotic assembly in the non-core and core regions are denoted $f_n$ and $f_c$, respectively. To test this assumption and obtain an estimate of the fractions, we used a model that accounts for the observed sigmoid-like kinetics

$$pm(t) = \frac{(t - d_{n/c})^{n_p}}{(t - d_{n/c})^{n_p} + K_p^{n_p}}, \text{ for } t \geq d_{n/c} \tag{3}$$

$$ip(t) = \frac{(t - d_{n/c})^{n_i}}{(t - d_{n/c})^{n_i} + K_i^{n_i}}, \text{ for } t \geq d_{n/c} \tag{4}$$

and $pm(t < d_{n/c}) = ip(t < d_{n/c}) = 0$. The parameters $n_p$, $K_p$ and $n_i$, $K_i$ characterize the postmitotic and interphase kinetics, respectively. The parameter $d_{n/c}$ accounts for an additional delay in NPC initiation. In the non-core region we assumed $d_n = 0$; in the core region due to the presence of kinetochore microtubule fibres[26], $d_c > 0$. The model is used to derive the underlying postmitotic and interphase assembly kinetics. Using equations (1 and 2), we obtain

$$ip(t) = \frac{f_n \, c(t) - f_c \, n(t)}{f_n - f_c}. \tag{5}$$

$$pm(t) = \frac{(1 - f_c) \, n(t) - (1 - f_n) \, c(t)}{f_n - f_c}. \tag{6}$$

**Parameter estimation.** For each Nup, there are two parameters that define the interphase ($n_i$, $K_i$) and postomitic ($n_p$, $K_p$) assembly, respectively. The fractions $f_n$ and $f_c$ and the delay $d_c$ in the core region were estimated globally for all Nups. In total, we have 43 parameter, $4 \times 10 = 40$ parameters describing the assembly kinetics and 3 global parameters, fitted to 4,446 data points.

In detail, to find the model parameters we minimized the mean squared distance between data and model for all the time points $M$

$$\chi^2 = \sum_{j=1}^{M} \left( \left( \frac{N(t_j) - n(t_j)}{\sigma_N(t_j)} \right)^2 + \left( \frac{C(t_j) - c(t_j)}{\sigma_C(t_j)} \right)^2 \right). \tag{7}$$

$N(t_j)$ and $C(t_j)$ are the mean background-subtracted and normalized fluorescence intensities in the non-core and core region with standard deviation $\sigma_N(t_j)$ and $\sigma_C(t_j)$, respectively, at time point $t = t_j$. We subtracted a background computed from the average of the first 3 time points. The data were normalized with the average value between 100 and 120 min after anaphase. In a first step, for each protein and cell line, we estimated the postmitotic fractions in the core and non-core region and the kinetic parameters. Overall, we computed 61 parameters (6 parameters per protein plus one delay parameter). For the postmitotic fractions, we obtained on average $f_n = 0.857$ [0.76, 0.95] and $f_c = 0.295$ [0.17, 0.4], where the number in brackets indicates the 95% confidence interval estimated using the profile likelihood method[44]. Importantly, the obtained postmitotic fractions are well in agreement with the previously reported estimates obtained from EM data[12]. The delay in pore formation between core and non-core region was estimated by systematically varying $d_c$ from 0 to 6 min in steps of 1 min. A value of $d_c = 2$ min, gave optimal result. In a second step, we used the previously estimated average postmitotic fractions, $f_n$ and $f_c$, and $d_c$ and recomputed the kinetics parameters for each protein. The model with reduced parameters well agrees with the data (Extended Data Figs. 5 and 6, $R^2 > 0.99$). To verify if the choice of common postmitotic fractions for all Nups is valid, we computed the Bayesian information criterion (BIC)[45]. The difference in BIC between the model with reduced parameters, 43 parameters, compared to the full model, 61 parameters, was −7, indicating that the model with reduced parameters is justified. The obtained parameter values are listed in Supplementary Table 3.

**Median assembly time and duration of assembly.** One can think of an assembly curve as a cumulative distribution function and its derivative as the corresponding probability density function (PDF) for binding events. The median assembly time, i.e. the time where 50% of binding events have occurred, is $K_i$ and $K_p$, for the interphase and postmitotic assembly respectively. We consider the time intrinsic to the assembly mechanism and independent of the initiation delay $d_c$. We further define the duration of assembly as the time interval where 80% of binding events occur, i.e. from a fraction of $\alpha_1 = 0.1$ up to $\alpha_2 = 0.9$, one obtains (see also Extended Data Fig. 7a)

$$\Delta T_p = K_p \left( \left( \frac{\alpha_2}{1 - \alpha_2} \right)^{\frac{1}{n_p}} - \left( \frac{\alpha_1}{1 - \alpha_1} \right)^{\frac{1}{n_p}} \right) \tag{8}$$

$$\Delta T_i = K_i \left( \left( \frac{\alpha_2}{1 - \alpha_2} \right)^{\frac{1}{n_i}} - \left( \frac{\alpha_1}{1 - \alpha_1} \right)^{\frac{1}{n_i}} \right). \tag{9}$$

The duration of assembly quantifies the width of the binding events PDF.

In Extended Data Fig. 7c we see a strong positive correlation between median assembly time and duration of assembly for the postmitotic assembly. This suggests a sequential assembly mechanism. The rationale is that for a strict sequential pathway, the binding events PDF for subsequent Nup is a convolution of all previous binding events PDF and so will broaden for later binding proteins. For example, if an early protein has an assembly duration of 1 h, within this time window binding sites for the subsequent protein will continue to appear. Therefore, the subsequent protein in the sequence will also accumulate for at least 1 h.

For the simplified case of irreversible sequential assembly and linear rate constants, a positive correlation can be demonstrated. This correlation is independent on whether the initiation is synchronous or spread within a time-period. For simplicity reasons, we omit the fact

that each Nup binds in multiple copies to the NPC. A Nup $P_i$ binds with rate constant $\kappa_i$ to a NPC intermediate $X_{i-1}$ according to the reaction scheme in Supplementary Fig. 2.

The corresponding system of ordinary differential equations is

$$\frac{dX_0}{dt} = g(t) - k_1 X_0$$

$$\frac{\mathrm{d}X_i}{\mathrm{d}t} = k_i X_{i-1} - k_{i+1} X_i, \text{ for } n > i > 0$$

$$\frac{\mathrm{d}X_n}{\mathrm{d}t} = k_n X_{n-1}$$

with $X_i(0) = 0$, $k_i = \kappa_i P_i^{\text{free}}$, and an excess of free Nup $P_i^{\text{free}} \approx$ constant. The function $g$ accounts for the appearance of NPC initiation sites after anaphase. We assume that NPC initiation is completed within a finite time after anaphase and neglect the slowly and continuous appearance of pores in G1—that is, $\int_0^\infty g(t)\mathrm{d}t =$ constant. The amount of $i$th Nup bound to a complex is $P_i = \sum_{j=i}^n X_j$ and its time derivative $f_i(t) = \frac{\mathrm{d}P_i}{\mathrm{d}t} = k_i X_{i-1}$ is proportional to the binding events PDF. We can quantify the assembly time $\tau_i$ and assembly duration $\theta_i$ by the mean and standard deviation of the binding events PDF, respectively[46],

$$\tau_i = \frac{\int_0^\infty t f_i(t)\mathrm{d}t}{\int_0^\infty f_i(t)\mathrm{d}t}$$

and

$$\theta_i = \sqrt{\frac{\int_0^\infty t^2 f_i(t)\mathrm{d}t}{\int_0^\infty f_i(t)\mathrm{d}t} - \tau_i^2}$$

After integration by parts, one obtains

$$\tau_i = \tau_g + \sum_{j=1}^i \frac{1}{k_j} \tag{10}$$

and

$$\theta_i = \sqrt{\theta_g^2 + \sum_{j=1}^i \frac{1}{k_j^2}}. \tag{11}$$

Where $\tau_g = \frac{\int_0^\infty t g(t)\mathrm{d}t}{\int_0^\infty g(t)\mathrm{d}t}$ and $\theta_g = \sqrt{\frac{\int_0^\infty t^2 g(t)\mathrm{d}t}{\int_0^\infty g(t)\mathrm{d}t} - \tau_g^2}$ are the mean and standard deviation of the initiation function, respectively. From equations (10 and 11), it is clear that $\tau_{i+1} > \tau_i$ and $\theta_{i+1} > \theta_i$ for any parameter combinations and independently on how synchronous the initiation of pores is. This shows that for a strictly sequential pathway a linear correlation between assembly time and duration is expected.

## Integrative modelling of the NPC assembly pathway

A model of the assembly pathway is defined by a series of static structures, including a static structure at each sampled time point along the assembly process. Therefore, we model the NPC assembly by first modelling static structures at each time point, independently from each other. We then enumerate alternative assembly pathways and rank them based on the static structure scores and plausibility of transitions between successive static structures.

**Integrative modelling of static structures at each time point.** The static structures are modelled by standard integrative structure modelling[30] as follows.

**Representing a static structure model.** The time points correspond to times with available electron tomography protein densities[11]: 5 min, 6 min, 8 min, 10 min and 15 min after anaphase onset. We divide the mature NPC structure (PDB: 5A9Q and 5IJO) into eight spokes and further divide each spoke into a set of rigid subcomplexes, including the Y-complex, the inner-ring Nup205–Nup155–Nup93 subcomplex, the inner-ring Nup93–Nup188–Nup155 subcomplex, and the central channel Nup62–Nup58–Nup54 subcomplex. For each domain, we coarse-grained the structure by grouping ten consecutive amino acid residues into a single bead at the centre of mass of those residues. Each subcomplex is represented as a rigid body. The nuclear envelope is represented as a fixed toroid surface embedded in two parallel planes. Thus, the variables of the model include the Euclidean coordinates of the Nup subcomplexes and the copy number of each Nup subcomplex.

We set the inner pore diameter and minor radius of the pore at each time point to the mean of previously determined nuclear envelope cross sections[11] with a pore diameter of 51.5 nm, 58.4 nm, 72.7 nm, 84.6 nm, 79.8 nm and 87 nm; and minor radius of 21.4 nm, 21.2 nm, 21.5, 20.3 nm, 17.1 nm and 15 nm for time points at 5 min, 6 min, 8 min, 10 min, 15 min and the mature pore, respectively.

**Scoring a static structure model.** The copy numbers of the NPC sub-complexes at each time point were restrained by a Gaussian function with mean and variance determined by the single-cell traces presented in this study. The relative likelihood of a set of copy numbers is proportional to the product of individual Gaussian likelihoods.

Distances between pairs of Nups that are in contact with each other in the native NPC structure[27,28] were restrained by a harmonic Gō-like model[47]. Inter-subcomplex contacts within 5 nm in the mature structure were restrained by a harmonic function (strength 0.01 kcal mol$^{-1}$ Å). Each Gō-like scoring term was scaled at each time point, from zero at the first time point to full strength at the mature pore time point. Distances between all pairs of Nups were also restrained by a harmonic excluded volume restraint (strength 0.01 kcal mol$^{-1}$ Å$^{-1}$). Proximity between Nup domains containing a membrane interacting ALPS-motif and the nuclear envelope was restrained by a harmonic term (strength 0.1 kcal mol$^{-1}$ Å$^{-1}$), based on their sequences. Overlap between the Nups and nuclear envelope surface was avoided by imposing a harmonic repulsion between the Nups and nuclear envelope surface (strength of 0.01 kcal mol$^{-1}$ Å$^{-1}$).

The shape of a static structure was restrained by a correlation coefficient between the model and electron tomography protein density[11]. The forward model density was represented by fitting each Nup sub-complex with a Gaussian mixture model of two components per subcomplex copy using the gmconvert utility[48]. Similarly, the electron tomography protein densities at each time point were represented with a Gaussian mixture model with 150 components fit to the experimental density.

**Sampling static structure models.** A state of the NPC at any given time point is defined by the copy numbers and coordinates of its components. Only copy number assignments and structures consistent with the C-8 symmetries were sampled. The assumption that the sub-complexes preform with 8-fold multiplicity before assembling into the NPC is supported by our previous electron microscopy analysis[11]. The averaged electron tomograms of the intermediates demonstrated 8-fold symmetry of the outer-ring complex at 5, 6, 8, 10, and 15 min after anaphase onset and 8-fold symmetry of the inner-ring complex at 10 and 15 min after anaphase onset[11], indicating that the majority of intermediates have approximate 8-fold symmetry for the outer- and inner rings. However, there are likely to be variations in the copy numbers at the single pore level. We only sampled structures for the top 20-scoring Nup copy number combinations. Each sampling started with the mature pore structure, followed by applying 10$^6$ Monte Carlo moves. These

moves included rotational and translational perturbations to each Nup subcomplex, drawn from a uniform distribution in the range from −0.04 to +0.04 radians and from −4 to +4 Å, respectively.

**Modelling the assembly pathway.** With the static structure models in hand, we connect them into complete alternative assembly pathways, as follows.

Each pathway is represented by a static structure at each sampled time point, starting with $t = 5$ min and culminating in the native structure; we do not model the completely disassembled NPC. The score of a pathway is the sum of the scores for the static structures on the pathway (defined above) and transitions between them. A transition score is uniform for all allowed transitions. A transition between two successive static structures is allowed if the subcomplexes in the first structure are included in the second structure. All possible pathways were enumerated, scored, and ranked. The best-scoring pathways were extracted for further analysis (Fig. 5). Molecular visualization was performed with UCSF ChimeraX[49].

## Immunofluorescence for visualizing Nup155 during assembly

Cells were treated, fixed and blocked as described in the previous section (STED microscopy for visualizing single nuclear pore assembly intermediates). Afterwards, cells were incubated overnight at 4 °C with rabbit anti-Nup155 (HPA037775; The Human Protein Atlas, 1:100), and then with an Abberior Star 635P goat anti-rabbit IgG (ST635P-1002-500UG, Abberior GmbH, 1:250) for 30 min at room temperature. After multiple washes in PBS, cells were mounted in ProLong Gold Antifade Mountant with DAPI (P36941, Invitrogen). Imaging was performed in 3D using a confocal microscope (LSM780; Carl Zeiss) and a 63× 1.4 NA Plan-Apochromat objective (Carl Zeiss) under the following conditions: 31 optical sections, section thickness of 0.7 μm, z-stacks of every 0.39 μm, and xy pixel size of 0.13 μm.

## Sample size determination and statistical analysis

For quantitative imaging in Fig. 1a,d, the data were from 4, 4, 4, 2, 3, 2, 3 and 2 independent experiments for Nup107, Seh1, Nup205, Nup93, Nup62, Nup214, Tpr and Nup358, respectively. STED imaging in Fig. 4a and Extended Data Fig. 4a and live imaging in Extended Data Fig. 10 were from two independent experiments. For dynamic quantitative imaging in Fig. 3, the data were from 4, 4, 4, 2, 3, 4, 3, 2, 2 and 4 independent experiments for Nup107, Seh1, Nup205, Nup93, Nup62, Nup214, Tpr, Nup358, Nup153 and Pom121, respectively. STED imaging in Figs. 1b,c and 4b, Southern blotting in Extended Data Fig. 1 and immunofluorescence microscopy in Extended Data Fig. 9 are from single experiments. Statistical analyses were performed only after all the data were taken. Sample sizes for each experiment are indicated in figure legends. Sample sizes were based on pilot experiments to determine the number of cells required to observe stable population averages with high Pearson's correlation between replicates. No blinding and randomization was done, as this study does not involve animals or human participants. Samples were organized into groups based on cell lines. Cells were imaged in randomly chosen fields of view per experiment. All imaged cells were further analyzed. Videos of dividing cells with rotating nuclei are removed from the analysis, because we cannot properly assign the non-core and core regions.

## Reporting summary

Further information on research design is available in the Nature Portfolio Reporting Summary linked to this article.

## Data availability

Fluorescence images were deposited to the Image Data Resource (IDR; https://idr.openmicroscopy.org/) under accession number idr0115.

Our integrative spatiotemporal model of the postmitotic assembly of the human NPC has been deposited to the nascent database of integrative structures PDB-Dev (https://pdb-dev.wwpdb.org/) under accession number PDBDEV_00000142. Source data are provided with this paper.

## Code availability

All source code is accessible on Github repository (for quantifying Nup intensity in core and non-core regions: https://github.com/mjh1m22/Quantitative_map_nups_Otsuka_Nature_2022; for decomposing postmitotic and interphase assembly kinetics: https://github.com/manerotoni/npc_assembly_Otsuka_2022). Integrative Modeling Platform (IMP) is an open-source program freely available under the LGPL license at http://integrativemodeling.org; all input files, scripts, and output files are available at http://integrativemodeling.org/npcassembly.

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

**Acknowledgements** We thank the EMBL Advanced Light Microscopy Facility (ALMF) for their support in STED super-resolution microscopy; S. Schnorrenberg for help with 3D STED; A. Brunner for help with automated live-cell imaging; B. Webb for help with IMP and J. Baker-Lepain for managing the Wynton computer cluster at QBI@UCSF. This work was supported by grants from the Baden Wuerttemberg Foundation (J.E. and A.S.), NIH/NIGMS R01GM083960 (A.S.), NIH/NIGMS P41GM109824 (A.S.), NIH/NIGMS R01GM112108 (A.S.) and the European Molecular Biology Laboratory (EMBL; S.O., A.Z.P., A.R., M.J.H., M.K., B.K. and J.E.). S.O. and A.R. were further supported by the EMBL Interdisciplinary Postdoc Programme (EIPOD) under Marie Curie Actions COFUND. S.O. was additionally supported by a Japan Society for the Promotion of Science fellowship (postdoctoral fellowship for research abroad).

**Author contributions** S.O., J.O.B.T., A.S. and J.E. designed the project. S.O. performed all the fluorescence microscopy experiments and analyses, except for the experiments in Fig. 4 and Extended Data Figs. 9 and 10. J.O.B.T. performed integrative structural modelling. W.Z. performed 3D STED, immunofluorescence time-course microscopy and live imaging shown in Fig. 4 and Extended Data Figs. 9 and 10. A.Z.P. carried out mathematical modelling for the nuclear pore assembly kinetics. A.R. developed an analysis pipeline for nuclear pore density measurement. M.J.H. established a computational image analysis pipeline to quantify fluorescence intensities in non-core and core regions in 3D time-lapse images. W.Z., M.K., A.C., B.K. and N.R.M. generated and validated genome-edited cell lines. A.S. and J.E. supervised the work. S.O., J.O.B.T., A.S. and J.E. wrote the paper. All authors contributed to the analysis and interpretation of data and provided input on the manuscript.

**Competing interests** The authors declare no competing interests.

**Additional information**
**Correspondence and requests for materials** should be addressed to Shotaro Otsuka or Jan Ellenberg.

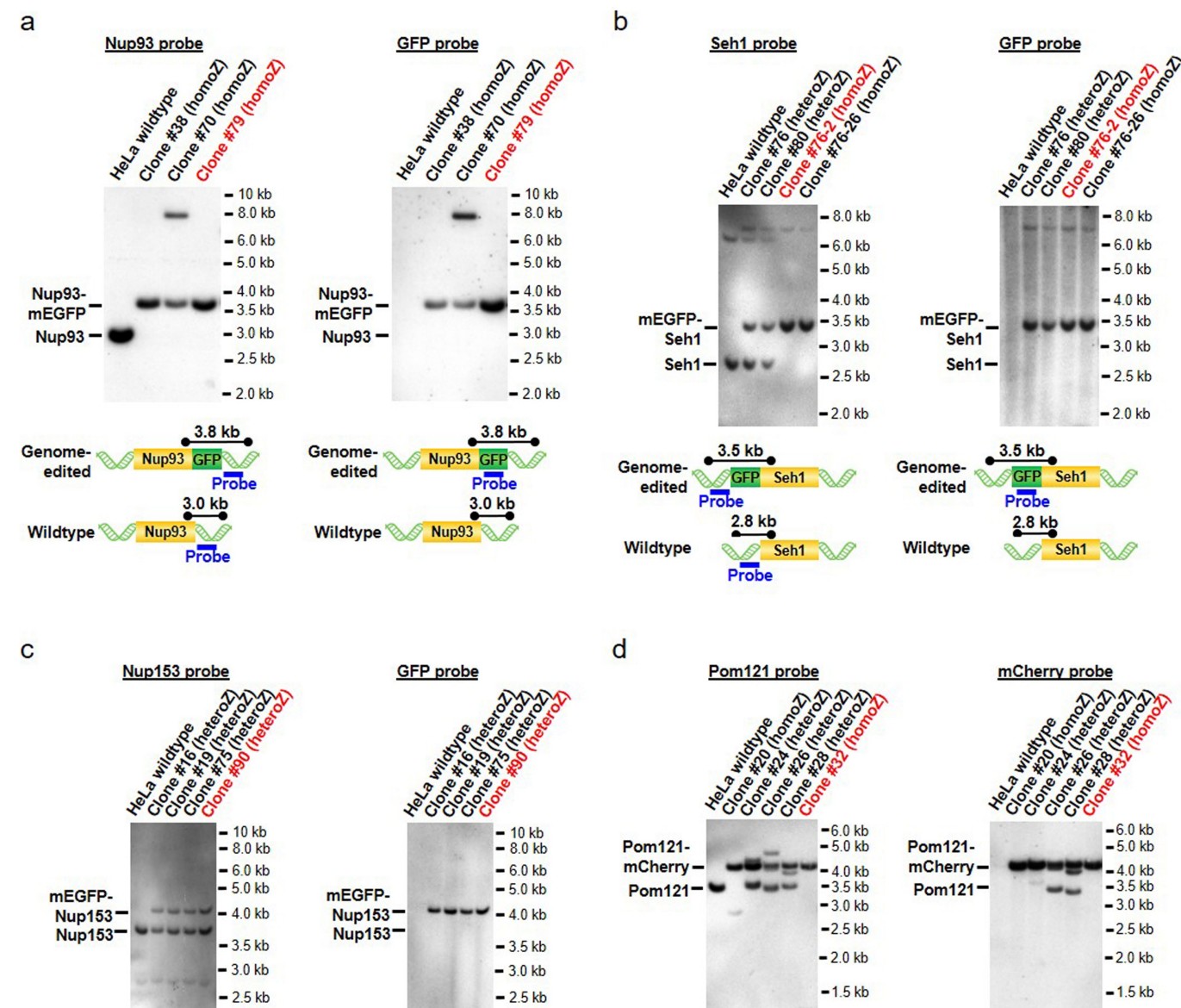

**Extended Data Fig. 1 | Southern blotting of genome-edited cell clones.**
Genomic DNA of each cell clone was digested with restriction enzymes and the fragments were detected by probes for Nup93 (**a**), Seh1 (**b**), Nup153 (**c**), and Pom121 (**d**), as well as GFP (**a**–**c**) and mCherry (**d**). The size of the fragments and the probe-binding regions are illustrated in the bottom panels. The clones indicated in red were used for this study. We regarded the clone #32 of Pom121-mCherry as subhomozygous because the protein abundance was much less than expected although the blot indicated homozygous tagging. The following Nup cell lines were validated to be homozygously-tagged in previous reports (Tpr, Nup214 and Nup358[17]; Nup107[12]; Nup205[11]; Nup62[18]).

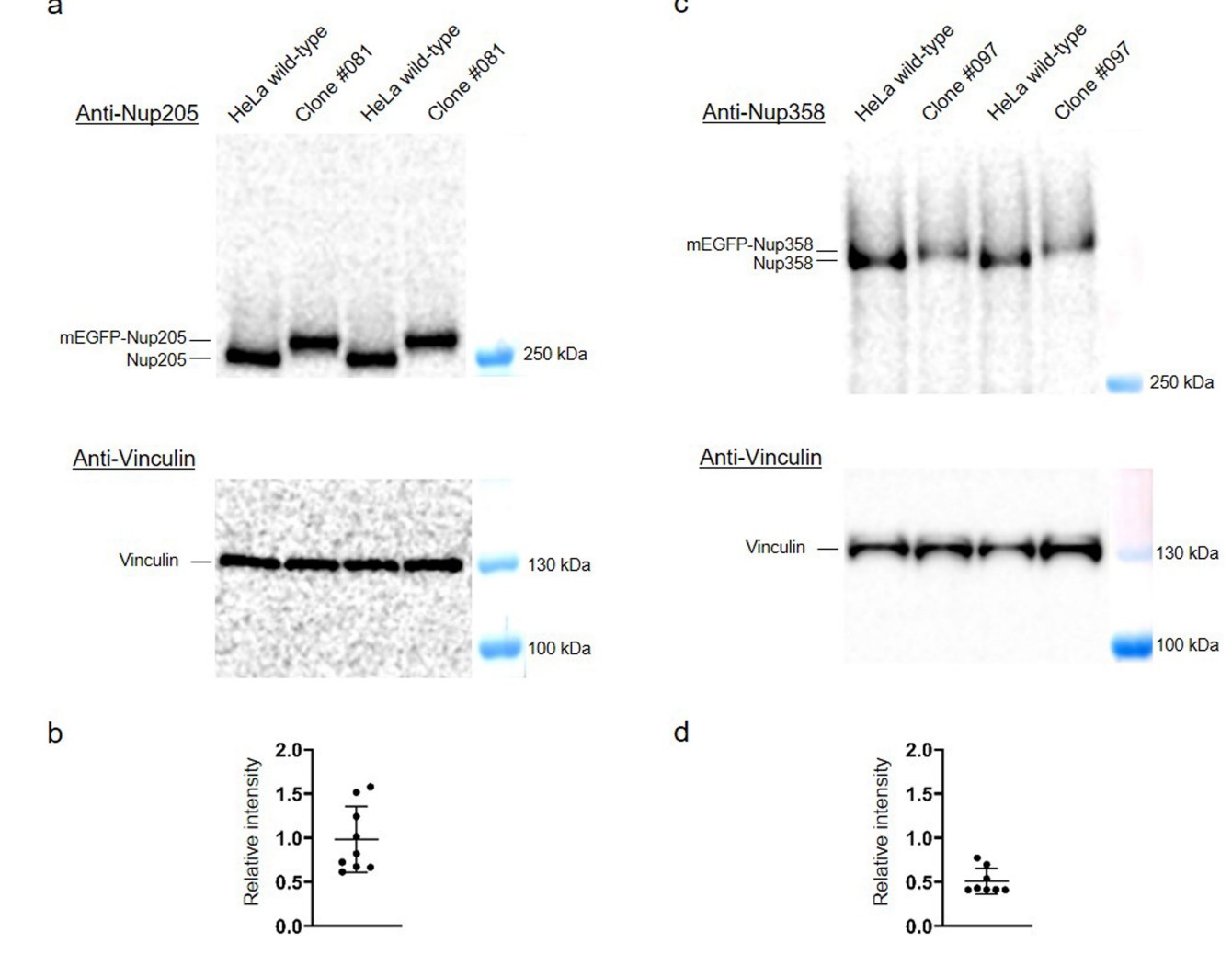

**Extended Data Fig. 2 | Immunoblot analysis of mEGFP-Nup205 (a,b) and mEGFP-Nup358 (c,d) cell lysates.** The membranes were cut at the position of around 200 kDa. The membranes above 200 kDa were blotted with antibodies against Nup205 (**a**) and Nup358 (**c**), and the membranes below 200 kDa were blotted with an antibody against Vinculin for loading control. The clone 081 (**a**) and 097 (**c**) are the cell lines that are used in this study. (**b,d**) The relative intensity of the bands for GFP-tagged Nups over endogenous Nups. The plots for Nup205 and Nup358 are from 9 and 8 pairs of bands (GFP-tagged vs endogenous) from 4 and 3 independent experiments, respectively. The mean is depicted as a horizontal line and the whiskers show the standard deviation.

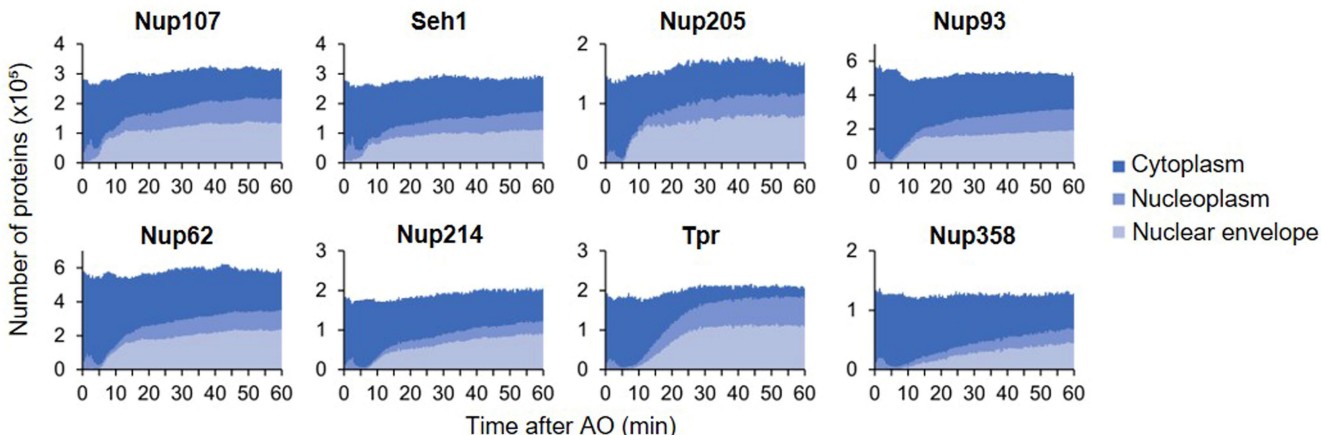

**Extended Data Fig. 3 | NPC assembly relies on and consumes almost half of the material inherited from the mother cell within one hour after mitosis.** FCS-calibrated 3D confocal microscopy was performed as in Fig. 2, and the Nup copy number was quantified in whole dividing cells (for details see Methods). The number of Nups in cytoplasm (dark blue), nucleoplasm (medium dark blue) and nuclear envelope (light blue) are plotted against time after anaphase onset. The plot is the mean of 15, 20, 13, 14, 22, 22, 19, and 24 cells for Nup107, Seh1, Nup205, Nup93, Nup62, Nup214, Tpr and Nup358, respectively.

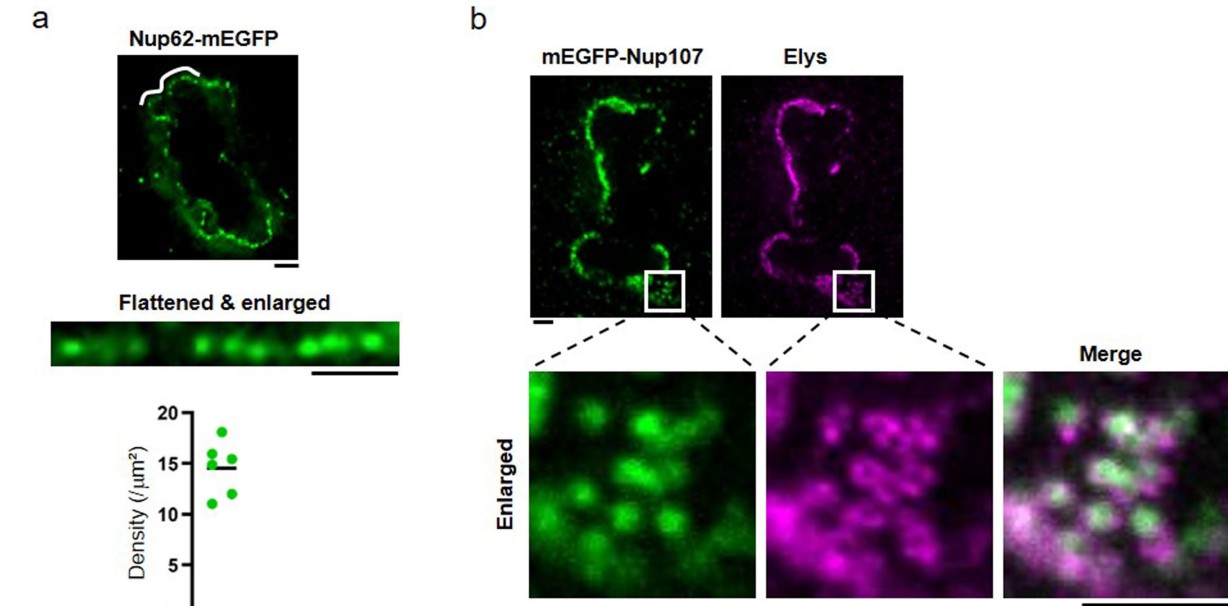

**Extended Data Fig. 4 | GFP-tagged Nups are recruited to the NPCs rather than nonspecifically accumulated on the NE. a**, Three-dimensional stimulated emission depletion (3D-STED) imaging of a Nup62-mEGFP genome-edited cell at early telophase. The region of the nuclear envelope indicated by a white line in the top image is flattened and shown in the bottom. Scale bars, 1 μm. The density of the Nup62 spots was 14.6 ± 2.6 NPCs/μm² (from 6 cells), which is consistent with our previous EM observation of the NPC density of 12–16 NPCs/μm² in early telophase cells[11]. **b**, 2D-STED imaging of a mEGFP-Nup107 genome-edited cell at early telophase that were stained with anti-GFP and anti-Elys antibodies. Enlarged images indicated by white boxes in the top images are shown in the bottom panel. Scale bars, 1 μm.

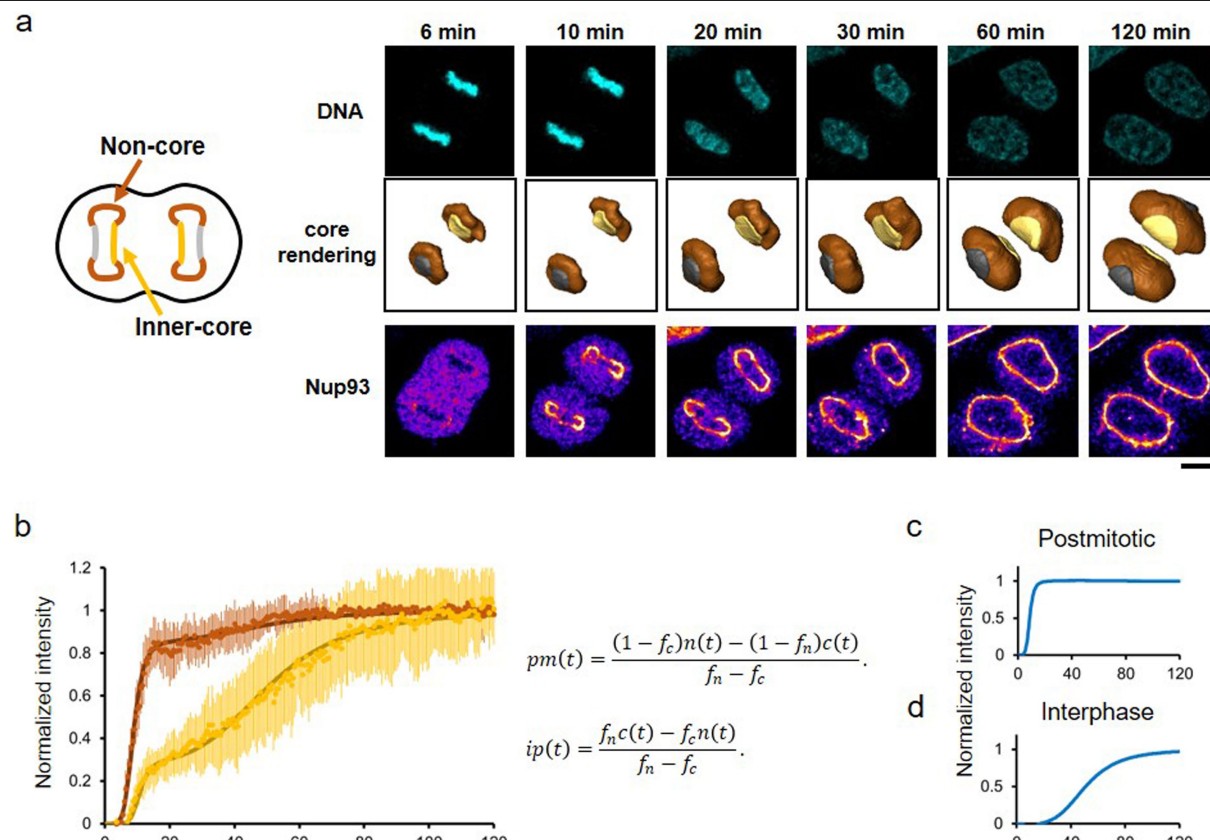

$$pm(t) = \frac{(1 - f_c)n(t) - (1 - f_n)c(t)}{f_n - f_c}.$$

$$ip(t) = \frac{f_n c(t) - f_c n(t)}{f_n - f_c}.$$

**Extended Data Fig. 5 | Postmitotic and interphase assembly are spatially distinguished for the first hour after mitotic exit and thus their kinetics can be decomposed. a**, Time-lapse 3D imaging of Nup93-mEGFP genome-edited cell. Single confocal sections of SiR-DNA and GFP channels are shown. Images were filtered with a median filter (kernel size: 0.25 × 0.25 μm). Scale bar, 10 μm. Time after AO is indicated. **b**, The fluorescence intensities at non-core (brown) and inner-core (yellow) regions were quantified. Dots represent the average and s.d. of measurements from 14 cells. The intensities were fitted with a sequential model of NPC assembly (bold lines) that allows for different population and rate constants for postmitotic and interphase assembly (described in detail in Methods and Supplementary Table 3). **c**,**d**, Decomposed kinetics of postmitotic (**c**) and interphase (**d**) assembly from (**b**).

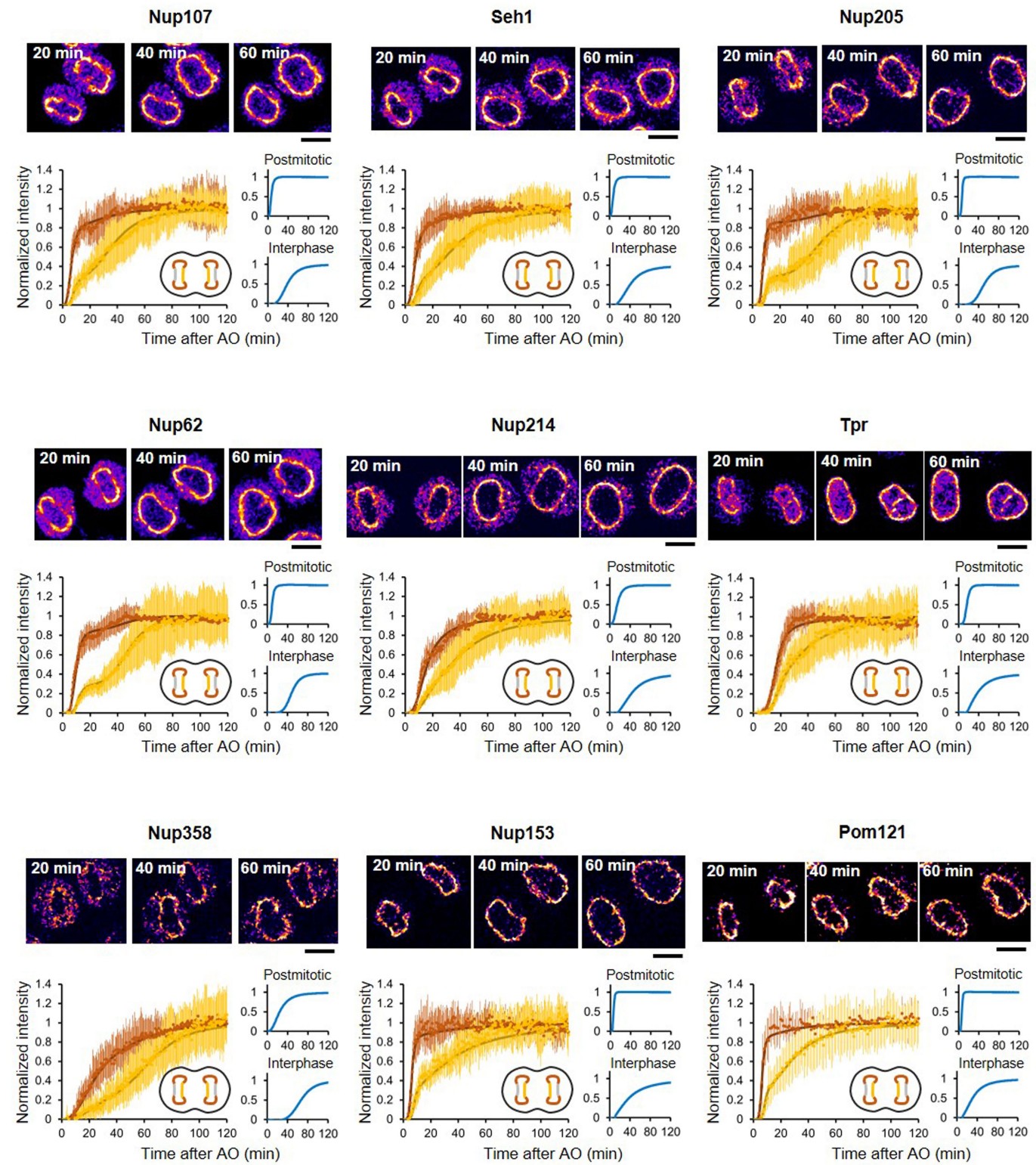

**Extended Data Fig. 6 | Kinetic decomposition of the two assembly processes for each nucleoporin.** The fluorescence intensities at non-core (brown) and inner-core (yellow) regions were plotted and fitted with a sequential model of NPC assembly (bold lines) as in Extended Data Fig. 5. Dots represent the average and s.d. of measurements from 15, 20, 13, 14, 22, 22, 19, 24, 14 and 13 cells for Nup107, Seh1, Nup205, Nup93, Nup62, Nup214, Tpr, Nup358, Nup153, and Pom121, respectively. Single confocal slices of cells at 20 min, 40 min, and 60 min after AO are shown. Images were filtered with a median filter (kernel size: $0.25 \times 0.25\,\mu m$) for presentation purposes. Scale bars, 10 μm.

a

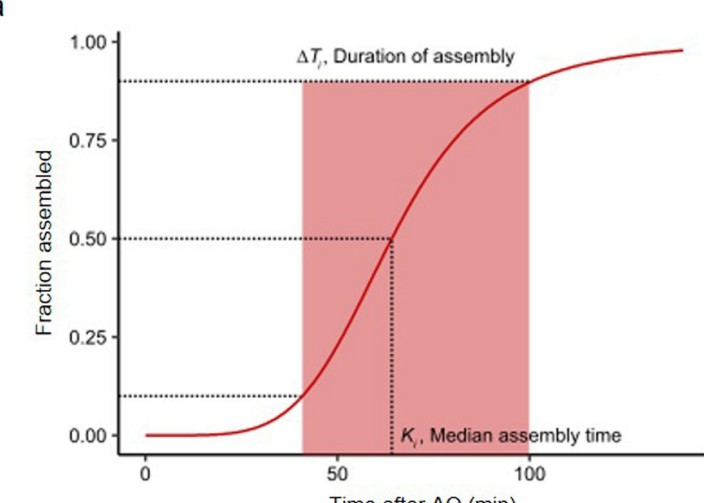

b

### Postmitotic assembly

### Interphase assembly

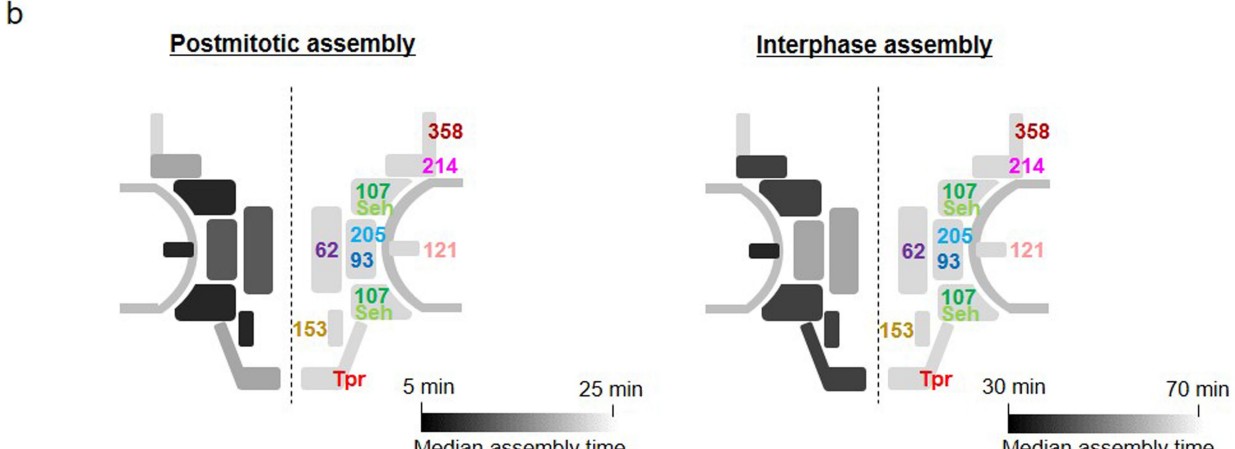

c

### Postmitotic assembly

### Interphase assembly

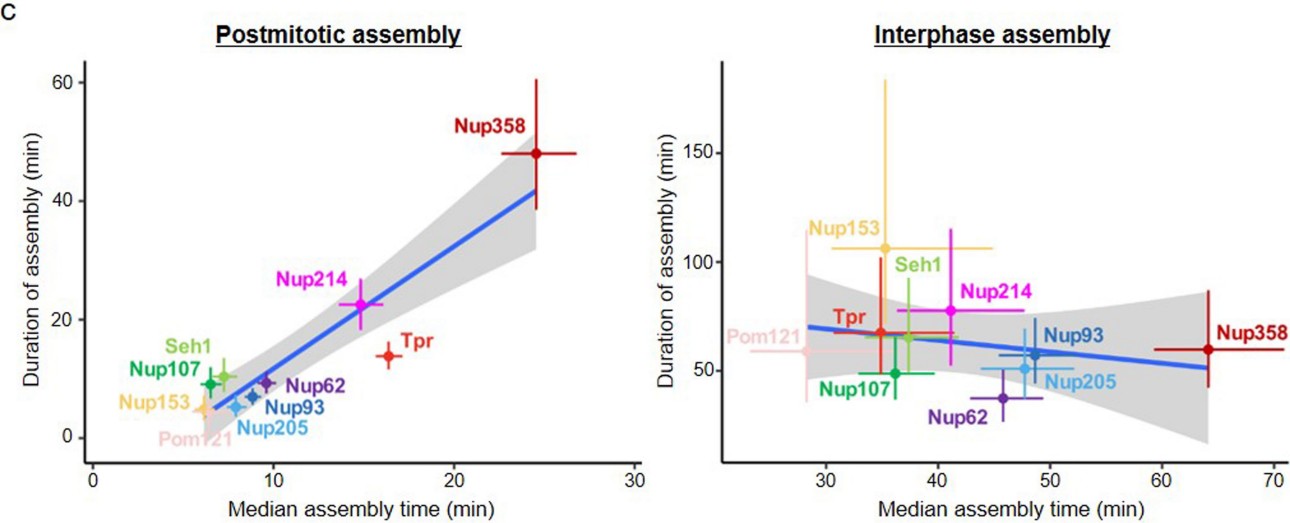

**Extended Data Fig. 7** | See next page for caption.

**Extended Data Fig. 7 | Computed parameters for the nucleoporin assembly kinetics and the remarkable difference between postmitotic and interphase assembly. a**, An example of computed quantities (Nup358, interphase assembly). **b**, Illustrations of median assembly time of nucleoporins in postmitotic (left) and interphase (right) assembly superimposed onto a simplified scheme of the NPC. The median assembly time of individual nucleoporins are displayed on a pseudo-colour scale. **c**, Plots of Nup assembly duration along the median assembly time for postmitotic and interphase assembly pathways. The crosses indicate the 95% confidence intervals. The dots are the mean values. Values are listed in Supplementary Table 3. The long straight line shows the result of a linear regression to the mean values. The gray area is the 95% confidence interval of the linear regression. Theoretically, for a sequential assembly mechanism where late steps depend on early steps, the observed ensemble kinetics of a late binding protein must contain the history of all previous events (see Methods for details). Indeed, postmitotic assembly showed a strong positive linear correlation, indicating a sequential assembly mechanism, which for example implies that Nup62 is incorporated into the NPC before Tpr can bind.

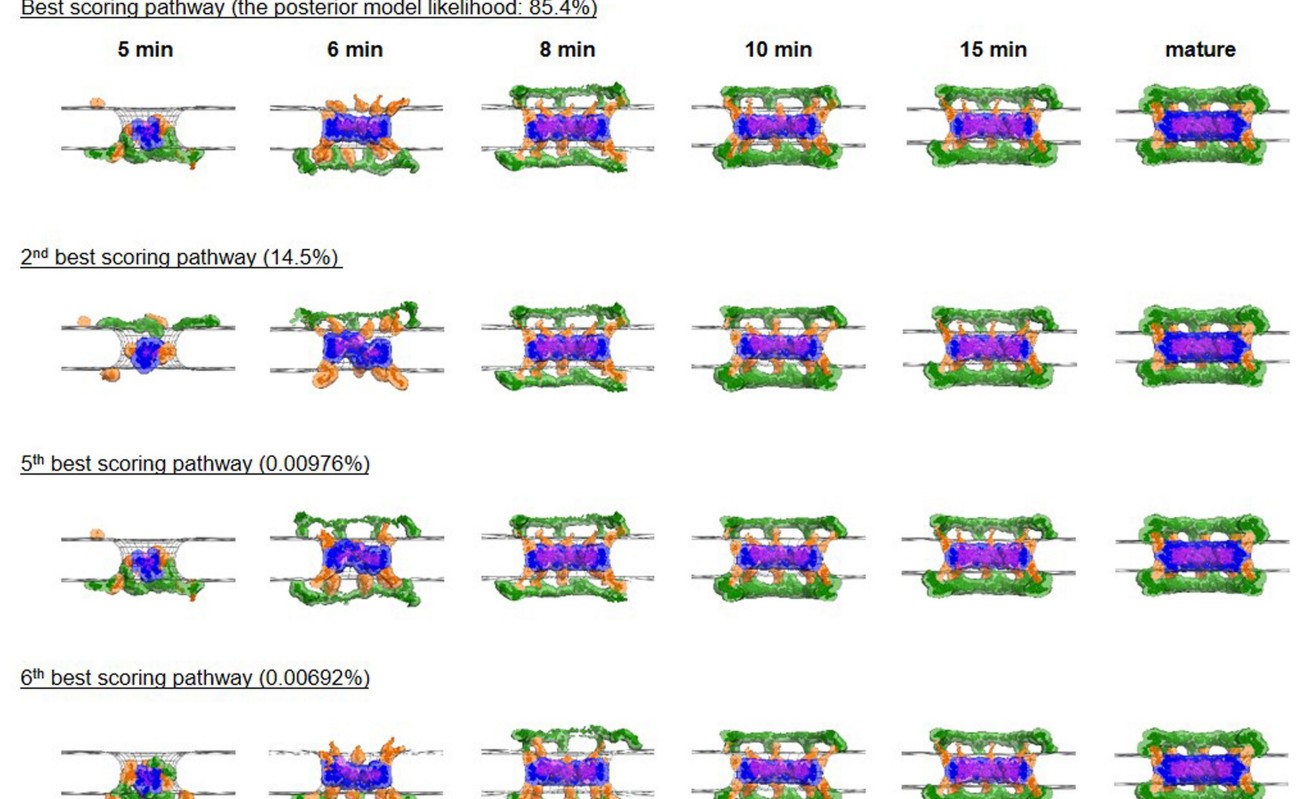

Best scoring pathway (the posterior model likelihood: 85.4%)

| 5 min | 6 min | 8 min | 10 min | 15 min | mature |

2nd best scoring pathway (14.5%)

5th best scoring pathway (0.00976%)

6th best scoring pathway (0.00692%)

**Extended Data Fig. 8 | The variations of the integrative models for postmitotic assembly pathway.** Detailed views of the best and lower-scoring pathways. The side views of the 3 spokes are shown. The NE surface is indicated by wireframe model (grey). The uncertainty of each Nup localization is indicated by the density of the corresponding color: The Y-complex (green), isolated fraction of Nup155 (orange), Nup93-Nup188-Nup155 and Nup205-Nup93-Nup155 complexes (blue), and Nup62-Nup58-Nup54 complex (purple). The posterior model likelihood for each pathway is indicated (see Methods for details). The variations in the structural models for the intermediates at 5 and 6 min after anaphase onset would be due to their lower protein density (lower signal-to-noise ratio) in the electron tomography images.

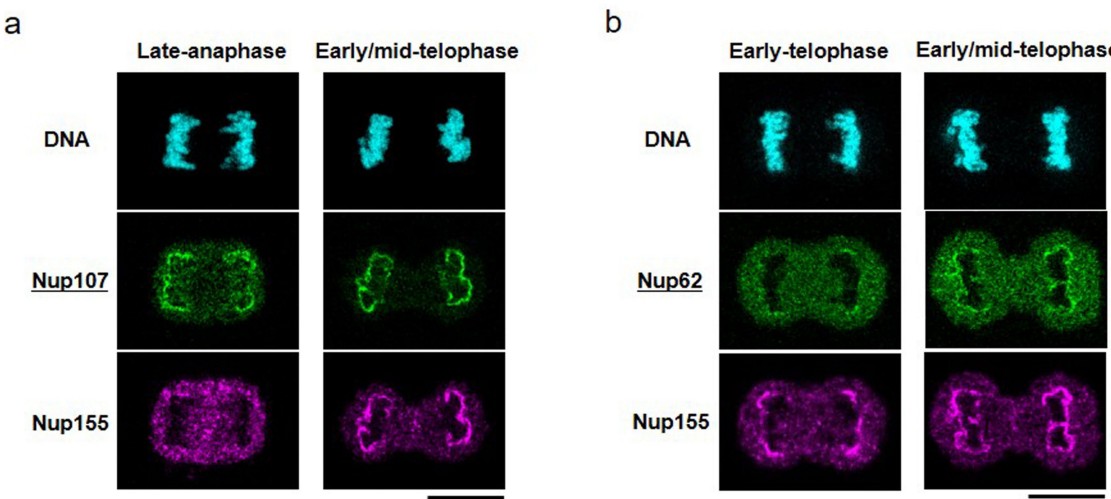

**Extended Data Fig. 9 | Nup155 assembles at a similar timing to Nup62 in postmitotic assembly pathway. a**,**b**, Immunofluorescence with an anti-Nup155 antibody and a GFP-Nanobody. Cells at anaphase and telophase were imaged by confocal microscopy. mEGFP-Nup107 (**a**) and Nup62-mEGFP (**b**) genome-edited cells were used. The cell nuclei were stained with SiR-DNA. Scale bars, 10 μm.

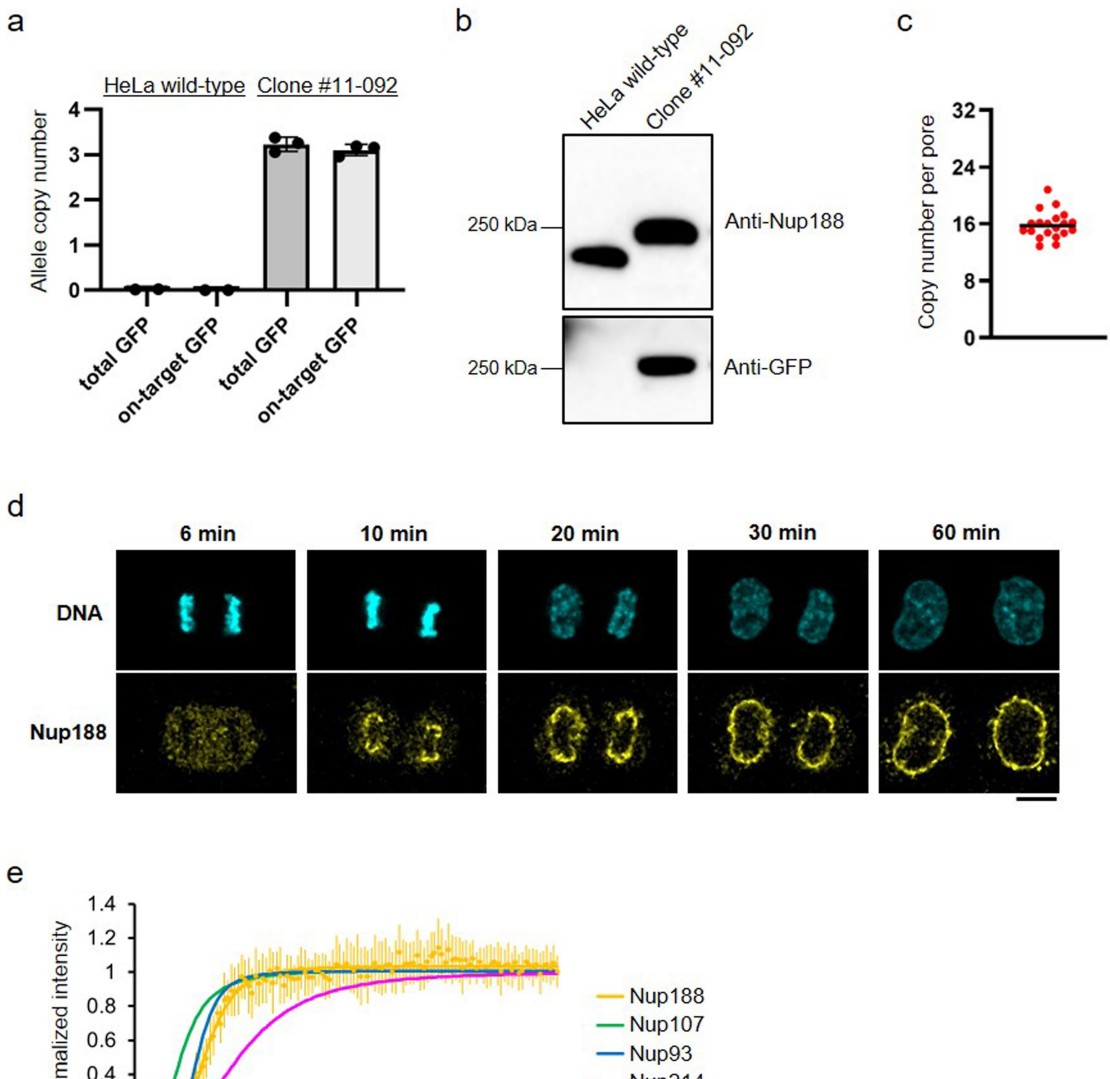

**Extended Data Fig. 10 | Postmitotic assembly kinetics of Nup188 is similar to other inner ring components. a**, Digital PCR analysis of Nup188-mEGFP genome-edited cells for assessing the homozygous tagging. Allele copy numbers of GFP integrations and on-target GFP integrations are quantified. The data are from 2 and 3 measurements for wild-type and Nup188-mEGFP genome-edited HeLa cells. Error bars represent the s.d. of the mean. **b**, Immunoblot analysis of Nup188-mEGFP cells using antibodies against Nup188 and γ-Tubulin. Uncropped immunoblot data are shown in Supplementary Fig. 1. **c**, Calculated copy number of Nup188 per nuclear pore as in Fig. 1. The plot is from 20 cells. The median is depicted as a line. **d**, Time-lapse 3D imaging of Nup188-mEGFP genome-edited cell. The cell nuclei were stained with silicon–rhodamine (SiR)-DNA. Single confocal sections of SiR and GFP channels are shown. Images were filtered with a median filter (kernel size: 0.25 × 0.25 µm). Scale bar, 10 µm. Time after AO is indicated. **e**, Postmitotic assembly kinetics of Nup188 was decomposed as in Extended Data Fig. 5. Dots represent the average and s.d. of measurements from 16 cells. The kinetics of Nup107, Nup93 and Nup214 are also shown for comparison.

# Reporting Summary

## Statistics

For all statistical analyses, confirm that the following items are present in the figure legend, table legend, main text, or Methods section.

| n/a | Confirmed | |
|---|---|---|
| ☐ | ☒ | The exact sample size (*n*) for each experimental group/condition, given as a discrete number and unit of measurement |
| ☐ | ☒ | A statement on whether measurements were taken from distinct samples or whether the same sample was measured repeatedly |
| ☐ | ☒ | The statistical test(s) used AND whether they are one- or two-sided *Only common tests should be described solely by name; describe more complex techniques in the Methods section.* |
| ☒ | ☐ | A description of all covariates tested |
| ☐ | ☒ | A description of any assumptions or corrections, such as tests of normality and adjustment for multiple comparisons |
| ☐ | ☒ | A full description of the statistical parameters including central tendency (e.g. means) or other basic estimates (e.g. regression coefficient) AND variation (e.g. standard deviation) or associated estimates of uncertainty (e.g. confidence intervals) |
| ☐ | ☒ | For null hypothesis testing, the test statistic (e.g. *F*, *t*, *r*) with confidence intervals, effect sizes, degrees of freedom and *P* value noted *Give P values as exact values whenever suitable.* |
| ☐ | ☒ | For Bayesian analysis, information on the choice of priors and Markov chain Monte Carlo settings |
| ☒ | ☐ | For hierarchical and complex designs, identification of the appropriate level for tests and full reporting of outcomes |
| ☐ | ☒ | Estimates of effect sizes (e.g. Cohen's *d*, Pearson's *r*), indicating how they were calculated |

*Our web collection on statistics for biologists contains articles on many of the points above.*

## Software and code

Policy information about availability of computer code

| Data collection | ZEN software 2012 SP1 (8,1,0,484) |
|---|---|
| Data analysis | ImageJ 1.53q, Amira 5.4.5, Fluctuation Analyzer 4G 150223 (https://www-ellenberg.embl.de/resources/data-analysis), FCSFitM v0.8 (https://git.embl.de/grp-ellenberg/FCSAnalyze), FCSCalibration v0.4.2 (https://git.embl.de/grp-ellenberg/FCSAnalyze), RStudio 1.1.383, R 3.4.1, MATLAB 2017a, Python v3.6.8, Integrative Modeling Platform (IMP) v.2.13.0, gmconvert v3.0, custom code in Github repository (https://github.com/integrativemodeling/npcassembly), ChimeraX v1.1.1. |

For manuscripts utilizing custom algorithms or software that are central to the research but not yet described in published literature, software must be made available to editors and reviewers. We strongly encourage code deposition in a community repository (e.g. GitHub). See the Nature Portfolio guidelines for submitting code & software for further information.

## Data

Policy information about availability of data

All manuscripts must include a data availability statement. This statement should provide the following information, where applicable:

- Accession codes, unique identifiers, or web links for publicly available datasets
- A description of any restrictions on data availability
- For clinical datasets or third party data, please ensure that the statement adheres to our policy

Fluorescence images were deposited to the Image Data Resource (IDR; https://idr.openmicroscopy.org/) under accession number idr0115. Our integrative

## Human research participants

Policy information about studies involving human research participants and Sex and Gender in Research.

| | |
|---|---|
| Reporting on sex and gender | *Use the terms sex (biological attribute) and gender (shaped by social and cultural circumstances) carefully in order to avoid confusing both terms. Indicate if findings apply to only one sex or gender; describe whether sex and gender were considered in study design whether sex and/or gender was determined based on self-reporting or assigned and methods used. Provide in the source data disaggregated sex and gender data where this information has been collected, and consent has been obtained for sharing of individual-level data; provide overall numbers in this Reporting Summary. Please state if this information has not been collected. Report sex- and gender-based analyses where performed, justify reasons for lack of sex- and gender-based analysis.* |
| Population characteristics | *Describe the covariate-relevant population characteristics of the human research participants (e.g. age, genotypic information, past and current diagnosis and treatment categories). If you filled out the behavioural & social sciences study design questions and have nothing to add here, write "See above."* |
| Recruitment | *Describe how participants were recruited. Outline any potential self-selection bias or other biases that may be present and how these are likely to impact results.* |
| Ethics oversight | *Identify the organization(s) that approved the study protocol.* |

Note that full information on the approval of the study protocol must also be provided in the manuscript.

# Field-specific reporting

Please select the one below that is the best fit for your research. If you are not sure, read the appropriate sections before making your selection.

☒ Life sciences  ☐ Behavioural & social sciences  ☐ Ecological, evolutionary & environmental sciences

For a reference copy of the document with all sections, see nature.com/documents/nr-reporting-summary-flat.pdf

# Life sciences study design

All studies must disclose on these points even when the disclosure is negative.

| | |
|---|---|
| Sample size | Sample sizes were based on pilot experiments to determine the number of cells required to observe stable population averages with high Pearson's correlation between replicates. Sample sizes are indicated in figure legends. |
| Data exclusions | Videos of dividing cells with rotating nuclei are removed from the analysis, because we cannot properly assign the non-core and core regions. |
| Replication | For quantitative imaging in Fig. 1a, d, the data were from 4, 4, 4, 2, 3, 2, 3, and 2 independent experiments for Nup107, Seh1, Nup205, Nup93, Nup62, Nup214, Tpr and Nup358, respectively. STED imaging in Fig. 4a and Extended Fig. 4a and live imaging in Extended Fig. 10 were from 2 independent experiments. For dynamic quantitative imaging in Fig. 3, the data were from 4, 4, 4, 2, 3, 4, 3, 2, 2, and 4 independent experiments for Nup107, Seh1, Nup205, Nup93, Nup62, Nup214, Tpr, Nup358, Nup153, and Pom121, respectively. STED imaging in Fig. 1b, c and Fig. 4b, southern blotting in Extended Data Fig. 1 and immuno-fluorescence microscopy in Extended Data Fig. 9 are from single experiments. |
| Randomization | No randomization was done, because this study does not involve animals or human participants. Samples were organized into groups based on cell lines. Cells were imaged in randomly chosen fields of view per experiment. All imaged cells were further analyzed. Appropriate controls were included in all experiments. |
| Blinding | No predefined group allocation was performed. All data that passed QC (described in data exclusion section above) were analyzed. The analysis was performed in a reproducible and automated fashion for all experiments as described in Methods. |

# Reporting for specific materials, systems and methods

We require information from authors about some types of materials, experimental systems and methods used in many studies. Here, indicate whether each material, system or method listed is relevant to your study. If you are not sure if a list item applies to your research, read the appropriate section before selecting a response.

## Materials & experimental systems

| n/a | Involved in the study |
|---|---|
| ☐ | ☒ Antibodies |
| ☐ | ☒ Eukaryotic cell lines |
| ☒ | ☐ Palaeontology and archaeology |
| ☒ | ☐ Animals and other organisms |
| ☒ | ☐ Clinical data |
| ☒ | ☐ Dual use research of concern |

## Methods

| n/a | Involved in the study |
|---|---|
| ☒ | ☐ ChIP-seq |
| ☒ | ☐ Flow cytometry |
| ☒ | ☐ MRI-based neuroimaging |

## Antibodies

| | |
|---|---|
| Antibodies used | Mouse anti-Nup62 (Cat. No. 610497; BD Biosciences, Franklin Lakes, NJ), anti-Tpr (Cat. No. HPA019661; The Human Protein Atlas), a GFP-nanobody (FluoTag®-X4 anti-GFP nanobody Abberior® Star 635P; Cat. No. N0304-Ab635P-L; NanoTag Biotechnologies, Göttingen, Germany), rabbit anti-Nup155 (Cat. No. HPA037775; The Human Protein Atlas), rabbit anti-Elys (Cat. No. HPA031658; The Human Protein Atlas), Abberior® STAR RED-conjugated goat anti-mouse IgG (Cat. No. 2-0002-011-2, Abberior GmbH, Göttingen, Germany), Abberior STAR635P goat anti-rabbit IgG (Cat. No. ST635P-1002-500UG, Abberior GmbH), and Alexa Fluor 594 goat anti-rabbit IgG (Cat. No. A-11037; Life Technologies), rabbit anti-Nup188 (Catalog No. A302-322A, Bethyl Laboratories, Montgomery, TX), rabbit anti-Nup205 (Catalog No. ab157090, Abcam, Cambridge, UK), anti-Nup358 (Catalog No. HPA023960, The Human Protein Atlas), rabbit anti-gamma-Tubulin (Cat. No. T5192, Sigma Aldrich), mouse anti-gamma-Tubulin (Cat. No. T5326, Sigma Aldrich), anti-Vinculin (Cat. No. ab219649, Abcam), anti-rabbit IgG horseradish peroxidase (HRP)-conjugated secondary (Cat. No. W4011, Promega), anti-mouse HPR-conjugated secondary (Catalog No. 040-655, Bio-Techne) and anti-rabbit HPR-conjugated secondary (Catalog No. 040-656, Bio-Techne) antibodies. |
| Validation | Antibodies for genome editing are validated by Western Blot for sensitivity and specificity (see Koch et al. Nature Protocols, vol. 13: 1465–1487 (2018), doi: 10.1038/nprot.2018.042). |

## Eukaryotic cell lines

Policy information about cell lines and Sex and Gender in Research

| | |
|---|---|
| Cell line source(s) | Wildtype HeLa Kyoto cells (RRID: CVCL_1922) were a kind gift from Prof. Narumiya, Kyoto University. All the cell lines used in this study are derivatives of this parental HeLa cell line. |
| Authentication | HeLa Kyoto cells were authenticated by sequencing. |
| Mycoplasma contamination | Cells were screened for mycoplasma contamination by PCR before use. It was always negative. |
| Commonly misidentified lines (See ICLAC register) | Cell line is not listed in the list of commonly misidentified cell lines. |

