## [Peer Review File · Nature]

Editorial Note: Figure 7 on page 41 in this Peer Review File is reproduced with permission from Nature Protocols

Manuscript Title: A quantitative map of nuclear pore assembly reveals two distinct mechanisms **Reviewer Comments & Author Rebuttals**

Reviewer Reports on the Initial Version:

Referees' comments:

Referee #1 (Remarks to the Author):

The assembly of nuclear pore complexes (NPCs) was studied using fluorescence microscopy and molecular modeling. In HeLe cells, ten nucleoporins (Nups) within the major subcomplexes of the NPC were tagged with fluorescent proteins. To quantify their arrival at NPCs, confocal microscopy was used after intensity calibration using FCS. Postmitotic and interphase NPC assembly could be kinetically separated. This separation was aided by a spatial division into non-core and inner-core regions where one or the other process dominates. The two pathways differed both in kinetics and in the order of events. A main finding is that in the postmitotic assembly the central ring is formed early, whereas in interphase assembly it is formed late. Integrative models were developed to provide a structural view of postmitotic assembly intermediates, using recently published electron tomographic reconstructions as additional input.

NPC assembly has great biological relevance and is -- in light of the complex structure -- of broad interest. The detailed measurements, the integrative modeling, and the insights gained do make a significant contribution. However, I have a number of concerns and issues, as listed below, grouped by topic and relevance. Most importantly, I ask for a stronger case concerning novelty and impact.

Novelty and impact.

1) The manuscript does not make clear what is genuinely new and what is known. For instance, differences between interphase and postmitotic pathways are well established (Doucet et al. Cell 2010) and have been reviewed in-depth (see eg Hampoelz et al. Annu. Rev. Biophys. 2019). The time-course of arrival of Nups has been studied extensively using confocal microscopy, including by the authors. Recent findings by the Weis group on the time-course of Nup interactions in NPC assembly (Onischenko et al., Cell 2020) are mentioned only very briefly and not discussed in detail. The authors should strengthen their case for novelty and relevance by distinguishing more clearly what they consider genuinely new findings from what is known in the field.

Structural modeling.

2) The modeling of intermediates adds a helpful and important structural view of the assembly process. However, validation of the 3D structural models is lacking. What do these models predict and how can we test them critically against independent information?

3) In the integrative modeling, the fluorescently labeled Nups were used as markers for the arrival of the subcomplexes they are part of. In case of the Y complex, there is substantial evidence for such a preassembly (here: Nup107 and Seh1 arrive together in both pathways). How about the other subcomplexes? Is it justified to assume that their parts arrive concertedly? Is there experimental evidence for the arrival of preformed and intact subcomplexes?

4) Are there large variations in the integrative models of assembly intermediates?

5) At the time points modeled in Fig. 4, we appear to have a very high degree of symmetry. All component stoichiometries are multiples of eight. Is this realistic? Shouldn't there be variations in the copy numbers? It would be helpful if possible variations in the structural models were somehow communicated, together with an assessment of how much variability one should expect.

6) The density maps also appear to be symmetric. Again, is this realistic for an individual NPC as compared to the average tomographic reconstruction? Based on the faded-out densities of the maps (Fig. 4a and c) I would assume that individual NPCs at the respective time points lack symmetry and exhibit copy number variations.

7) A related issue concerns model scoring. The ET densities are averages of superimposed and likely incomplete assembly intermediates. Is cross-correlation scoring the best way to deal with the resulting fractional densities of elevated symmetry? Was unoccupied density simply not scored?

8) The components in the "5min" model shown in Fig. 4c are not all contiguous. Is this an artifact? Does Nup155 (orange) indeed assemble all by itself in the periphery? In the assembled NPC, it borders Nup205 (cyan).

9) In the integrative modeling, native structural contacts were increasingly enforced as the assembly process proceeded. Is this justified?

10) For an informed assessment of the models, it will be important to show at the very least zoom-ins on their structural elements. At least the top-scoring model at each time point should be made accessible.

11) Integrative structural modeling was performed only for the postmitotic assembly pathway. Were attempts made to model also interphase assembly? What would What, if anything, can we learn about intermediates of interphase assembly?

Kinetic modeling.

12) The kinetic modeling builds on Ref. 38. For the signaling cascade studied in Ref. 38, the duration of the signal makes sense (as the time over which a "signal" is typically present). Here it is not clear what is meant by duration as compared to the mean time of the assembly. Mathematically, for the assumed irreversible process at each step, τ_i is the mean time to complete assembly step i (equation 10). The so-called "duration" θ_i is the standard deviation of the time to complete assembly step i . Is this distinction relevant here? Furthermore, it is not clear to me how the "duration" (as compared to the mean time) was extracted from the sigmoidal curves.

13) In the kinetic modeling, I wonder if the delay "d" was subtracted from the times. As written, equations 3 and 4 do not quite make sense. Why should they apply only after time d (as this would result in a "jump" at time $t=d$)?

General points.

14) How were the Nups selected for labeling? Why these ten and not others?

15) Can we be sure that the slow inner-core assembly is representative of the interphase pathway?

16) Fusion of the nuclear envelope to create an initial pore is a critical event in interphase NPC assembly. Can the data be used to time this event?

17) I suspect that the densities in Fig. 4a are from Ref. 12. Is this correct?

18) In the time courses of copy numbers (Fig. 2b), there appears to be an early Nup transfer (0-5min) from nucleoplasm to cytoplasm. How should we interpret the drop of the nucleoplasm counts to zero at ~5 min after AO?

19) What do the boxes indicate in Fig. 3b?

Referee #2 (Remarks to the Author):

In the manuscript entitled "A quantitative map of nuclear pore assembly reveals two distinct mechanisms", Otsuka et al use FCS to derive assembly rate constants for representative components of the NPC. They find, as was previously proposed, that two pathways exist; a rapid one following mitotic exit and a slow pathway that is more generalized during interphase. This is a well presented and well rounded study integrating live cell imaging with biochemical assembly kinetics and represents, in my opinion, how microscopy data should be utilized to determine molecular mechanism. I have a few minor points and a broad interpretation question listed below. Minor points:

1) There are a few (very few) typos that should be addressed.

2) I assume the live cell imaging parameters were the same as those listed for the FCS confocal imaging methods. This should be made clear in the methods particularly given the importance of imaging to the conclusions drawn.

Broad Question:

In looking at the data, I feel that a second interpretation could be that the pathways emerge from two different events: assembly and re-assembly. The kinetics presented support the idea that there is in-fact a single pathway for assembly that occurs at some rate throughout interphase and is halted in mitosis. Upon mitotic exit the assembly resumes at the "slow" rate. Convolved within this kinetic data is the re-assembly pathway, which consists of fully assembled NPCs breaking down in mitosis and then rapidly reforming upon mitotic exit. In this light, a kinetically slow event in assembly is rapid in reassembly possibly due to some molecular change that occurs upon initial assembly.

Is there a reason that this can not be the case? I would expect to see a slightly more diverse discussion of possible pathways given that the conclusion drawn here are in overall agreement with previous assumptions of NPC assembly. In short, I believe this point should be addressed prior to publication.

Referee #3 (Remarks to the Author):

Summary of the key results

In cells with open mitosis, NPCs use two distinct assembly modes. At the end of mitosis, NPCs reassemble into the reforming nuclear envelope in a fast and highly synchronous manner, which occurs mainly in the so-called non-core region of the reforming nuclear envelope. Later, NPCs integrate into the closed nuclear envelope in a much slower interphase assembly mode, in early G1 mostly in the core region of the nuclear envelope. In this manuscript, Otsuka et al use FCS-calibrated live-cell imaging in HeLa cells to determine the quantitative nuclear envelope recruitment kinetics of key nucleoporins, the proteins forming NPCs, to generate models of postmitotic and interphase NPC assembly pathways.

The main findings are that core structural components assemble in the same order in postmitotic

and interphase assembly: The Y-complex, which is the major building block of the nuclear and cytoplasmic ring structures, precedes inner ring components (Nup93 and Nup205) and the central transport channel (Nup62). Nup358 is integrated late. Surprisingly, the nuclear basket component TPR and the cytoplasmic localized Nup214 are recruited earlier in the interphase assembly pathway as compared to the postmitotic mode. This very extensive and high-quality set of data will be very instrumental for our understanding of NPC structure and function and, as other publications from the Ellenberg lab, a reference point for the field.

Originality and significance: if not novel, please include reference

The postmitotic assembly order is consistent with previous studies including publications from the Ellenberg lab, cited in the MS). A few aspects of the interphase assembly orders have been known (Pom121 before the Y-complex, Nup358 late, also properly cited work by the Ellenberg lab) but this comprehensive data set including all different NPC structural components will be very instrumental for the field. The proposed early recruitment of TPR (as compared to postmitotic assembly) is unexpected. It remains, however, open whether this is crucial for interphase assembly.

Data & methodology: validity of the approach, quality of data, quality of presentation

The study uses an innovative and well-concealed research approach. The findings and models in this work are based on high-quality data obtained with FCS-calibrated live-cell imaging and super-resolution microscopy, which are state-of-art methodologies fully adequate to the quantitative purpose of this research. They are reasonably well described and referenced in the materials and methods section. Homozygously mEGFP/mCherry tagging of nucleoporins at their endogenous gene locus is used. The modeling assumptions seem reasonable and falsifiable.

Appropriate use of statistics and treatment of uncertainties

Sample size determination and statistical analysis are well described, although no Pearson's correlation or p-values are found in the manuscript.

Conclusions: robustness, validity, reliability, Suggested improvements: experiments, data for possible revision

- In lines 112-115 and Figure 2B, the range of each nucleoporin molecule per cell is indicated. These values are key results and are necessary for the later quantitation and models. For these calculations, a precise account of single-cell cytoplasmic and nuclear volumes is mandatory. While the volume from the nuclei has been calculated by the DNA counterstaining in each cell from this study, a single average value for the cytoplasm volume from another publication of the authors has been used for the calculations. For more accuracy and to avoid that the values given in this work are inexact due to the cell-to-cell variability (clearly visible in the figure 2A) or to experimental factors that might affect the cell volume, I suggest validating the values of cytoplasmic volume in, at least, a subset of the presented data. Performing the experiments measuring the cell volume as the authors describe in Cai, Y. et al 2018 would be an option. An alternative proxy to validate the results would be to determine the single cell volume using the volumetric segmentation of the eGFP images in those cells in metaphase and early mitotic exit (i.e. the cytoplasmic signal of the Nups).

- line 179-182: the authors argue that the poorer correlation between duration and time point of assembly of nucleoporins in the interphase assembly mode suggests a less strictly sequential pathway. Could the differences in correlation be due to a highly synchronous initiation of postmitotic NPC assembly in telophase whereas for interphase mode assembly of NPCs is initiated during a much longer period, in principle the entire G1 phase/interphase?

- line 242-244: the authors argue that the early recruitment of TPR in interphase assembly key role of the protein in the assembly mode. Experimental evidence would significantly strengthen the manuscript. Of note, early recruitment, e.g. of Nup153 in postmitotic NPC assembly does not necessarily correlate with the requirement of this Nup for postmitotic NPC assembly.

References: appropriate credit to previous work?
Previous work is properly cited.

Clarity and context: lucidity of abstract/summary, appropriateness of abstract, introduction, and conclusions

The manuscript is clearly written and easy to follow. The abstract summarises the important findings.

Additional points:

- Clarify the terminology disagreement between text (line 98) and Extended Data Fig. 1. Nup153 and Pom121 are excluded from quantitation in figures 1 and 2 because, as is indicated in the text, they display lower concentrations likely due to "subhomozygous" tagging. However, in supplementary figure 1, Nup153 is clearly heterozygous, as indicated. The Pom121 clone used in this work is seemingly homozygous. If by "subhomozygous" it is referred the low MW band in the Southern blot in the clone 32 from Pom121, a clarification about the meaning of this term would be recommended (it does not exist as a concept as far as I know and could check).

- For DNA counterstaining, SiR-DNA is referred in most of the text but in line 400 and in Extended Data Fig. 2 the alternative term (SiR) Hoechst is used. Consistent terminology would be recommended.

- In Figure 2B, plots displaying the range and distribution of the values in the temporal series would be recommended to better communicate the variability of the data set.

Line 207-209: The statement that the nuclear ring first assembly mode (of postmitotic NPC assembly) is consistent with the eight-fold symmetric protein density uses ref 11, which is a study on interphase NPC assembly. It would be appropriate to mention this explicitly, e.g. as "... on the inner nuclear membrane at early stages of interphase NPC assembly"

- line 495: Lap2alpha is used as a core marker. Given the reported nuclear localization of this protein in contrast to NE localization to other Lap2 isoforms this is unexpected. Is this a typing error?

Referee #4 (Remarks to the Author):

A- Summary of the key results

In this manuscript, the authors quantitatively analyzed the timing of recruitment to the NE of nine GFP- and one mCherry-tagged Nups (generated by genome editing in HeLa cells) in the two hours following mitotic exit. Based on this extensive analysis, and using the fact that, as previously described, NPCs first assemble after mitosis in the "non core" region of the reforming NE through the post-mitotic pathway, and then following the "interphasic" pathway in the "core" region of the NE, they confirm previous data regarding the fact that post-mitotic and interphase NPC assembly proceed by strikingly different molecular mechanisms. Novel to this study, their data indicate that unlike in post-mitotic assembly, the nuclear basket protein TPR and the cytoplasmic filament protein Nup214 are recruited before the central ring complex during the interphase assembly process.

Finally, they used computational modeling to combine these data with their previous single prepores subtomogram averaging (Otsuko et al., 2018), to propose an integrative modeling of the post-mitotic assembly pathway of the main NPC scaffold, with a refined order of recruitment of its structural modules, revealing stepwise assembly of the 4 copies of Y-complexes, nested with the assembly of the inner ring.

B. Originality and significance:

While this study is well-performed and will be of interest to specialized audience eager to understand the exquisite details of NPC assembly pathways, it may not provide enough novelty, either conceptually, mechanistically, or technically, to engage a broader audience.

Indeed, as stated by the authors, it was previously demonstrated that the "postmitotic and interphase NPC assembly possess distinct kinetic, molecular and structural features". For instance, the inverse order of recruitment of the Y-shaped complex and Pom121- (Dultz, E. & Ellenberg, J. Cell Biol 2010). Therefore, the title of the manuscript is somewhat misleading.

As a consequence, with the exception of hypotheses regarding evolution, largely based on their observation of an early Tpr recruitment in interphase as compared to its reported late recruitment in budding yeast, no new concept arises in the discussion.

On line 247, that authors state : « The order of Nup assembly for interphase assembly we observed here is consistent with the recently reported order of NPC assembly in budding yeast that undergoes a closed mitosis, except for the Tpr homologues Mlp1/2 which assemble late in yeast ». Yet, one of their key novel hypothesis, namely « the unexpectedly early presence of Tpr suggests a key role of this large coiled-coil protein during interphase assembly » is not tested by additional experiments.

Moreover, the statement that the order they observed is "consistent" with the one reported by Onischenko et al. (Cell 2020) does not seem to reflect the data presented in that publication. Indeed, in their recent study, Onischenko et al. conclude "Taken together, NPC maturation begins with the symmetric NUPs, continues with the asymmetric ones, and concludes with the Mlp proteins". Hence, in budding yeast, there is to my knowledge no evidence that the central ring assembles after Y-complex. In addition, Onischenko et al. describe an unexpected asynchronous assembly of the stem and the head NUPs of the Y complex in budding yeast, but this is not observed by Otsuka et al in this ms.

Here, a possible reason/caveat might be the choice of Otsuka et al to follow, for technical reasons, the interphase assembly of NPCs in telo/G1 cells and not their assembly between G1 and G2, when NPC number slowly doubles. From their quantitative study, the authors conclude that « NPC assembly relies on and consumes almost half of the material inherited from the mother cell within one hour after mitosis ». Could this assembly path be different at later stages of the cell cycle, when assembly would possibly rely on newly-synthesized components ? Could that explain the observation that in their study, « cytoplasmic Y-rings, including the base of the cytoplasmic filaments with Nup214, are already "prebuilt" within the inner membrane evagination », which is not observed in budding yeast ?

Might there be in mammalian cells, three pathways of NPC assembly, post-mitotic, Telo/G1 and S/G2 ?

C- Data & methodology: validity of approach, quality of data, quality of presentation

The approach is appropriate, yet, as discussed above, it only allows to follow an early and possibly specialized phase of "interphase" NPC assembly.

The "FCS qualibrated live Imaging" highlighted in their abstract (carried out as described in a previous report), is based on calibration in interphase cells using GFP-Nup107, while other values were based on the comparison of other cell lines with the Nup107 cell line, acquired the same day in a distinct well.

Hence, this approach manly serves here to validate that the tagging does not affect the incorporation of Nups in NPC. Indeed, the same conclusion could be reached from the previously established copy number of these Nups in mature NPC (the fact that there are 32 copies of Nup107 per mature NPC, would have been sufficient to calibrate the confocal images of the other GFP-Nups).

How Pom121-mCherry copy number was quantified in unclear.

D- Use of statistics and treatment of uncertainties seems appropriate

E- Conclusions: robustness, validity, reliability

I do not feel qualified to determine how robust their integrative model is. Yet, as Nup155 was not imaged in this study, it seems that its positioning in the model mainly arises from the previous EM data. May that affect the overall interpretation?

Another possible limit is that while individual NPCs are imaged in interphase cells thanks to STED microscopy, the quantitative data are based on confocal microscopy. Hence, possible recruitment of several Nups to the NE rather than to assembling NPCs cannot be excluded.

F- Suggested improvements: experiments, data for possible revision

To strengthen this study, the authors may consider to validate their hypothesis regarding the role of TPR in NPC assembly by performing functional studies, for instance by combining this quantitative analysis with acute degron-induced depletion of TPR in mitosis (see study from the Dasso lab, Nat Commun. 2020).

Imaging of the core region of the Telo/G1 cells by STED, combining the GFP lines with antibodies (or using a double-labelled GFP/mCherry line) may help to strengthen their conclusion regarding TPR or inverted assembly pathways.

- The postmitotic assembly model would deserve validation by quantifying Nup155 (and Nup188) recruitment (which would require to generate the corresponding GFP-cell lines). Indeed, Nup155 appears in their model in two distinct complexes as well as an isolated component.
- Nup62 does not solely belong to the central channel, but also interacts with Nup214; how is this taken in account in this model?
- The postmitotic assembly model does not include the peripheral components. Is the timing of Nup358 recruitment compatible with its implication in the assembly of the second Y outer ring on the cytoplasmic side as previously proposed (von Appen, Nature 2015) ?

Minor comments:

The authors should clarify figure 4.

- the origin of the data for panel (a) should be stated in the figure legend by referring (I assume) to Otsuka et al., 2016).
(the protein density could be colored differently to be better distinguished from the NE surface model).
- What are the protein densities at 5 min and 6 min on the nuclear side of the NPC that subsequently disappear at 8 min (panels a)? Why don't they appear on panel c?
- The similar colors in b and c refers to Nucleoporins in (b) and nucleoporin complexes in (c). While this is fine for Nup107, that only belongs to the Y-shape complex, it leads to confusion as Nup93, colored in blue in (b), appears in the blue and cyan complexes in c.
Nup155 is considered to be part of 2 different complexes and also appears as isolated protein (orange)....

Line 184: Integrative modeling of the NPC assembly pathway

Title of this chapter is misleading as only the postmitotic assembly pathway or the core NPC is modeled.

G- References: appropriate credit to previous work

General comment: Wherever this study will be published, the authors should be careful to properly acknowledge other teams who previously made the critical discoveries and to clarify what was already known, even from their previous studies (previously published lines, methodologies,...)

Although previous studies are in general cited, the way they are cited in the text is not appropriate,

as it gives most of the credit to the previous contribution of the authors.

- As example, line "As we and others have demonstrated previously (12,24),...."

This does not give proper credit to Maeshima et al. (ref 24) who initially demonstrated the use of pore-free islands in the inner core region, that was subsequently used by Otsuko et al in 2016 to generate the two-component model (ref 12) that is reused in this study.

- likewise, the title of the manuscript "A quantitative map of nuclear pore assembly reveals two distinct mechanisms" is misleading since the existence of two distinct mechanisms was previously established by multiple labs .

Similarly, it is not appropriate to write (line 214) "Our data revealed that the two assembly pathways employ strikingly different molecular mechanisms", while they mainly confirmed that concept.

As discussed above, the recent study regarding NPC assembly in budding yeast (Onischenko et al. Cell 2020) is only discussed in respect to timing of TPR recruitment. Yet, other striking differences are neither mentioned nor discussed.

The observation from the Hetzer lab that Tpr depletion leads to an increase in the total number of NPC per cell nucleus (Genes and Dev, 2018) seems to be inconsistent with their hypothesized requirement of TPR in interphase assembly; yet, this paper is not cited.

Conversely, the authors do not always clearly state that they had already published/described some of the tools or experimental approaches used in this study.

- For instance, when reading "To quantitatively analyze.....we genome-edited HeLa cells, homozygously tagging the endogenous genes for ten different Nups with mEGFP or mCherry", as no references are included, except one reference to a method paper (in which GFP-Nup107 was generated along many other lines), one expects that these cell lines were new tools generated for this study. Yet, reading the method it appears that only 4 out of the 10 cell lines presented were generated in this study, of which two are heterozygous. The other ones were previously published by the Ellenberg's team.

Proper phrasing would be: 'We used 10 genome-edited HeLa cell lines of which 6 were previously published.

- likewise, in line 127, the authors should clearly state that they had previously used the two-component model of a fast (postmitotic) and a slow (interphase) assembly process to fit the experimental data regarding Nup107 and Nup358 assembly (Otsuko et al, Elife 2016, Fig 4).

Author Rebuttals to Initial Comments:

Point-by-point response (referee comments in *blue italics*, our response in black)

Referee #1 (Remarks to the Author):

The assembly of nuclear pore complexes (NPCs) was studied using fluorescence microscopy and molecular modeling. In HeLe cells, ten nucleoporins (Nups) within the major subcomplexes of the NPC were tagged with fluorescent proteins. To quantify their arrival at NPCs, confocal microscopy was used after intensity calibration using FCS. Postmitotic and interphase NPC assembly could be kinetically separated. This separation was aided by a spatial division into non-core and inner-core regions where one or the other process dominates. The two pathways differed both in kinetics and in the order of events. A main finding is that in the postmitotic assembly the central ring is formed early, whereas in interphase assembly it is formed late. Integrative models were developed to provide a structural view of postmitotic assembly intermediates, using recently published electron tomographic reconstructions as additional input.

NPC assembly has great biological relevance and is -- in light of the complex structure -- of broad interest. The detailed measurements, the integrative modeling, and the insights gained do make a significant contribution. However, I have a number of concerns and issues, as listed below, grouped by topic and relevance. Most importantly, I ask for a stronger case concerning novelty and impact.

We thank the reviewer for the constructive suggestions on how to improve our manuscript. The specific recommendations for further validation and extension of our study are very helpful and have led us to perform numerous experiments that corroborated the key findings of our manuscript, as explained below.

Novelty and impact.

1) The manuscript does not make clear what is genuinely new and what is known. For instance, differences between interphase and postmitotic pathways are well established (Doucet et al. Cell 2010) and have been reviewed in-depth (see eg Hampoelz et al. Annu. Rev. Biophys. 2019). The time-course of arrival of Nups has been studied extensively using confocal microscopy, including by the authors. Recent findings by the Weis group on the time-course of Nup interactions in NPC assembly (Onischenko et al., Cell 2020) are mentioned only very briefly and not discussed in detail. The authors should strengthen their case for novelty and relevance by distinguishing more clearly what they consider genuinely new findings from what is known in the field.

We thank the reviewer for pointing this out and apologize for our very short description of novelty due to the space constraints. To clarify the novelty and relevance of our work, we have rewritten large parts of the discussion (page 9-12). We have also performed additional experiments to strengthen the findings (New Figure 4, Extended Data Figures 2, 7 and 8).

For the convenience of the reviewer, we summarize here the main novel findings and the general impact of our study:

1. Mechanistic insights into postmitotic NPC assembly from integrative structural modeling

It is correct that the relative time-course of several Nups during postmitotic assembly had been studied previously by live cell imaging. However, how several hundred of Nups self-organize to form the NPC channel has remained largely enigmatic. Structural and mechanistic insights were difficult to

obtain from previous studies because they could only provide relative kinetic data as they relied on ectopic expression of fluorescently-tagged Nups.

By contrast, in this study, we have performed FCS calibrated live imaging of endogenously fluorescently-tagged Nups and by combining it with super-resolution data on pore density we could for the first time determine the changes in absolute Nup stoichiometry over time for postmitotic as well as interphase assembly. The dynamic stoichiometry data enabled us to build the first structural model of postmitotic NPC assembly intermediates. Our structural model allows to make multiple very interesting mechanistic predictions, such as for example that the central ring complex might be needed early in postmitotic assembly to prevent sealing and promote expansion of ER holes after mitosis, and the hydrophobic FG-repeats of central channel Nups might play a role in the initial dilation of the small membrane hole into the larger, NPC sized channel. These hypotheses illustrate the value of our quantitative dataset and the structural assembly model that we were able to derive from it.

In the revised manuscript, we have now validated our integrative model predictions by additionally investigating the assembly kinetics of two previously not studied nucleoporins, Nup155 and Nup188 (Extended Data Figures 7 and 8, the details are explained below in the section of “*Structural modeling 2*”). We believe that our data adds the much-needed information about molecular mechanism and difference between the two pathways of NPC assembly, that were not provided by the previous relative time-course analysis of Nup assembly.

2. Assembly order and stoichiometry for interphase assembly

So far it has been shown that the molecular requirements between postmitotic and interphase assembly pathways are different (Franz CR et al, *EMBO Rep*, 2007; Doucet et al, *Cell*, 2010; Talamas JA et al, *JCB*, 2011; Funakoshi et al, *MoBC*, 2011; Vollmer B, *Dev Cell*, 2015; Rampello et al, *JCB*, 2020). However, very little has been known about the Nup assembly order and kinetics for interphase pathway due to the rare and sporadic nature of the interphase assembly. While some kinetic data have been available for a small subset of Nups for interphase assembly pathway (Pom121, Nup107 and Nup93; D'Angelo et al., *Science*, 2006. DOI: 10.1126/science.1124196; Dultz & Ellenberg, *JCB*, 2010. DOI: 10.1083/jcb.201007076), no data have been available for other key Nups composing cytoplasmic filaments, central channel, and nuclear basket, nor for distinct subunits of the Y-shaped and inner-ring complexes. In addition, like for postmitotic assembly, previous studies relied on ectopic expression of fluorescently-tagged Nups and therefore could only provide relative kinetic data, which has limited the ability to obtain insights into the assembly mechanism.

In this study, we therefore conceptually advanced the field of interphase assembly, by revealing the changes in subunit composition and stoichiometry during interphase assembly of ten Nups that represent all the major building blocks of the NPC. Our data for the first time systematically identified the key mechanistic differences between the two assembly pathways. During postmitotic assembly, the Y-complex is rapidly combined with components of the central ring, building the inner core of the pore prior to addition of either cytoplasmic or nuclear filament proteins, that follow much later. By contrast, during interphase assembly, the Y-complex is first combined with the nuclear filament protein Tpr and the base of the cytoplasmic filament Nup214, while the central ring complex is added only afterwards. To our best knowledge, it is for example a novel and unexpected finding that the central ring and nuclear filaments follow an inverted molecular order between the postmitotic and interphase NPC assembly. In the revised manuscript, we have corroborated this very interesting finding further by 3D-STED super-resolution microscopy at the level of single assembling NPCs inside the cell (New Figure 4a, the details are explained in the next section).

Our unprecedented stoichiometry data for interphase assembly thus lays the foundation for understanding molecular mechanism of interphase assembly, a process that involves dramatic and still mysterious membrane topology and conformational changes. Especially, if additional experimental

constraints are obtained from emerging methods such as 3D super-resolution microscopy to image the molecular architecture of the NPC intermediates (Thevathasan et al., *Nat Methods*, 2019. DOI: 10.1038/s41592-019-0574-9; Sabinina et al., *Mol Biol Cell*, 2021. DOI: 10.1091/mbc.E20-11-0728) or dynamic super-resolution microscopy such as MINFLUX to follow some of the conformational changes in real time in living cells (Gwosch et al., *Nat Methods*, 2020. DOI: 10.1038/s41592-019-0688-0), integrative structure modelling will become possible for interphase assembly similar to what we have achieved for the postmitotic pathway in this study. In addition, the distinct assembly order of the NPC for postmitotic and interphase pathways sheds new light on the question of how the nuclear pore may have evolved, which was a key step for subcellular compartmentalisation during the evolution of early eukaryotes.

3. Earlier recruitment of Tpr in interphase assembly and key differences compared to yeast

The recent Onischenko et al. study (Onischenko et al., *Cell*, 2020. DOI: 10.1016/j.cell.2020.11.001) investigated the assembly order of major Nups in budding yeast by using metabolic labelling and mass spectrometry. Our study and the Onischenko et al. study are highly complementary, as they use very different biological models (our study human cells, their study budding yeast) and orthogonal approaches (our study quantitative live imaging/structural modelling, their study metabolic labelling and mass spectrometry). However, we think that our study goes much beyond their findings in at least two ways. First, it maps a major NPC assembly process, the postmitotic pathway, that does not occur in yeast due to its closed mitosis. Second, our work revealed striking differences of the interphase pathway between human and yeast cells. Most prominently, our study shows that, unexpectedly, in human cells both the cytoplasmic nucleoporin Nup214 and the nuclear basket protein Tpr assemble earlier than central channel/inner ring nucleoporins. Our study also revealed a more synchronous assembly of Y-complex components in human cells.

To provide additional evidence for the surprisingly early recruitment of Tpr, we have now immuno-stained NPCs by antibodies against Tpr and the later assembling nucleoporin Nup62 and visualised their co-occurrence in single assembling interphase pores in early G1 by 3D-STED super-resolution microscopy. This single NPC imaging could directly demonstrate that a large number of assembling NPCs in the core region, where interphase assembly dominates, contain Tpr but not Nup62, while most of the assembled NPCs in the non-core region of the same cells, stemming from postmitotic assembly, contain both Tpr and Nup62 (New Figure 4a,c). We have also examined if this surprisingly early recruitment of Tpr also occurs during the more sporadic NPC assembly later in interphase in S-phase/G2 nuclei, and indeed found that also here a significant number of NPCs contain Tpr but not Nup62 in S-phase/G2 nuclei (New Figure 4b,c). Both of these results confirmed the surprisingly early recruitment of Tpr in interphase assembly pathway, and suggest that there is no “third pathway” of later interphase assembly, as the assembly order was consistent between early G1 and G2 for these Nups.

4. Providing a new paradigm for studying dynamic molecular machines inside cells

Beyond our conceptual advances for understanding the mechanism of nuclear pore complex assembly, we expect that the combination of quantitative 4D molecular mapping inside cells and integrative computational structure modeling of dynamic molecular processes we employed here will provide a useful approach for how to study the assembly mechanisms and functional cycles of other large molecular machines inside cells.

Structural modeling.

2) The modeling of intermediates adds a helpful and important structural view of the assembly process. However, validation of the 3D structural models is lacking. What do these models predict and how can we test them critically against independent information?

We thank the reviewer for the thoughtful comment. Our integrative model makes structural and mechanistic predictions. Structurally, our model predicts that all the inner-ring components (Nup205, Nup188, Nup155, and Nup93) are the major constituents of the early assembly intermediates. Since we had examined only Nup205 and Nup93, we decided to experimentally validate the presence of Nup188 and Nup155 in the early assembly intermediates as the model predicted. For Nup188, we have generated a homozygous GFP-Nup188 knock-in cell line by genome editing and examined the assembly kinetics of Nup188 in postmitotic and interphase assembly pathways by live imaging as we did for other Nups (Extended Data Figure 8). These new experiments demonstrated that the assembly kinetics of Nup188 is indeed similar to other inner ring components (Nup93 and Nup205) as the model predicted. For Nup155, it was unfortunately technically very challenging to generate a homozygous GFP-Nup155 knock-in cell line due to the multi-allelic nature of this gene in the HeLa genome. To address this nevertheless, we have imaged endogenous Nup155 in a fixed cell time course in late-anaphase, early- and mid-telophase using an anti-Nup155 antibody (Extended Data Figure 7a, b). We have used mEGFP-Nup107 and Nup62-mEGFP genome-edited cells to compare the relative assembly timing of Nup155. This new immuno-fluorescence time course microscopy has shown that Nup155 assembles at a similar timing to Nup62, which again is consistent with the model prediction. We have added these new validation experiments in the revised manuscript (page 9, line 203-204; Extended Data Figures 7 and 8).

In addition to the structural prediction, our integrative model allows to make multiple very interesting mechanistic predictions. As we described above, it predicts that the central ring complex might be needed to prevent sealing and promote expansion of ER holes after mitosis, and the hydrophobic FG-repeats of central channel Nups might play a role in the initial dilation of the small membrane hole into the larger, NPC sized channel. In order to test these hypotheses, one would need to acutely deplete Nups just before assembly starts and then investigate the ultrastructure of the assembly intermediates by high-spatial resolution microscopy such as correlative electron tomography. This is certainly a very interesting objective of future research projects, however in our opinion beyond the scope of a revision of the present study. Such experiments are technically very demanding because they require selective and cell cycle controlled acute depletion and time-resolved correlative ultrastructural investigation. Engineering the molecular and cellular tools and developing the required imaging methodologies will require years of very interesting work, for which our integrative model for the first time provides the basis to choose the best candidate molecules.

3) In the integrative modeling, the fluorescently labeled Nups were used as markers for the arrival of the subcomplexes they are part of. In case of the Y complex, there is substantial evidence for such a preassembly (here: Nup107 and Seh1 arrive together in both pathways). How about the other subcomplexes? Is it justified to assume that their parts arrive concertedly? Is there experimental evidence for the arrival of preformed and intact subcomplexes?

To address this, we have examined the diffusion coefficient of our GFP-labeled Nups by FCS. It showed that all the measured Nups (Seh1, Nup205, Nup93, Nup62, Nup214, Tpr and Nup358) diffuse not as monomers but as multimers with the apparent diameter of around 12–30 nm in the cytosol of metaphase cells (new Extended Data Table 4). This new data and analysis support the idea that not only the outer-ring complex but the other subcomplexes (at least the inner-ring complex) are preformed in the metaphase cytosol before assembling into the NPC. We have added this point in the revised manuscript (page 35, line 734-739) and provided the new FCS result in the extended data (Extended Data Table 4). In addition, we note recent study from Tom Kirchhausen's lab (Chou et al.,

Dev Cell, 2021. DOI: 10.1016/j.devcel.2021.05.015) which suggested that the outer-ring Nups (Nup133 and Nup107) and the inner-ring Nups (Nup205) remain assembled with 8-fold multiplicity during mitosis.

4) Are there large variations in the integrative models of assembly intermediates?

Our MCMC (Markov chain Monte Carlo) simulations localize in most runs to single stable minima which motivates our use of simple clustering. In the revised manuscript, we have added the detailed views of other trajectory examples (Extended Data Figure 6) to demonstrate the variance in the pathway models. The structural models of the intermediates at 8, 10 and 15 min after anaphase onset are almost identical between the best and lower-scoring pathways, while there are variations in the structural models for the intermediates at 5 and 6 min due to the lower protein density in our electron tomography images. We have clarified this point in the legends of new Figure 5 and Extended Data Figure 6.

5) At the time points modeled in Fig. 4, we appear to have a very high degree of symmetry. All component stoichiometries are multiples of eight. Is this realistic? Shouldn't there be variations in the copy numbers? It would be helpful if possible variations in the structural models were somehow communicated, together with an assessment of how much variability one should expect.

We think it is reasonable to model the assembly pathway assuming that the component stoichiometry of the outer- and inner-ring complexes is multiple of eight in the assembly intermediates. This is consistent with our previous electron microscopy analysis that demonstrated 8-fold symmetry of the outer-ring complex at 5, 6, 8, 10 and 15 min after anaphase onset and 8-fold symmetry of the inner-ring complex at 10 and 15 min after anaphase onset (Supplementary figure 7, Otsuka et al., *Nat Struct Mol Biol*, 2018, DOI: 10.1038/s41594-017-0001-9); the single tomograms of intermediates indicate that the majority of intermediates have approximate 8-fold symmetry for the outer- and inner rings. However, as the reviewer pointed out, we also think that there are likely to be variations in the copy numbers at the single pore level. In order to assess such variations of Nup stoichiometry experimentally, one would need to perform quantitative live imaging of the addition of individual subunits to single NPCs. Unfortunately this is technically extremely challenging and should be an interesting objective of future studies. Admittedly, if modelling allowed deviation from the 8-fold symmetry in the intermediates, the ensemble of the resulting trajectories would be less precise.

6) The density maps also appear to be symmetric. Again, is this realistic for an individual NPC as compared to the average tomographic reconstruction? Based on the faded-out densities of the maps (Fig. 4a and c) I would assume that individual NPCs at the respective time points lack symmetry and exhibit copy number variations.

Please see the response above.

7) A related issue concerns model scoring. The ET densities are averages of superimposed and likely incomplete assembly intermediates. Is cross-correlation scoring the best way to deal with the resulting fractional densities of elevated symmetry? Was unoccupied density simply not scored?

Again, the reviewer raises a very good point about a limitation of the correlation coefficient for quantifying the match between a model and the density in electron microscopy in general. Unfortunately, we are not aware of any solution to this problem. Nevertheless, the cross-correlation score in our model was chosen as a simple first approximation to rank the candidate shapes of the prepore complexes. Since the target density does not contain sufficient information regarding absolute protein density, we viewed this scoring function as a reasonable and computationally tractable option.

8) The components in the "5min" model shown in Fig. 4c are not all contiguous. Is this an artifact? Does Nup155 (orange) indeed assemble all by itself in the periphery? In the assembled NPC, it borders Nup205 (cyan).

Now we have validated that the assembly kinetics of Nup155 is similar to other inner ring components (Extended Data Figure 7a, b). Whether Nup155 by itself assembles in the periphery of assembly intermediates is not well determined by the model and will be the subject of future studies.

9) In the integrative modeling, native structural contacts were increasingly enforced as the assembly process proceeded. Is this justified?

We made this assumption to allow for more exploration of subcomplex configurations in early snapshots. We found this flexibility allowed for improved fits to the data in early snapshots. Since the mature pore is known to have fully formed and stable native contacts, we scaled the strength of the scoring function linearly between no native interaction at the earliest snapshots and full native interaction in the mature snapshot. In other words, we needed to impose native contacts towards the final stages of assembly or we would not have obtained the native structure sufficiently precisely with the present data about the assembly pathway. At the same time, if we imposed native contacts in the earliest stages of assembly, we would probably over-restrain the search for acceptable assembly pathways, resulting in an overly precise model. Thus, we preferred to err on the side of caution and introduced native contacts progressively along the assembly pathway.

10) For an informed assessment of the models, it will be important to show at the very least zoom-ins on their structural elements. At least the top-scoring model at each time point should be made accessible.

We thank the reviewer for the comment. As the reviewer suggested, we have provided zoom-in pictures of the top-scoring model in the revised manuscript (new Figure 5c). In addition, to make the model accessible, we are depositing the model into the nascent database of integrative structures PDB-Dev (<https://pdb-dev.wwpdb.org/>). The accession code will be available by the time of publication.

11) Integrative structural modeling was performed only for the postmitotic assembly pathway. Were attempts made to model also interphase assembly? What would What, if anything, can we learn about intermediates of interphase assembly?

We focused on the topologically simpler postmitotic assembly. However, with the methodological ground work having been laid, we expect that the interphase assembly that involves more dramatic membrane topology and conformational changes can also be simulated in the future future (e.g. as mentioned in the discussion section of the revised manuscript). A structural model for interphase assembly would provide novel predictions for its unique, inside out evaginating, mechanism as below:

Example 1. Sequence of the molecular events

Our study has demonstrated that the cell not only builds the two nuclear Y-rings but also accumulates the material for their cytoplasmic counterparts and then combines them with the nuclear filament proteins, all prior to the time of membrane fusion. This suggests that the cytoplasmic Y-rings, interestingly including the base of the cytoplasmic filaments with Nup214, are already "prebuilt" within the inner membrane evagination, where they must be present in a very different conformation than in the fully mature pore after fusion. The structural models, if available, can be analysed in terms of the sequence of molecular events, such as the formation of particular protein-protein interfaces.

This analysis will allow us to predict the impact on assembly of knocking out a specific gene, deleting a domain in a protein, or a point mutation at an interface, which can be tested experimentally.

Example 2. Physical principle underlying the inside out membrane evagination

A computational model for interphase assembly would also allow us to uncover the physical principle underlying the inner membrane evagination during assembly. One can calculate how much force is needed to deform and stabilize membrane and examine whether the self-assembly of Nups and their conformational change is enough to produce the force or it requires additional force such as ATP hydrolysis.

Computation of the interphase assembly should certainly be a very interesting objective of future research projects.

Kinetic modeling.

12) The kinetic modeling builds on Ref. 38. For the signaling cascade studied in Ref. 38, the duration of the signal makes sense (as the time over which a "signal" is typically present). Here it is not clear what is meant by duration as compared to the mean time of the assembly. Mathematically, for the assumed irreversible process at each step, τ_i is the mean time to complete assembly step i (equation 10). The so-called "duration" θ_i is the standard deviation of the time to complete assembly step i . Is this distinction relevant here? Furthermore, it is not clear to me how the "duration" (as compared to the mean time) was extracted from the sigmoidal curves.

We completely reworked the section on kinetic modelling. We clarified and renamed some of the concepts. The duration of assembly characterizes the time interval where the bulk of assembly events occur, and the mean time characterizes when on average a specific protein binds. This distinction is important as a protein can assemble late (large mean time) but for a very short period (small duration). This is what we observe for some of the proteins in the interphase assembly. To be specific, in the simple kinetic model with linear rates, τ_i and θ_i characterize the binding event probability density function (PDF) for protein i . For the sigmoid model, the derivative of the assembly curve is also proportional to a binding event PDF. Unfortunately, we cannot use the integral definition as used in the linear model as the standard deviation is not bounded for some parameter combinations. Instead, we use a median binding time and the time interval required to accumulate 80% of the proteins. How these values are calculated is shown in Extended Data Figure 5 and given by the equations 8 and 9 in the revised method section (page 29-33).

13) In the kinetic modeling, I wonder if the delay "d" was subtracted from the times. As written, equations 3 and 4 do not quite make sense. Why should they apply only after time d (as this would result in a "jump" at time t=d)?

We thank the reviewer for pointing this out. Indeed, for the parameter fitting, we used a slightly different expression. We corrected equations 3-4 accordingly in the revised method section (page 29). Now we report the median time, which characterizes the assembly mechanism independently of the delay in Extended Data Table 3 and Extended Data Figure 5.

General points.

14) How were the Nups selected for labeling? Why these ten and not others?

Figure from Kosinski et al., *Science*, 2016. DOI: 10.1126/science.aaf0643. The Nups we selected in this study are highlighted by red underlines.

We selected the Nups that represent all the major building blocks (the selected Nups are highlighted by red underlines in the above figure). Nup107 and Seh1 are components of the stem and the arm of the outer rings (also called Y-complexes), Nup205 and Nup93 form different layers of the inner ring complex, Tpr and Nup153 are the distinct nuclear basket components, and Nup62, Nup214, Nup358, and Pom121 are the components of the other building blocks (the central channel, the cytoplasmic Nup214 complex, the cytoplasmic filaments, and transmembrane nucleoporins).

15) Can we be sure that the slow inner-core assembly is representative of the interphase pathway?

To examine if the NPC assembly in the inner-core region is representative of the interphase pathway, we have investigated whether the surprisingly early recruitment of Tpr also occurs during the more sporadic NPC assembly in S-phase/G2 nuclei. We have immuno-stained NPCs by antibodies against Tpr and a late assembling nucleoporin Nup62 and visualised their distribution at the level of single NPCs by 3D-STED super-resolution microscopy. The data shows that a significant number of NPCs contain Tpr but not Nup62 in S-phase/G2 nuclei (New Figure 4b,c), as is the case in the inner-core region at early G1 (New Figure 4a,c). These results indicate that interphase assembly at later cell-cycle stages follows the same order.

We have added these new STED results in the main body of the text (page 7-8, line 171-180) and provided a new figure (New Figure 4).

16) Fusion of the nuclear envelope to create an initial pore is a critical event in interphase NPC assembly. Can the data be used to time this event?

Definitely. Our kinetic data on interphase NPC assembly, together with our previous correlative light-

electron microscopy (CLEM) analysis of the interphase NPC assembly intermediates (Otsuka et al., *Elife*, 2016. DOI: 10.7554/eLife.19071), indicates the time point when the outer and inner nuclear membrane (ONM and INM) fusion happens most frequently. In our previous CLEM analysis, we had observed that the abundance of mushroom-shaped NPC intermediates under the INM decreased at around 60 min after anaphase onset (AO) when the abundance of mature NPCs increased. Although the data suggested that the ONM/INM fusion happened sometime at 60 min after AO, we were not able to precisely determine when the membrane fusion happens most frequently due to the limited number of cells that we could analyse by the time-consuming CLEM. Now our kinetic analysis showed that the cytoplasmic filament component Nup358 assembles with the average time-point of assembly at 65 min after AO during interphase NPC assembly (Figure 3). Since Nup358 is expected to be recruited to the NPC after ONM/INM fusion, we can deduce that the fusion events happen most frequently at 60–65 min after AO.

17) I suspect that the densities in Fig. 4a are from Ref. 12. Is this correct?

Yes, it is correct. We apologize for the lack of the reference, and have added the reference to the legend (the legend of new Figure 5a).

18) In the time courses of copy numbers (Fig. 2b), there appears to be an early Nup transfer (0-5min) from nucleoplasm to cytoplasm. How should we interpret the drop of the nucleoplasm counts to zero at ~5 min after AO?

The reason why the copy number of Nups drop at around 5 min after AO is the chromosome compaction by axial shortening that has been shown to occur in mid/late-anaphase (Mora-Bermúdez et al., *Nat Cell Biol*, 2007. DOI: 10.1038/ncb1606). The volume of the nucleus is smallest at around 5 min after AO and the large proteins such as Nups are excluded from the nucleus due to the chromosome compaction (Cuylen-Haering et al., *Nature*, 2020. DOI: 10.1038/s41586-020-2672-3). This is the reason why the number of Nups in the nucleoplasm drops to $0.043\text{--}0.22 \times 10^5$, which is too little to see in the plots in Figure 2b. We will provide a source data for the plots so that the readers can examine exactly how much proteins are in each cell compartment during mitosis exit.

19) What do the boxes indicate in Fig. 3b?

One box indicates the cytoplasmic nucleoporin Nup214 and the nuclear basket protein Tpr, and the other box indicates central channel/inner ring Nups. We boxed them in order to highlight the difference in their order of assembly between postmitotic and interphase assembly. We clarified this point in the legend of Figure 3b.

Referee #2 (Remarks to the Author):

In the manuscript entitled “A quantitative map of nuclear pore assembly reveals two distinct mechanisms”, Otsuka et al use FCS to derive assembly rate constants for representative components of the NPC. They find, as was previously proposed, that two pathways exist; a rapid one following mitotic exit and a slow pathway that is more generalized during interphase. This is a well presented and well rounded study integrating live cell imaging with biochemical assembly kinetics and represents, in my opinion, how microscopy data should be utilized to determine molecular mechanism. I have a few minor points and a broad interpretation question listed below.

We thank the reviewer for the appreciation of our work and the thoughtful suggestions for further improvements.

Minor points:

1) There are a few (very few) typos that should be addressed.

We have corrected the typos.

2) I assume the live cell imaging parameters were the same as those listed for the FCS confocal imaging methods. This should be made clear in the methods particularly given the importance of imaging to the conclusions drawn.

We have clarified in the method section that all the live imaging was performed in the same way (page 22, line 437).

Broad Question:

In looking at the data, I feel that a second interpretation could be that the pathways emerge from two different events: assembly and re-assembly. The kinetics presented support the idea that there is in fact a single pathway for assembly that occurs at some rate throughout interphase and is halted in mitosis. Upon mitotic exit the assembly resumes at the “slow” rate. Convolved within this kinetic data is the re-assembly pathway, which consists of fully assembled NPCs breaking down in mitosis and then rapidly reforming upon mitotic exit. In this light, a kinetically slow event in assembly is rapid in reassembly possibly due to some molecular change that occurs upon initial assembly.

Is there a reason that this can not be the case? I would expect to see a slightly more diverse discussion of possible pathways given that the conclusion drawn here are in overall agreement with previous assumptions of NPC assembly. In short, I believe this point should be addressed prior to publication.

We think that there are two main factors that enable the NPC assembly after mitosis so much faster than the NPC assembly during interphase. First, the mitotic cell contains a high concentration of ‘assembly ready’ NPC subcomplexes, already synthesized in the mother cell and disassembled by mitotic phosphorylation. This stockpile of building blocks becomes permissive for assembly synchronously by the reduction in mitotic kinase activity. A second fundamental difference is the topology of the membrane into which the new NPC channels are built. In the mitotic cell, the ER sheets that form the nuclear membrane are highly fenestrated and contain a large number of small membrane discontinuities on the surface of the chromosomes. At this time, NPC assembly would not require a new local membrane fusion event, but could use a pre-existing hole. Interphase NPC assembly by contrast requires a *de novo* local fusion event, and it appears that the formation of the nuclear ring and growing mushroom-shaped density that drives the membrane evagination consumes most of the time required for interphase assembly. Membrane fusion could, thus, be a rate limiting step for the interphase assembly mechanism.

Referee #3 (Remarks to the Author):

Summary of the key results

In cells with open mitosis, NPCs use two distinct assembly modes. At the end of mitosis, NPCs reassemble into the reforming nuclear envelope in a fast and highly synchronous manner, which occurs mainly in the so-called non-core region of the reforming nuclear envelope. Later, NPCs integrate into the closed nuclear envelope in a much slower interphase assembly mode, in early G1 mostly in the core region of the nuclear envelope. In this manuscript, Otsaka et al use FCS-calibrated live-cell imaging in HeLa cells to determine the quantitative nuclear envelope recruitment kinetics of key nucleoporins, the proteins forming NPCs, to generate models of postmitotic and interphase NPC assembly pathways.

The main findings are that core structural components assemble in the same order in postmitotic and interphase assembly: The Y-complex, which is the major building block of the nuclear and cytoplasmic ring structures, precedes inner ring components (Nup93 and Nup205) and the central transport channel (Nup62). Nup358 is integrated late. Surprisingly, the nuclear basket component TPR and the cytoplasmic localized Nup214 are recruited earlier in the interphase assembly pathway as compared to the postmitotic mode. This very extensive and high-quality set of data will be very instrumental for our understanding of NPC structure and function and, as other publications from the Ellenberg lab, a reference point for the field.

We thank the reviewer for the positive feedback on our work and for the constructive suggestions on how to further improve the manuscript.

Originality and significance: if not novel, please include reference

The postmitotic assembly order is consistent with previous studies including publications from the Ellenberg lab, cited in the MS). A few aspects of the interphase assembly orders have been known (Pom121 before the Y-complex, Nup358 late, also properly cited work by the Ellenberg lab) but this comprehensive data set including all different NPC structural components will be very instrumental for the field. The proposed early recruitment of TPR (as compared to postmitotic assembly) is unexpected. It remains, however, open whether this is crucial for interphase assembly.

Data & methodology: validity of the approach, quality of data, quality of presentation

The study uses an innovative and well-concealed research approach. The findings and models in this work are based on high-quality data obtained with FCS-calibrated live-cell imaging and super-resolution microscopy, which are state-of-art methodologies fully adequate to the quantitative purpose of this research. They are reasonably well described and referenced in the materials and methods section. Homozygously mEGFP/mCherry tagging of nucleoporins at their endogenous gene locus is used. The modeling assumptions seem reasonable and falsifiable.

Appropriate use of statistics and treatment of uncertainties

Sample size determination and statistical analysis are well described, although no Pearson's correlation or p-values are found in the manuscript.

Conclusions: robustness, validity, reliability, Suggested improvements: experiments, data for possible revision

- In lines 112-115 and Figure 2B, the range of each nucleoporin molecule per cell is indicated. These values are key results and are necessary for the later quantitation and models. For these calculations, a precise account of single-cell cytoplasmic and nuclear volumes is mandatory. While the volume from the nuclei has been calculated by the DNA counterstaining in each cell from this study, a single

average value for the cytoplasm volume from another publication of the authors has been used for the calculations. For more accuracy and to avoid that the values given in this work are inexact due to the cell-to-cell variability (clearly visible in the figure 2A) or to experimental factors that might affect the cell volume, I suggest validating the values of cytoplasmic volume in, at least, a subset of the presented data. Performing the experiments measuring the cell volume as the authors describe in Cai, Y. et al 2018 would be an option. An alternative proxy to validate the results would be to determine the single cell volume using the volumetric segmentation of the eGFP images in those cells in metaphase and early mitotic exit (i.e. the cytoplasmic signal of the Nups).

We appreciate the useful comment. As the reviewer suggested, we have quantified the volume of our mEGFP-Nup knockin cells at metaphase using the cytosolic signal of the Nups. The cytosolic GFP signal of some of the Nups (e.g. Nup205, Nup214, Tpr, Nup358) are too low to precisely segment the cytosol in the z-slices close to the glass surface due to the relatively high background fluorescence signal. Therefore, we have measured the cell volume using the brightest Nup62-mEGFP cell line (the volume was $4900 \mu\text{m}^3$ with the s.d. of $330 \mu\text{m}^3$, $n=8$ cells), which was similar to the value that had been determined in a histone H2b-mCherry cell line using fluorescent dextran ($5300 \mu\text{m}^3$; Cai et al., *Nature*, 2018. DOI: 10.1038/s41586-018-0518-z). For the other Nup cell lines, we have measured the cytosol area on a middle z-slice plane and calculated the ratios to the one of the Nup62 cell line (the ratios were 0.97, 0.97, 1.09, 1.14, 1.03, 1.07, and 1.02 for Nup107, Seh1, Nup205, Nup93, Nup214, and Nup358, respectively). From the measured volume of the Nup62-mEGFP cell and the ratios of cytosol area to the Nup62 cell, we have estimated the volume of other cell lines (the volume was 4670, 4660, 5550, 6030, 5110, 5420, and $5080 \mu\text{m}^3$ for Nup107, Seh1, Nup205, Nup93, Nup214, and Nup358, respectively). Assuming that the cell volume changes in the same degree during mitosis exit as the H2b-mCherry cell, we calculated the cytoplasmic volume of the Nup cell lines using the ratio to the volume at metaphase. With the new values of the cytoplasmic volume, we have re-calculated the number of Nups in each cell line.

We have replaced the plots in Figure 2b and mentioned the above modification in the method section (page 27-28, line 557-581).

- line 179-182: the authors argue that the poorer correlation between duration and time point of assembly of nucleoporins in the interphase assembly mode suggests a less strictly sequential pathway. Could the differences in correlation be due to a highly synchronous initiation of postmitotic NPC assembly in telophase whereas for interphase mode assembly of NPCs is initiated during a much longer period, in principle the entire G1 phase/interphase?

We thank the reviewer for the comment. Our previous observation had shown that the initiation of interphase assembly is mostly happening within 1 hour after anaphase in the core region (Otsuka et al. *Elife*, 2016. DOI: 10.7554/eLife.19071). It turns out that our conclusion with respect to the assembly order holds true also if initiation is not perfectly synchronous but occurs within a defined time period. In the revised method section, we have modified the kinetic linear rate constant model accordingly to include the possibility of a prolonged initiation (page 29-33).

- line 242-244: the authors argue that the early recruitment of TPR in interphase assembly key role of the protein in the assembly mode. Experimental evidence would significantly strengthen the manuscript. Of note, early recruitment, e.g. of Nup153 in postmitotic NPC assembly does not necessarily correlate with the requirement of this Nup for postmitotic NPC assembly.

We thank the reviewer for the thoughtful suggestion. To try to address this point, we obtained a DLD-1 cell line in which Tpr can be acutely degraded in an auxin-inducible manner from Marry Dasso's Lab (Aksenova et al., *Nat Commun*, 2020. DOI: 10.1038/s41467-020-18266-2). We transfected these

cells with fluorescently-tagged Nups and tried to examine if the assembly is altered after acute Tpr depletion. Although we were able to monitor Nup assembly during mitotic exit in DLD-1 cells, we could not separately extract postmitotic and interphase assembly kinetics from the non-core and core regions, because the nuclei of DLD-1 cells rotate rapidly after cell division, precluding reliable identification of the core regions. Unfortunately, these experiments to gain further insights into the Tpr requirements in interphase assembly have therefore remained inconclusive for the moment. Building new, more precisely controlled tools to achieve them is in our view beyond the scope of a revision of the current study. In the revised manuscript, we have modified our statement on the role of Tpr in interphase pathway to indicate that early recruitment suggests, but does not prove an early requirement (page 10, line 237).

References: appropriate credit to previous work?

Previous work is properly cited.

Clarity and context: lucidity of abstract/summary, appropriateness of abstract, introduction, and conclusions

The manuscript is clearly written and easy to follow. The abstract summarises the important findings.

Additional points:

- Clarify the terminology disagreement between text (line 98) and Extended Data Fig. 1. Nup153 and Pom121 are excluded from quantitation in figures 1 and 2 because, as is indicated in the text, they display lower concentrations likely due to “subhomozygous” tagging. However, in supplementary figure 1, Nup153 is clearly heterozygous, as indicated. The Pom121 clone used in this work is seemingly homozygous. If by “subhomozygous” it is referred the low MW band in the Southern blot in the clone 32 from Pom121, a clarification about the meaning of this term would be recommended (it does not exist as a concept as far as I know and could check).

We have clarified this point in the legend of Extended Data Figure 1.

- For DNA counterstaining, SiR-DNA is referred in most of the text but in line 400 and in Extended Data Fig. 2 the alternative term (SiR) Hoechst is used. Consistent terminology would be recommended.

We have changed the term from “SiR-Hoechst” to “SiR-DNA” in the legend of a new Extended Data Figure 3. Since we need to specify how we labelled DNA in the method section, we did not change the term “SiR-Hoechst” in the method.

- In Figure 2B, plots displaying the range and distribution of the values in the temporal series would be recommended to better communicate the variability of the data set.

Since displaying the distribution of the values in the small plots of Figure 2b is challenging, we will provide a source data with the values of standard deviation.

Line 207-209: The statement that the nuclear ring first assembly mode (of postmitotic NPC assembly) is consistent with the eight-fold symmetric protein density uses ref 11, which is a study on interphase NPC assembly. It would be appropriate to mention this explicitly, e.g. as “... on the inner nuclear membrane at early stages of interphase NPC assembly”

We are afraid to say that the ref 11 is a study on postmitotic assembly, and not on interphase assembly. In the study of ref 11, an eight-fold symmetric protein density was observed on the inner nuclear membrane at early stages of postmitotic assembly.

- line 495: *Lap2alpha* is used as a core marker. Given the reported nuclear localization of this protein in contrast to NE localization to other *Lap2* isoforms this is unexpected. Is this a typing error?

This is not a typing error. It is true that Lap-2 α localizes in the nucleus after telophase, but it transiently localizes in the core region of the chromosome in late anaphase/early telophase (Dechat et al., *J. Cell Sci.*, 2004. DOI: 10.1242/jcs.01529).

Referee #4 (Remarks to the Author):

A- Summary of the key results

In this manuscript, the authors quantitatively analyzed the timing of recruitment to the NE of nine GFP- and one mCherry-tagged Nups (generated by genome editing in HeLa cells) in the two hours following mitotic exit. Based on this extensive analysis, and using the fact that, as previously described, NPCs first assemble after mitosis in the "non core" region of the reforming NE through the post-mitotic pathway, and then following the "interphasic" pathway in the "core" region of the NE, they confirm previous data regarding the fact that post-mitotic and interphase NPC assembly proceed by strikingly different molecular mechanisms. Novel to this study, their data indicate that unlike in post-mitotic assembly, the nuclear basket protein TPR and the cytoplasmic filament protein Nup214 are recruited before the central ring complex during the interphase assembly process.

Finally, they used computational modeling to combine these data with their previous single pre-pores subtomogram averaging (Otsuko et al., 2018), to propose an integrative modeling of the post-mitotic assembly pathway of the main NPC scaffold, with a refined order of recruitment of its structural modules, revealing stepwise assembly of the 4 copies of Y-complexes, nested with the assembly of the inner ring.

B. Originality and significance:

While this study is well-performed and will be of interest to specialized audience eager to understand the exquisite details of NPC assembly pathways, it may not provide enough novelty, either conceptually, mechanistically, or technically, to engage a broader audience.

We agree with the reviewer that we should have explained the novelty of our study more clearly, as was also pointed out by reviewer#1. This is partly due to the space restrictions of the journal format. In order to clarify the novelty and impact of our work and make our manuscript more accessible for readers from diverse fields, we have rewritten large parts of the discussion (page 9-12). In addition, as the reviewer suggested, we have performed numerous experiments to corroborate the key findings of our manuscript. Encouraged also by the endorsement of reviewers #2 and #3, we are convinced that our study provides novelty for a broader audience, as explained point-by-point below.

Indeed, as stated by the authors, it was previously demonstrated that the "postmitotic and interphase NPC assembly possess distinct kinetic, molecular and structural features". For instance, the inverse order of recruitment of the Y-shaped complex and Pom121- Dultz, E. & Ellenberg, J. Cell Biol 2010). Therefore, the title of the manuscript is somewhat misleading.

It is correct that some kinetic data have been available for a small subset of Nups for interphase assembly pathway (Pom121, Nup107 and Nup93; D'Angelo et al., *Science*, 2006. DOI: 10.1126/science.1124196; Dultz & Ellenberg, *JCB*, 2010. DOI: 10.1083/jcb.201007076) and the inverse order of recruitment of Nup107 and Pom121 has been shown between postmitotic and interphase assembly. However, no kinetic data have been available for interphase assembly for other key Nups composing cytoplasmic filaments, central channel, and nuclear basket, nor for distinct subunits of the Y-shaped and inner-ring complexes. Moreover, the previous studies relied on ectopic expression of fluorescently-tagged Nups and therefore could only provide relative kinetic data, which has limited the ability to obtain insights into the assembly mechanism.

In this study, we therefore conceptually advanced the field of interphase assembly, by revealing the changes in subunit composition and stoichiometry during interphase assembly of ten Nups that represent all the major building blocks of the NPC. Our data for the first time systematically identified the key mechanistic differences between the two assembly pathways. During postmitotic assembly,

the Y-complex is rapidly combined with components of the central ring, building the inner core of the pore prior to addition of either cytoplasmic or nuclear filament proteins, that follow much later. By contrast, during interphase assembly, the Y-complex is first combined with the nuclear filament protein Tpr and the base of the cytoplasmic filament Nup214, while the central ring complex is added only afterwards. To our best knowledge, it is for example a novel and unexpected finding that the central ring and nuclear filaments follow an inverted molecular order between the postmitotic and interphase NPC assembly.

To provide additional evidence for the surprisingly early recruitment of Tpr, we have now immuno-stained NPCs by antibodies against Tpr and the later assembling nucleoporin Nup62 and visualised their co-occurrence in single assembling interphase pores in early G1 by 3D-STED super-resolution microscopy. This single NPC imaging could directly demonstrate that a large number of assembling NPCs in the core region, where interphase assembly dominates, contain Tpr but not Nup62, while most of the assembled NPCs in the non-core region of the same cells, stemming from postmitotic assembly, contain both Tpr and Nup62 (New Figure 4a,c). We have also examined if this surprisingly early recruitment of Tpr also occurs during the more sporadic NPC assembly later in interphase in S-phase/G2 nuclei, and indeed found that also here a significant number of NPCs contain Tpr but not Nup62 in S-phase/G2 nuclei (New Figure 4b,c). Both of these results confirmed the surprisingly early recruitment of Tpr in interphase assembly pathway, and suggest that there is no “third pathway” of later interphase assembly, as the assembly order was consistent between early G1 and G2 for these Nups.

As a consequence, with the exception of hypotheses regarding evolution, largely based on their observation of an early Tpr recruitment in interphase as compared to its reported late recruitment in budding yeast, no new concept arises in the discussion.

We apologize our very short description of novelty due to the limited space in the text. In addition to the implication for the evolutionary origin of the NPC and the unexpected early recruitment of Tpr in interphase, we think that our study makes two more conceptual, mechanical and technological advances as below:

1. Mechanistic insights into postmitotic NPC assembly from integrative structural modeling

Our dynamic stoichiometry data enabled us to build the first structural model of postmitotic NPC assembly intermediates from the data by computer simulations. The structural model allows to make multiple very interesting mechanistic predictions, such as for example that the central ring complex might be needed early in postmitotic assembly to prevent sealing and promote expansion of ER holes after mitosis, and the hydrophobic FG-repeats of central channel Nups might play a role in the initial dilation of the small membrane hole into the larger, NPC sized channel. These hypotheses illustrate the value of our quantitative data set and the structural assembly model we were able to derive from it.

As the reviewer suggested we have additionally investigated the assembly kinetics of two previously not studied nucleoporins, Nup155 and Nup188 in the revised manuscript (Extended Data Figures 7 and 8, the details are explained below in the section of “*F- Suggested improvements: experiments, data for possible revision*”), and validated our integrative model predictions. We believe that our data adds the much-needed information about molecular mechanism and difference between the two pathways of NPC assembly, that were not provided by the previous relative time-course analysis of Nup assembly.

2. Providing a new paradigm for studying dynamic molecular machines inside cells

Having laid the methodological ground work for dynamic structure simulations of assembly, we expect that processes such as interphase assembly that involves much more dramatic membrane

topology and protein conformation changes can also be simulated in the future. Especially, if additional experimental constraints are obtained from emerging methods such as 3D super-resolution microscopy to image the molecular architecture of the NPC intermediates (Thevathasan et al., *Nat Methods*, 2019. DOI: 10.1038/s41592-019-0574-9; Sabinina et al., *Mol Biol Cell*, 2021. DOI: 10.1091/mbc.E20-11-0728) or dynamic super-resolution microscopy such as MINFLUX to follow some of the conformational changes in real time in living cells (Gwosch et al., *Nat Methods*, 2020. DOI: 10.1038/s41592-019-0688-0), integrative structure modelling will become possible for interphase assembly similar to what we have achieved for the postmitotic pathway in this study.

Beyond our conceptual advances for understanding the mechanism of nuclear pore complex assembly, we expect that the combination of quantitative 4D molecular mapping inside cells and integrative computational structure modeling of dynamic molecular processes we employed here will provide a useful approach for how to study the assembly mechanisms and functional cycles of other large molecular machines inside cells.

We have clarified these points in the discussion (page 9-12).

On line 247, that authors state : « The order of Nup assembly for interphase assembly we observed here is consistent with the recently reported order of NPC assembly in budding yeast that undergoes a closed mitosis, except for the Tpr homologues Mlp1/2 which assemble late in yeast ». Yet, one of their key novel hypothesis, namely « the unexpectedly early presence of Tpr suggests a key role of this large coiled-coil protein during interphase assembly » is not tested by additional experiments.

In the later section F (*F- Suggested improvements: experiments, data for possible revision*), the reviewer suggested how to examine the Tpr-requirement in interphase pathway; i.e. *by performing functional studies, for instance by combining this quantitative analysis with acute degron-induced depletion of TPR in mitosis (see study from the Dasso lab, Nat Commun. 2020).*

We thank the reviewer for suggesting these experiments. We obtained a DLD-1 cell line in which Tpr can be acutely degraded in an auxin-inducible manner from Marry Dasso Lab (Aksenova et al., *Nat Commun*, 2020. DOI: 10.1038/s41467-020-18266-2). We transfected the cell with fluorescently-tagged Nups and tried to examine how the assembly is altered by acute Tpr depletion. Although we were able to monitor Nup assembly during mitotic exit in DLD-1 cells, we could not separately extract postmitotic and interphase assembly kinetics from the non-core and core regions, because the nuclei of DLD-1 cells rotate rapidly after cell division, precluding reliable identification of the core regions. Unfortunately, these experiments to gain further insights into the Tpr requirements in interphase assembly have therefore remained inconclusive for the moment. Building new, more precisely controlled tools to achieve them is in our view beyond the scope of a revision of the current study. In the revised manuscript, we have modified our statement on the role of Tpr in interphase pathway to indicate that early recruitment suggests, but does not prove an early requirement (page 10, line 237).

Moreover, the statement that the order they observed is “consistent” with the one reported by Onischenko et al. (Cell 2020) does not seem to reflect the data presented in that publication. Indeed, in their recent study, Onischenko et al. conclude "Taken together, NPC maturation begins with the symmetric NUPs, continues with the asymmetric ones, and concludes with the Mlp proteins". Hence, in budding yeast, there is to my knowledge no evidence that the central ring assembles after Y-complex. In addition, Onischenko et al. describe an unexpected asynchronous assembly of the stem and the head NUPs of the Y complex in budding yeast, but this is not observed by Otsuka et al in this ms.

We appreciate the useful comment. Indeed, there are striking differences in the interphase pathway between human and yeast cells. In human cells we distinctly observed that the central ring components assemble after the Y-complex components (Figs. 2, 3, and new Extended Data Figure 7a). In addition, human cells exhibit a more synchronous assembly of Y-complex components, while in yeast the stem (e.g. Nup107) and the head (e.g. Seh1) of the Y-complex do not assemble simultaneously (Figs. 2, 3).

We have extended the discussion about the difference between yeast and human in the order of NPC assembly in the revised text (page 10-11, line 245-252).

Here, a possible reason/caveat might be the choice of Otsuka et al to follow, for technical reasons, the interphase assembly of NPCs in telo/G1 cells and not their assembly between G1 and G2, when NPC number slowly doubles. From their quantitative study, the authors conclude that « NPC assembly relies on and consumes almost half of the material inherited from the mother cell within one hour after mitosis ». Could this assembly path be different at later stages of the cell cycle, when assembly would possibly rely on newly-synthesized components ? Could that explain the observation that in their study, « cytoplasmic Y-rings, including the base of the cytoplasmic filaments with Nup214, are already “prebuilt” within the inner membrane evagination », which is not observed in budding yeast ?

Might there be in mammalian cells, three pathways of NPC assembly, post-mitotic, Telo/G1 and S/G2 ?

This is indeed a very interesting aspect, as a third NPC assembly pathway might exist in mammalian cells. To examine this possibility, we have investigated whether the surprisingly early recruitment of Tpr also occurs during the more sporadic NPC assembly in S-phase/G2 nuclei. We have immunostained NPCs by antibodies against Tpr and a late assembling nucleoporin Nup62 and visualised their distribution at the level of single NPCs by 3D-STED super-resolution microscopy. The data shows that a significant number of NPCs contain Tpr but not Nup62 in S-phase/G2 nuclei (New Figure 4b,c), as is the case in the inner-core region at early G1 (New Figure 4a,c). These results indicate that interphase assembly at later cell-cycle stages follows the same order.

We have added these new STED results in the main body of the text (page 7-8, line 171-180) and provided a new figure (New Figure 4).

C- Data & methodology: validity of approach, quality of data, quality of presentation

The approach is appropriate, yet, as discussed above, it only allows to follow an early and possibly specialized phase of “interphase” NPC assembly.

The “FCS qualibrated live Imaging” highlighted in their abstract (carried out as described in a previous report), is based on calibration in interphase cells using GFP-Nup107, while other values were based on the comparison of other cell lines with the Nup107 cell line, acquired the same day in a distinct well.

Hence, this approach mainly serves here to validate that the tagging does not affect the incorporation of Nups in NPCS. Indeed, the same conclusion could be reached from the previously established copy number of these Nups in mature NPCs (the fact that there are 32 copies of Nup107 per mature NPC, would have been sufficient to calibrate the confocal images of the other GFP-Nups).

How Pom121-mCherry copy number was quantified is unclear.

We apologize our short description of the method for quantifying the copy number of Pom121-mCherry. For Pom121-mCherry, we have performed FCS using Alexa Fluor 568 NHS ester instead of Alexa Fluor 488 NHS ester to measure the confocal volume, and then performed FCS using the cells

that transiently-express mCherry alone instead of mEGFP to obtain a calibration curve for converting fluorescence intensity to the concentration.

We have clarified these points in the revised method (page 24, line 476-479).

D- Use of statistics and treatment of uncertainties seems appropriate

E- Conclusions: robustness, validity, reliability

I do not feel qualified to determine how robust their integrative model is. Yet, as Nup155 was not imaged in this study, it seems that its positioning in the model mainly arises from the previous EM data. May that affect the overall interpretation?

As the reviewer pointed out, we did not image Nup155 although our integrative model predicted its key role in the assembly intermediate structure. Therefore, we decided to experimentally validate that Nup155 assembles as the model predicted. Since it is unfortunately technically very challenging to generate a homozygous GFP-Nup155 knock-in cell line due to the multi-allelic nature of this gene in the HeLa cell genome, we have imaged endogenous Nup155 in fixed cells in late-anaphase, early- and mid-telophase using an anti-Nup155 antibody (Extended Data Figure 7a, b). We have used mEGFP-Nup107 and Nup62-mEGFP genome-edited cells to compare the assembly timing of Nup155. The immune-fluorescence microscopy has shown that Nup155 assembles at a similar timing to Nup62, which was consistent with the model prediction.

We have added this point in the main body of the text (page 9, line 203-204) and provided a new figure in the extended data (Extended Data Figure 7).

Another possible limit is that while individual NPCs are imaged in interphase cells thanks to STED microscopy, the quantitative data are based on confocal microscopy. Hence, possible recruitment of several Nups to the NE rather than to assembling NPCs cannot be excluded.

In order to exclude the possibility that the Nups are accumulated on the NE rather than recruited to the NPCs, we have visualised localization of one of the Nups Nup62-GFP using GFP-Nanobody by 3D-STED super-resolution microscopy in early telophase cells. The STED imaging showed a discrete localization of Nup62 along the NE (Extended Data Figure 2a). The density of the Nup62 spots was 14.6 ± 2.6 NPCs/ μm^2 , which is consistent with our previous EM observation of the NPC density of 12–16 NPCs/ μm^2 in early telophase cells (Otsuka et al., *Nat. Struct. Mol. Biol.*, 2018. DOI: 10.1038/s41594-017-0001-9. Supplementary Figure 4). We have also examined the localization of GFP-Nup107 together with an earliest-recruited Nup Elys using anti-GFP and anti-Elys antibodies (Extended Data Figure 2b). The 2D-STED imaging showed that GFP-Nup107 co-localizes with endogenous Elys in early telophase cells. These pieces of evidence support our assumption that the Nups are indeed recruited to the NPCs rather than nonspecifically accumulated on the NE.

We have added this point in the main body of the text (page 5, line 121-123) and provided a new figure in the extended data (Extended Data Figure 2).

F- Suggested improvements: experiments, data for possible revision

*To strengthen this study, the authors may consider to validate their hypothesis regarding the role of TPR in NPC assembly by performing functional studies, for instance by combining this quantitative analysis with acute degron-induced depletion of TPR in mitosis (see study from the Dasso lab, *Nat Commun.* 2020).*

We thank the reviewer for suggesting these experiments. As we described above, we obtained a DLD-1 cell line in which Tpr can be acutely degraded in an auxin-inducible manner from Marry Dasso Lab

(Aksenova et al., *Nat Commun*, 2020. DOI: 10.1038/s41467-020-18266-2). However, we could not extract interphase assembly kinetics because the nuclei of DLD-1 cells rotate during cell division and it was hard to identify where the core regions are. Unfortunately we cannot gain further insights into the Tpr requirement in interphase assembly at the moment, and it will be the subject of future studies. In the revised manuscript, we have modified our statement on the role of Tpr in interphase pathway, and clarified the other aspects of the novelty of our study in the discussion (page 9-12).

Imaging of the core region of the Telo/G1 cells by STED, combining the GFP lines with antibodies (or using a double-labelled GFP/mCherry line) may help to strengthen their conclusion regarding TPR or inverted assembly pathways.

We thank the reviewer for the constructive suggestion. We have immuno-stained NPCs by antibodies against Tpr and a late assembling nucleoporin Nup62 and visualised their distribution in the core region of early-G1 cells by 3D-STED super-resolution microscopy. The single NPC imaging has demonstrated that a large number of assembling NPCs in the inner-core region in early G1 contain Tpr but not Nup62, while most of the assembled NPCs in the non-core region of the same cells contain both Tpr and Nup62 (New Figure 4a,c), confirming the surprisingly early recruitment of Tpr in interphase assembly pathway.

We have added the result in the main body of the text (page 7-8, line 171-180) and provided a new figure (New Figure 4).

- The postmitotic assembly model would deserve validation by quantifying Nup155 (and Nup188) recruitment (which would require to generate the corresponding GFP-cell lines). Indeed, Nup155 appears in their model in two distinct complexes as well as an isolated component.

In addition to Nup155, we have also performed a validation experiment for our integrative model using Nup188. We have generated a homozygous GFP-Nup188 knock-in cell line by genome editing and examined the assembly kinetics of Nup188 in postmitotic and interphase assembly pathways by live imaging as we did for other Nups (Extended Data Figure 8). The experiment has demonstrated that the assembly kinetics of Nup188 is similar to other inner ring components (Nup93 and Nup205) as the model predicted. Regarding the Nup155 validation experiment, it is described above.

We have added this point in the main body of the text (page 9, line 203-204) and provided a new figure in the extended data (Extended Data Figure 8).

- Nup62 does not solely belong to the central channel, but also interacts with Nup214; how is this taken in account in this model?

In our structural model, we used a pseudoatomic model of the mature NPC of Martin Beck's lab (von Appen et al., *Nature*, 2015. DOI: 10.1038/nature15381; Kosinski et al., *Science*, 2016. DOI: 10.1126/science.aaf0643), which does not include the Nup214-bound fraction of Nup62. Our structural model predicts 32 copies of Nup62 in the fully mature NPC, although there are 48 copies. It might indeed be the case that the remaining 16 copies of Nup62 are bound to Nup214.

We have clarified these points in the revised method (page 34, line 701-704).

-The postmitotic assembly model does not include the peripheral components. Is the timing of Nup358 recruitment compatible with its implication in the assembly of the second Y outer ring on the cytoplasmic side as previously proposed (von Appen, Nature 2015) ?.

Our model excludes Nup358 and so cannot directly account for its interaction with the Y-complex. However, given that Nup358 arrives late in the postmitotic assembly, it seems that it is potentially not necessary for initial assembly of the cytoplasmic Y-complex.

Minor comments:

The authors should clarify figure 4.

- the origin of the data for panel (a) should be stated in the figure legend by referring (I assume) to Otsuka et al., 2016).

(the protein density could be colored differently to be better distinguished from the NE surface model).

We apologize for the lack of the reference, and have added the reference to the legend (the legend of new Figure 5a). In addition, we have modified the color of the structural model in the figure.

- What are the protein densities at 5 min and 6 min on the nuclear side of the NPC that subsequently disappear at 8 min (panels a)? Why don't they appear on panel c?

The dome-like density at 5 min and 6 min is the noise from electron tomography. It is not apparent at the later times as the protein signal becomes stronger and is removed by thresholding the density to present a clear view of the data. What the panel C shows are the protein densities that the structural model predicts and not the electron tomography densities. We apologize that our limited explanation of this figure led to misunderstanding.

We have clarified these points in the legend (the legend of new Figure 5a).

- The similar colors in b and c refers to Nucleoporins in (b) and nucleoporin complexes in (c). While this is fine for Nup107, that only belongs to the Y-shape complex, it leads to confusion as Nup93, colored in blue in (b), appears in the blue and cyan complexes in c.

We thank the reviewer for the comment. We have improved the color of the Nup93 complex in the revised manuscript (new Figure 5b, c).

Nup155 is considered to be part of 2 different complexes and also appears as isolated protein (orange)....

Indeed, Nup155 is considered to be part of inner-ring complexes and also reside at the pore as isolated proteins. This is from a previous pseudoatomic model of the native NPC structure based on cryo-electron microscopy observations (von Appen et al., *Nature*, 2015. DOI: 10.1038/nature15381; Kosinski et al., *Science*, 2016. DOI: 10.1126/science.aaf0643). Since our integrative model predicts that the isolated Nup155 plays a key structural role in the assembly intermediates, we had distinguished it from the Nup155 forming the inner-ring complexes by highlighting it with different color.

In the revised manuscript, we have clarified this point by highlighting Nup155 in the figure and the legend (new Figure 5b, c). In addition, we have provided zoom-in pictures of our integrative model (new Figure 5c) in order to better see the location of the isolated Nup155 in the NPC assembly intermediates.

Line 184: Integrative modeling of the NPC assembly pathway

Title of this chapter is misleading as only the postmitotic assembly pathway or the core NPC is modeled.

We have modified the text to make it clear that the integrative modelling was performed only for the postmitotic NPC assembly pathway (page 8, line 182).

G- References: appropriate credit to previous work

General comment: Wherever this study will be published, the authors should be careful to properly

acknowledge other teams who previously made the critical discoveries and to clarify what was already known, even from their previous studies (previously published lines, methodologies,...)

Although previous studies are in general cited, the way they are cited in the text is not appropriate, as it gives most of the credit to the previous contribution of the authors.

- As example, line "As we and others have demonstrated previously (12,24),...."

This does not give proper credit to Maeshima et al. (ref 24) who initially demonstrated the use of pore-free islands in the inner core region, that was subsequently used by Otsuko et al in 2016 to generate the two-component model (ref 12) that is reused in this study.

We did not intend to undervalue the studies of others. We had to shorten/simplify the text due to the space restriction of the journal format. In order to avoid such misunderstandings, we have modified the text (page 6, line 127-131).

- likewise, the title of the manuscript "A quantitative map of nuclear pore assembly reveals two distinct mechanisms" is misleading since the existence of two distinct mechanisms was previously established by multiple labs .

We do not fully agree that the existence of two distinct mechanisms was previously established.

Although it has been shown that the molecular requirements are different between postmitotic and interphase assembly pathways (Franz CR et al, *EMBO Rep*, 2007; Doucet et al, *Cell*, 2010; Talamas JA et al, *JCB*, 2011; Funakoshi et al, *MoBC*, 2011; Vollmer B, *Dev Cell*, 2015; Rampello et al, *JCB*, 2020), little has been known about the Nup assembly order and kinetics for interphase pathway due to the rare and sporadic nature of the interphase assembly. As we described above, our data for the first time revealed subunit composition and stoichiometry during postmitotic and interphase assembly of ten Nups that represent all the major building blocks of the NPC. To our knowledge, this is the first time that sufficiently comprehensive and quantitative data is available to allow integrative structure modeling of the assembly process, which allows to make mechanistic predictions. The model and our data have revealed for the first time systematically how the molecular mechanism of the two assembly pathways are different, as summarized in the discussion of the manuscript, with the major difference being that the central ring and nuclear filaments follow an inverted molecular order between the postmitotic and interphase NPC assembly.

Similarly, it is not appropriate to write (line 214) "Our data revealed that the two assembly pathways employ strikingly different molecular mechanisms", while they mainly confirmed that concept.

We have rewritten the discussion to clarify the novelty and impact of our work in the revised manuscript (page 9-12).

As discussed above, the recent study regarding NPC assembly in budding yeast (Onischenko et al. Cell 2020) is only discussed in respect to timing of TPR recruitment. Yet, other striking differences are neither mentioned nor discussed.

We thank the reviewer for pointing this out. As already mentioned above, we have extended the discussion about the difference between yeast and human in the order of NPC assembly in the revised text (page 10-11, line 245-252).

The observation from the Hetzer lab that Tpr depletion leads to an increase in the total number of NPC per cell nucleus (Genes and Dev, 2018) seems to be inconsistent with their hypothesized requirement of TPR in interphase assembly; yet, this paper is not cited.

The requirement of Tpr in interphase assembly is not necessarily inconsistent with the observation from Hetzer lab (McCloskey et al., *Genes & Dev*, 2018. DOI: 10.1101/gad.315523.118). McCloskey

et al has shown that Tpr is required for phosphorylating Nup153, which interferes with its recruitment of the Nup107/160 complex and thus prevents new NPC assembly in the proximity of existing NPCs. The early recruitment of Tpr in interphase assembly suggests a role of Tpr in preventing de novo NPC formation in the vicinity so that the NPC assembly intermediates can be isolated enough not to interfere with other NPCs. We have added this point to the discussion and cited the McCloskey et al paper in the revised manuscript (page 10, line 240-244).

Conversely, the authors do not always clearly state that they had already published/described some of the tools or experimental approaches used in this study.

- For instance, when reading "To quantitatively analyze.....we genome-edited HeLa cells, homozygously tagging the endogenous genes for ten different Nups with mEGFP or mCherry", as no references are included, except one reference to a method paper (in which GFP-Nup107 was generated along many other lines), one expect that these cell lines were new tools generated for this study. Yet, reading the method it appears that only 4 out of the 10 cell lines presented were generated in this study, of which two are heterozygous. The other ones were previously published by the Ellenberg's team.

Proper phrasing would be: 'We used 10 genome-edited HeLa cell lines of which 6 were previously published.'

We have modified the text accordingly (page 4, line 75-82).

- likewise, in line 127, the authors should clearly state that they had previously used the two-component model of a fast (postmitotic) and a slow (interphase) assembly process to fit the experimental data regarding Nup107 and Nup358 assembly (Otsuko et al, Elife 2016, Fig 4).

We actually modified our previous two-component model. Our current model allows to determine the fraction of postmitotic and interphase assembly in an unbiased way. We have clarified this point in the main text and methods (page 6, line 132; page 29-33).

Reviewer Reports on the First Revision:

Referees' comments:

Referee #1 (Remarks to the Author):

As stated in my previous report, NPC assembly has great biological relevance and is -- in light of the complex structure -- of broad interest. The detailed measurements, the integrative modeling, and the insights gained in the present work do make a significant contribution. In the revision, the authors have made a serious effort to address all of my concerns. Points 4, 5 and 6 require further (minor) clarifications.

4) Variations in assembly pathway. I found it intriguing that the dominant pathway (85%) with early nuclear-ring assembly appears to compete with a cytoplasmic-ring-first pathway (15%; see new Ext. Data Fig. 6). Is this an expected uncertainty or an artifact of the modeling, or do the two assembly paths indeed run in parallel? Are there experiments with which one could exclude flux through the alternative pathway? I wonder if this should at least be commented on?

5) Imposition of symmetry in modeling. I agree that at this stage it is reasonable to restrict the modeling to symmetric intermediates. However, I am confused by the assertion on p. 35 that "the assumption that the subcomplexes preform with 8-fold multiplicity before assembling into the NPC is supported by our diffusion coefficient measurement of the GFP-labeled Nups by FCS." As I interpret the FCS data, the Nups are all part of larger complexes. However, I do not see any concrete evidence for having subcomplexes present in stoichiometric numbers ("8-fold multiplicity") at each site of NPC emergence. Please delete or clarify.

6) Faded-out tomogram density. I may not have been clear in the way I formulated my earlier point on the tomogram density of intermediates fading out towards the periphery. One possible (and to me likely) cause for the partial density is that the ET reconstructions are averages over partially incomplete assembly intermediates, each one of which may not obey full C8 symmetry in the faded-out parts. A second possible factor is that some of these parts are dynamic. To be clear, I am fine with modeling under the assumption of 8-fold symmetry at this stage, I would have preferred a more cautious wording

Referee #2 (Remarks to the Author):

I appreciate the authors' response and the detailed rebuttal they have provided. I believe the revised manuscript is much improved. In particular I find the discussion of how two separate NPC assembly pathways are deconvolved using their in vivo approach is novel and now appropriately described in the new version. Derivation of biochemical parameters using live cell imaging is powerful and underutilized in Cell Biology, thus it is critically important that work of this nature appear in top line journals.

I fully support publication at this time.

Referee #3 (Remarks to the Author):

The authors have satisfactorily addressed the reviewers' comments. I still think this is an important work which will help the scientific community understanding nuclear pore complex assembly pathways.

I have one question which the authors might want to comment on in the manuscript: I am astonished by the low Nup205 copy numbers (Fig 1C). This is contradicting previous data from the Beck lab (<https://doi.org/10.1038/msb.2013.4>) which suggested that Nup205 is equally abundant as members of the Y-complex. It is also at odd with the recent NPC structures e.g. by the Hoelz

lab (doi: <https://doi.org/10.1101/2021.10.26.465796>). In addition, Nup188 is expected to be present in half the copy numbers as compared to Nup205 (again <https://doi.org/10.1038/msb.2013.4> and <https://doi.org/10.1101/2021.10.26.465796>).

Referee #4 (Remarks to the Author):

In the revised version of their manuscript, the authors have taken in account most of my comments (as well as those from other reviewers) either by performing additional experiments or by modifying the text.

The revised manuscript is at the same time stronger and clearer than the previous one. It now properly cites previous studies.

As detailed below, two aspects could however still be improved (or should at least be properly discussed in the main text) in view of the recent literature regarding (1) the moonlighting localization of Nups previously defined as inner ring and central channel components (Nup93, Nup205 and Nup62) and related to this first point, (2) the stoichiometry of some Nups defined by their FCS-calibrated confocal microscopy as $\frac{NUP}{SEP}$ compared to recent studies.

1) Lines 80 and 190 (and likely elsewhere in the text) : In view of the recent cryoEM/ET studies (previously on Biorxiv since end of 2021, now published in Science), that now place 24 copies of Nup93 and NUP205 on the cytoplasmic (16 copies) and nuclear (8 copies) rings of the vertebrate NPCs, in addition to the 32 (Nup93) and 16 (NUP205) copies present in the central ring, it no longer seems appropriate to define these two nucleoporins solely as « central ring complex members ».

Likewise, as already indicted in my previous review, Nup62 is also present on the cytosolic side along with NUP214, and its definition as “channel Nups” is therefore ambiguous. Although the authors indicate in their answer to this previous comment that they used a model from 2016, and now indicate this possibility in the method section, a general audience will not read the methods. The models presented in the Figure 5 and Extended Data Fig. 5 b and c are therefore out of date and not appropriate. A proper (tiny but updated) scheme of the NPC may deserve to appear in the main figure.

2) How do the authors explain the important (more than two-fold) discrepancy between the copy number per pore they measured for Nup205 and Nup358, (Fig 1D) as compared to these recent studies (16 copies in their study versus 40 copies by cryoET for Nup205, and Nup358 ; 48 versus 56 for Nup93 – the latter falls likely in the error range...).

Although these Nups were homozygously mEGFP-tagged, subsequent protein degradation or altered mRNA splicing, export or translation cannot be excluded. Therefore, as indicated (lines 99-102), « for Nup153 and Pom121 that exhibited lower $\frac{NUP}{SEP}$ concentrations likely due to subhomozygous tagging » (Fig. 2a, Extended Data Fig. 1), should their stoichiometry not also be « normalized to the expected number of copies in the mature pore ? ». Would that modify some conclusions?

Not also that as indicated by the reviewer 1, Pom121 does not seem to be « subhomozygously » tagged, but as now indicated in the revised figure legend, the authors « regarded the clone #32 of Pom121-mCherry as subhomozygous (please correct spelling) because the protein abundance was much less than expected although the blot indicated homozygous tagging ». (this should be clarified in the main text and not hidden in an extended Fig legend).

Western blots with anti-Nups in untagged versus tagged cell lines should be provided (I assume the authors might have been already performed this important control), to also exclude partial proteolytic cleavage of the GFP tag (as shown now for Nup188 in extended data) and compare endogenous/tagged protein levels.

Would proteolytic removal of the tag possibly modify some interpretations/ conclusions/models ?

(Note that by chance, Nup188 is solely localized on the inner ring, and the copy number of this Nup in the tagged cell line is as expected). While this likely validate the model, the other points should nevertheless be clarified.

Minor point:

In view of the previous literature, and despite the new data and concepts provided by this excellent study, I still consider the title to be misleading.

Something like "strengthens the divergences between two mechanisms" would be more appropriate than "reveals two distinct mechanisms".

Author Rebuttals to First Revision:

Point-by-point response (referee comments in *blue italics*, our response in black)

Referee #1 (Remarks to the Author):

As stated in my previous report, NPC assembly has great biological relevance and is -- in light of the complex structure -- of broad interest. The detailed measurements, the integrative modeling, and the insights gained in the present work do make a significant contribution. In the revision, the authors have made a serious effort to address all of my concerns. Points 4, 5 and 6 require further (minor) clarifications.

4) Variations in assembly pathway. I found it intriguing that the dominant pathway (85%) with early nuclear-ring assembly appears to compete with a cytoplasmic-ring-first pathway (15%; see new Ext. Data Fig. 6). Is this an expected uncertainty or an artifact of the modeling, or do the two assembly paths indeed run in parallel? Are there experiments with which one could exclude flux through the alternative pathway? I wonder if this should at least be commented on?

This is indeed an uncertainty of our structural models. Most of the variations are for the assembly intermediates at early stages (5 and 6 min after anaphase onset). We think it is due to the lower protein density in our electron tomography images for the earlier assembly intermediates. We have clarified this point in the legend of Extended Data Figure 7. In order to experimentally examine if all the nuclear pores assemble with the same nuclear-ring-first pathway, the assembly of the inner and outer-ring components would need to be visualised in a large number of single NPCs at a spatial resolution of at least 10 nm in xyz 5–6 min after anaphase onset. While this is technically very challenging and beyond the scope of this study, in the future it should be possible by using recently developed super-resolution microscopy approaches such as MINFLUX (Gwosch et al., Nat Methods, 2020. DOI: 10.1038/s41592-019-0688-0).

5) Imposition of symmetry in modeling. I agree that at this stage it is reasonable to restrict the modeling to symmetric intermediates. However, I am confused by the assertion on p. 35 that “the assumption that the subcomplexes preform with 8-fold multiplicity before assembling into the NPC is supported by our diffusion coefficient measurement of the GFP-labeled Nups by FCS.” As I interpret the FCS data, the Nups are all part of larger complexes. However, I do not see any concrete evidence for having subcomplexes present in stoichiometric numbers (“8-fold multiplicity”) at each site of NPC emergence. Please delete or clarify.

We agree with the reviewer that this statement was confusing and have removed it.

6) Faded-out tomogram density. I may not have been clear in the way I formulated my earlier point on the tomogram density of intermediates fading out towards the periphery. One possible (and to me likely) cause for the partial density is that the ET reconstructions are averages over partially incomplete assembly intermediates, each one of which may not obey full C8 symmetry in the faded-out parts. A second possible factor is that some of these parts are dynamic. To be clear, I am fine with

modeling under the assumption of 8-fold symmetry at this stage, I would have preferred a more cautious wording.

As we have described in our previous point-by-point response letter, we think that the majority of intermediates have approximate 8-fold symmetry for the outer- and inner rings, because our previous electron microscopy analysis demonstrated 8-fold symmetry of the outer-ring complex in the intermediates at 5, 6, 8, 10 and 15 min after anaphase onset and 8-fold symmetry of the inner-ring complex at 10 and 15 min after anaphase onset (Supplementary figure 7, Otsuka et al., Nat Struct Mol Biol, 2018, DOI: 10.1038/s41594-017-0001-9). However, as the reviewer pointed out, we also think that there are likely to be variations in the copy numbers at the single pore level. We have clarified this point further in the methods (section “Structural modeling of NPC assembly pathway”).

Referee #2 (Remarks to the Author):

I appreciate the authors' response and the detailed rebuttal they have provided. I believe the revised manuscript is much improved. In particular I find the discussion of how two separate NPC assembly pathways are deconvolved using their in vivo approach is novel and now appropriately described in the new version. Derivation of biochemical parametrics using live cell imaging is powerful and underutilized in Cell Biology, thus it is critically important that work of this nature appear in top line journals.

I fully support publication at this time.

We thank the reviewer for the appreciation of our work.

Referee #3 (Remarks to the Author):

The authors have satisfactorily addressed the reviewers' comments. I still think this is an important work which will help the scientific community understanding nuclear pore complex assembly pathways.

I have one question which the authors might want to comment on in the manuscript: I am astonished by the low Nup205 copy numbers (Fig 1C). This is contradicting previous data from the Beck lab (<https://doi.org/10.1038/msb.2013.4>) which suggested that Nup205 is equally abundant as members of the Y-complex. It is also at odd with the recent NPC structures e.g. by the Hoelz lab (doi: <https://doi.org/10.1101/2021.10.26.465796>). In addition, Nup188 is expected to be present in half the copy numbers as compared to Nup205 (again <https://doi.org/10.1038/msb.2013.4> and <https://doi.org/10.1101/2021.10.26.465796>).

The reviewer is correct that there are differences between the copy number estimates for Nup205 between our study based on quantitative live cell imaging (estimated at 16 copies) and recent structural studies that fitted Nup structures into cryo-EM densities (estimated at 40 copies) (e.g. Mosalaganti et al., *Science*, 2022, DOI: 10.1126/science.abm9506; Petrovic et al., *Science*, 2022, DOI: 10.1126/science.abm9798). There are multiple possible reasons for the different estimates. The cell line we have used may express lower than physiological levels of tagged Nup205. However, we carefully validated homozygous knock-in of GFP into the endogenous Nup205 gene by Southern

blotting as well as integrity of the tagged protein by Western blotting. We did not see any signs of untagged alleles nor proteolytic degradation of the tagged protein. On the other hand, the higher estimate of Nup205 by the recent structural work is based on the fitting of a modelled Nup205 structure into the cryo-EM density obtained after averaging many NPCs. To our knowledge, the presence of additional 24 copies of Nup205 in the outer rings was so far not validated by another experimental approach. In addition, it is worth noting that the averaged cryo-EM densities and structural models aim for a “maximally occupied” NPC model and do not capture pore-to-pore heterogeneity in Nup stoichiometry, for which there is increasing evidence by multiple approaches. By contrast, our data averages over all pores in the living cell and will therefore reports lower average stoichiometry for Nups where variation from the maximum number per pore exists. Further studies and likely new methods are clearly needed in the future to determine the physiological copy number with single NPC precision. As suggested by the reviewer, we have provided a comment regarding this issue in the discussion of the manuscript and also stated that the estimate we make is different in the second paragraph of the section “Quantitative live imaging of ten nucleoporins”.

Referee #4 (Remarks to the Author):

In the revised version of their manuscript, the authors have taken in account most of my comments (as well as those from other reviewers) either by performing additional experiments or by modifying the text.

The revised manuscript is at the same time stronger and clearer than the previous one. It now properly cites previous studies.

As detailed below, two aspects could however still be improved (or should at least be properly discussed in the main text) in view of the recent literature regarding (1) the moonlighting localization of Nups previously defined as inner ring and central channel components (Nup93, Nup205 and Nup62) and related to this first point, (2) the stoichiometry of some Nups defined by their FCS-calibrated confocal microscopy as compared to recent studies.

1) Lines 80 and 190 (and likely elsewhere in the text) : In view of the recent cryoEM/ET studies (previously on Biorxiv since end of 2021, now published in Science), that now place 24 copies of Nup93 and NUP205 on the cytoplasmic (16 copies) and nuclear (8 copies) rings of the vertebrate NPCs, in addition to the 32 (Nup93) and 16 (NUP205) copies present in the central ring, it no longer seems appropriate to define these two nucleoporins solely as « central ring complex members ». Likewise, as already indicated in my previous review, Nup62 is also present on the cytosolic side along with NUP214, and its definition as “channel Nups” is therefore ambiguous. Although the authors indicate in their answer to this previous comment that they used a model from 2016, and now indicate this possibility in the method section, a general audience will not read the methods. The models presented in the Figure 5 and Extended Data Fig. 5 b and c are therefore out of date and not appropriate. A proper (tiny but updated) scheme of the NPC may deserve to appear in the main figure.

The reviewer is correct that there are differences between the copy number estimates for Nup93 and Nup205 between our study based on quantitative live cell imaging (estimated at 48 and 16 copies) and recent structural studies that fitted Nup structures into cryo-EM densities (estimated at 56 and 40

copies) (e.g. Mosalaganti et al., *Science*, 2022, DOI: 10.1126/science.abm9506; Petrovic et al., *Science*, 2022, DOI: 10.1126/science.abm9798). There are multiple possible reasons for the different estimates. The cell line we have used may express lower than physiological levels of tagged Nup93 and Nup205. However, we carefully validated homozygous knock-in of GFP into the endogenous Nup93/205 genes by Southern blotting as well as integrity of the tagged protein by Western blotting. We did not see any signs of untagged alleles nor proteolytic degradation of the tagged protein. On the other hand, the higher estimate of Nup93/205 by the recent structural work is based on the fitting of a modelled Nup93/205 structures into the cryo-EM density obtained after averaging many NPCs. To our knowledge, the presence of additional 24 copies of Nup93/205 in the outer rings was so far not validated by another experimental approach. In addition, it is worth noting that the averaged cryo-EM densities and structural models aim for a “maximally occupied” NPC model and do not capture pore-to-pore heterogeneity in Nup stoichiometry, for which there is increasing evidence by multiple approaches. By contrast, our data averages over all pores in the living cell and will therefore reports lower average stoichiometry for Nups where variation from the maximum number per pore exists. Further studies and likely new methods are clearly needed in the future to determine the physiological copy number with single NPC precision.

We therefore feel that the uncertainty regarding maximal copy number of some Nups, such as Nup205, does not invalidate our integrative modelling approach, which fits the experimentally determined copy number changes over time of many Nups into the orthogonally experimentally determined low resolution protein density, during assembly. The main predictions of the integrative model, such as the sequential assembly of nuclear and cytoplasmic rings and the radial dilation of nuclear membrane via recruitment of inner-ring and central channel complexes would remain unchanged, even if the final fully assembled NPC would have additional copies of some Nups. In our view the presented model thus provides interesting predictions and represents our integrative approach well.

We agree with the reviewer that it is indeed important that future studies clarify the physiological copy number of Nups in the NPC and its variability with single pore resolution to arrive at a new consensus model. We have now explicitly stated the difference in the estimates in the main text (second paragraph of the section “Quantitative live imaging of ten nucleoporins”) and added a comment for the need to better study this in the future in the discussion. Regarding the fraction of Nup62 that binds Nup214, it is unfortunately not possible to add a detailed explanation to the main text due to space constraints, and we have therefore included an explicit note for the reader to refer to the Methods in the main text (the second paragraph of the section “Integrative modeling of the postmitotic NPC assembly”).

2) How do the authors explain the important (more than two-fold) discrepancy between the copy number per pore they measured for Nup205 and Nup358, (Fig 1D) as compared to these recent studies (16 copies in their study versus 40 copies by cryoET for Nup205, and Nup358 ; 48 versus 56 for Nup93 – the latter falls likely in the error range...).

Although these Nups were homozygously mEGFP-tagged, subsequent protein degradation or altered mRNA splicing, export or translation cannot be excluded. Therefore, as indicated (lines 99-102), « for Nup153 and Pom121 that exhibited lower concentrations likely due to subhomozygous tagging » (Fig. 2a, Extended Data Fig. 1), should their stoichiometry not also be « normalized to the expected number of copies in the mature pore ? ». Would that modify some conclusions?

As discussed above, there are multiple possible reasons for the differences between the copy number estimates for Nup205 and Nup358 between our study based on quantitative live cell imaging (estimated at 16 copies each) and recent structural studies that fitted Nup structures into cryo-EM densities (estimated at 40 copies each). As we described above, the uncertainty regarding maximal copy number of some Nups does not invalidate our integrative modelling approach. To make this issue clear, we have provided a comment in the discussion of the manuscript and also stated that the estimate we make is different in the second paragraph of the section “Quantitative live imaging of ten nucleoporins”.

Not also that as indicated by the reviewer 1, Pom121 does not seem to be « subhomozygously » tagged, but as now indicated in the revised figure legend, the authors « regarded the clone #32 of Pom121-mCherry as subhomozygRous (please correct spelling) because the protein abundance was much less than expected although the blot indicated homozygous tagging ». (this should be clarified in the main text and not hidden in a extended Fig legend).

As the reviewer requested, we have clarified this point in the main text (second paragraph of the section “Quantitative live imaging of ten nucleoporins”) and corrected the spelling.

Western blots with anti-Nups in untagged versus tagged cell lines should be provided (I assume the authors might have been already performed this important control), to also exclude partial proteolytic cleavage of the GFP tag (as shown now or Nup188 in extended data) and compare endogenous/tagged protein levels.

Would proteolytic removal of the tag possibly modify some interpretations/ conclusions/models ? (Note that by chance, Nup188 is solely localized on the inner ring, and the copy number of this Nup in the tagged cell line is as expected). While this likely validate the model, the other points should nevertheless be clarified.

We have now provided Western blot analysis for GFP-Nup205 and GFP-Nup358 cell lines (Supplementary Figs. 2, 3), as well as the uncropped Western blots for Nup188-GFP cells (Supplementary Fig. 1). The immunoblots show no proteolytic degradation of GFP-tagged Nup205, Nup358 and Nup188. We have clarified this issue in the main text (second paragraph of the section “Quantitative live imaging of ten nucleoporins”) and in the discussion.

Minor point:

In view of the previous literature, and despite the new data and concepts provided by this excellent study, I still consider the title to be misleading.

Something like “strengthens the divergences between two mechanisms” would be more appropriate than “reveals two distinct mechanisms”.

We thank the reviewer for this consideration. Also in the light of the strict length constraints of the title (75 character including space), we remain of the opinion that the title of our manuscript is reasonable, because the model and our data have revealed for the first time systematically how the molecular mechanisms of the two assembly pathways differ, with the major difference being that the central ring and nuclear filaments follow an inverted molecular order between the postmitotic and interphase NPC assembly.

Reviewer Reports on the Second Revision:

Referees' comments:

Referee #1 (Remarks to the Author):

My concerns have been adequately addressed. I support publication of this important and interesting study in Nature.

As one last very minor issue, I wonder if in the caption of Extended Data Figure 7 it should be "lower" not "higher signal-to-noise ratio".

Referee #3 (Remarks to the Author):

the authors have sufficiently addressed all points raised.

Referee #4 (Remarks to the Author):

As detailed below, the authors did not really take my last comments into account and partially explained it by stating that "it is unfortunately not possible to add a detailed explanation to the main text due to space constraints,".

I consider that the authors should be allowed by the editor to use a few extra lines in the main text to properly clarify the fact that: 1) their model used stoichiometry data for Nup205 and Nup358 (16 copies) different from those published by others (Ori et al., 2013, and the recent structure papers in Science 2022) most likely because the cell lines they used express lower levels of tagged Nup358 and NUP205 and that 2) likewise the now established dual location of Nup62 and Nup205 was not taken into account in their study.

1) Indeed, in their rebuttal, the authors mainly try to partially dismiss the fact that their cell lines might have some issue by stating that :

" we carefully validated homozygous knock-in of GFP into the endogenous Nup93/205 genes by Southern blotting as well as integrity of the tagged protein by Western blotting. We did not see any signs of untagged alleles nor proteolytic degradation of the tagged protein".

In fact, they also have performed the same controls for the Pom121 cell line. While, this strain went through all controls, it nevertheless had lower than expected protein levels, as admitted by the authors that «regarded the clone #32 of Pom121-mCherry as subhomozygous because the protein abundance was much less than expected although the blot indicated homozygous tagging ». (text buried in the extended figure legend 1, that would deserve to appear in a main text...).

"To our knowledge, the presence of additional 24 copies of Nup93/205 in the outer rings was so far not validated by another experimental approach".

The presence of additional copies of Nup93 (at least 16 additional ones, if not 24) is validated by the authors themselves, as they found about 48 copies of Nup93, while only 32 would be anticipated otherwise. Where are these extra-copies of Nup93 localized in their model?? A higher stoichiometry was also previously validated by quantitative mass-spectrometry measurements of HeLa cell nuclear envelopes, that concluded that Nup205 and Nup358, had comparable stoichiometry as Nup107(that is about 32 copies of each NUP) [Ori et al., Mol Syst Biol. 2013; 9: 648].

2) Regarding the dual location of NUP62, this is now only mentioned in line 723 (in the method section that non-specialized reader may not go read) by the sentence " "It might be the case that" the remaining 16 copies of Nup62 are bound to Nup214 (Line 723). This should be placed in the main text or main figure legend as "the remaining 16 copies of Nup62 are likely bound to Nup214 ",

with appropriate references, and likewise, it should be indicated for Nup205 that additional copies are also present on the cytoplasmic and nuclear rim of the NPC. The authors should clearly indicate that they were so far not able to take this dual location in account in their models, and that this will deserve further analyses.

3) Western blots with anti-Nups in untagged versus tagged cell lines should be provided to .. compare endogenous/tagged protein levels.

The western blot for NUP358 now provided as supplemental figure was previously published, in an inverted manner (Fig 7 along with its loading control [Koch et al, Nat protocol, 2018 Jun;13(6):1465-1487. doi: 10.1038/nprot.2018.042 - attached figure]. Of note, this blot clearly shows that the protein levels vary in-between cell lines, and the choice of the clone (not the one with the strongest expression, at least another one had no more degradation but apparently higher level of tagged protein) is therefore not clear to me.

Moreover, as this blot was not probed with an anti-NUP358 antibody, one cannot compare to the level of the endogenous proteins.

Figure 7

Koch et al., Generation and validation of homozygous fluorescent knock-in cells using CRISPR-Cas9 genome editing, Nat protocol, 2018

Regarding Nup205, there is no loading control provided, and a range of the control sample would be required to ensure that there is not a two-fold decrease of the total GFP-Nup205 protein level compared to endogenous NUP205 (indeed, the signal on this line seems to be fainter).

I did not find the legend for these figures. The authors should also ensure that the Nup205 blot was not previously published.

Author Rebuttals to Second Revision:

Point-by-point response (referee comments in *blue italics*, our response in black)

Referee #1 (Remarks to the Author):

My concerns have been adequately addressed. I support publication of this important and interesting study in Nature.

As one last very minor issue, I wonder if in the caption of Extended Data Figure 7 it should be “lower” not “higher signal-to-noise ratio”.

We thank the reviewer for the positive feedback on our work and for pointing out the typo in the caption of Extended Data Figure 7. Indeed, it should be “lower signal-to-noise ratio” not “higher signal-to-noise ratio”. We have corrected it.

Referee #3 (Remarks to the Author):

the authors have sufficiently addressed all points raised.

Thank you very much.

Referee #4 (Remarks to the Author):

As detailed below, the authors did not really took my last comments in account and partially explained it by stating that "it is unfortunately not possible to add a detailed explanation to the main text due to space constraints,".

I consider that the authors should be allowed by the editor to use a few extra lines in the main text to properly clarify the fact that: 1) their model used stoichiometry data for Nup205 and Nup358 (16 copies) different from those published by others (Ori et al., 2013, and the recent structure papers in Science 2022) most likely because the cell lines they used express lower levels of tagged Nup358 and NUP205 and that 2) likewise the now established dual location of Nup62 and Nup205 was not taken in account in their study.

We appreciate that the editor has waived space considerations to discuss this issue in the main text. As detailed below, paragraphs four, six and seven of the discussion now provide a more detailed consideration. In summary, the following points are now addressed: (i) the constraint of the fully assembled state of the NPC imposed on our assembly process model, (ii) the differences between copy number estimates for some of the Nups in our study derived by quantitative live cell imaging and recent structural studies derived by fitting individual Nup proteins into average cryo-EM densities, (iii) the possible reasons for these differences due to methodological limitations, including pore-to-pore variability under physiological conditions and lower expression levels of knock-in proteins, and (iv) providing new Western Blotting data that indicate that lower expression levels may explain the differences for Nup358 but are unlikely to explain the differences for Nup205.

1) Indeed, in their rebuttal, the authors mainly try to partially dismiss the fact that their cell lines might have some issue by stating that :

" we carefully validated homozygous knock-in of GFP into the endogenous Nup93/205 genes by Southern blotting as well as integrity of the tagged protein by Western blotting. We did not see any signs of untagged alleles nor proteolytic degradation of the tagged protein".

In fact, they also have performed the same controls for the Pom121 cell line. While, this strain went through all controls, it nevertheless had lower than expected protein levels, as admitted by the authors that «regarded the clone #32 of Pom121-mCherry as subhomozygous because the protein abundance was much less than expected although the blot indicated homozygous tagging ». (text buried in the expended figure legend 1, that would deserve to appear in a main text...).

We apologize if the brevity of our explanation caused misunderstanding. We did not intend to dismiss the comments of the reviewer. Following the reviewer's suggestion, we now also discuss the Pom121 cell line in the Discussion section of the main text.

"To our knowledge, the presence of additional 24 copies of Nup93/205 in the outer rings was so far not validated by another experimental approach".

The presence of additional copies of Nup93 (at least 16 additional ones, if not 24) is validated by the authors themselves, as they found about 48 copies of Nup93, while only 32 would be anticipated otherwise. Where are these extra-copies of Nup93 localized in their model?? A higher stoichiometry was also previously validated by quantitative mass-spectrometry measurements of HeLa cell nuclear envelopes, that concluded that Nup205 and Nup358, had comparable stoichiometry as Nup107(that is about 32 copies of each NUP) [Ori et al., Mol Syst Biol. 2013; 9: 648].

As outlined above, we now discuss the differences in copy number estimates between our study and other studies in the Discussion section of the main text.

2) Regarding the dual location of NUP62, this is now only mentioned line 723 (in the method section that non-specialized reader may not go read) by the sentence " It might be the case that " the remaining 16 copies of Nup62 are bound to Nup214 (Line 723). This should be place in the main text or main fig legend as "the remaining 16 copies of Nup62 are likely bound to Nup214 ", with appropriate references, and likewise, it should be indicated for Nup205 that additional copies are also present on the cytoplasmic and nuclear rim of the NPC.

The authors should clearly indicate that they were so far not able to take this dual location in account in their models, and that this will deserve further analyses.

Following the suggestion, we have moved the statement of the dual location of Nup62 from the method section to the main text (the Discussion section and the legend of Figure 5). We have also added a reference paper that showed the Nup214-Nup62 interaction in human cells using proximity-dependent biotin identification (Kim D. I. et al. Proc. Natl. Acad. Sci. U. S. A., 2016, doi: 10.1073/pnas.1406459111). In addition, we now explicitly state that our computational model does not take the assembly of potential additional copies of Nup62 and 205 into account in the Discussion section of the main text.

3) Western blots with anti-Nups in untagged versus tagged cell lines should be provided to .. compare endogenous/tagged protein levels.

The western blot for NUP358 now provided as supplemental figure was previously published, in an inverted manner (Fig 7 along with its loading control [Koch et al, Nat protocol, 2018 Jun;13(6):1465-1487. doi: 10.1038/nprot.2018.042 - attached figure]. Of note, thos blot clearly

shows that the protein levels vary in-between cell lines, and the choice of the clone (not the one with the strongest expression, at least another one had no more degradation but apparently higher level of tagged protein) is therefore not clear to me.

Moreover, as this blot was not probed with an anti-NUP358 antibody, one cannot compare to the level of the endogenous proteins.

First of all, we sincerely apologize for the confusion we caused by showing the Western blot for Nup358 from our previous publication (Koch et al, 2018, Nat Protocols, doi: 10.1038/nprot.2018.042), where this cell line was originally reported, in an inverted manner. We should have double checked the orientation of the blot and explicitly pointed out again in the extended data figure legend that this data was previously published.

As the reviewer correctly pointed out, this blot was done using anti-GFP antibody and while it shows that full length protein is tagged, it does not allow to compare the expression level of GFP-tagged Nup358 relative to the endogenous, untagged Nup358 in the parental, not genome edited, cell line. To address the reviewer's comment, we have now tested anti-Nup358 antibodies and provide new Western blotting data (Supplementary Fig. 2). This data indicates that the Nup358 cell line expresses lower levels than the endogenous protein in the parental cell line, despite homozygous integration of the tag as verified by Southern Blot previously. We discuss this data as a potential explanation of the lower copy number estimate compared to other studies in the Discussion section of the main text.

Regarding Nup205, there is no loading control provided, and a range of the control sample would be required to ensure that there is not a two-fold decrease of the total GFP-Nup205 protein level compared to endogenous NUP205 (indeed, the signal on this line seems to be fainter).

To address the reviewer's comment, we now provide new Western blots with a loading control for the Nup205 cell line (Supplementary Fig. 2). The Western blots show that our cell line expresses GFP-tagged Nup205 at comparable levels to the endogenous protein in the parental cell line. As mentioned above, we now discuss this data and other possible reasons for differences in copy number estimate between our approach and structural approaches, as well as the need for robust quantitative methods with single protein complex resolution and sensitivity under physiological conditions in the future in the Discussion section of the main text.

We note that we furthermore carried out similar new Western Blot analysis for the Nup188 cell line generated for the revision of this study (Extended Fig. 9b). As for Nup205, the Western blots using an anti-Nup188 antibody show that our cell line expresses GFP-tagged Nup188 at comparable levels to the endogenous protein in the parental cell line.

I did not find the legend for these figures. The authors should also ensure that the Nup205 blot was not previously published.

The legends for supplementary figures were provided but in a different file (SI Guide). To avoid missing this information, we have now combined the supplementary figures with their legends in a single PDF file. The Nup188, Nup205 and Nup358 blots shown in the new supplementary figures have not been published previously.

Reviewer Reports on the Third Revision:

Referees' comments:

Referee #4 (Remarks to the Author):

In this revised version, the authors have performed western blots to better characterize the NUP358 and NUP205 cell lines they used, which revealed substoechiometric levels of Nup358 but not Nup205. Some aspects are also clarified in the discussion.

At that stage, no more experiments are required, but I consider that several points should still be modified in the text or figures. This should take no more than a few hours to the authors.

1) In figures 1 and 2, a star should be added on Nup358 as on Nup153 and Pom121 in figure 2 as the authors have now established that NUP358 is also substoechiometric (or Nup358 should be removed from figure 1 as done for Pom121 and Nup153).

2) A clearer explanation should be provided in the figure legend : Currently Line 406 : ^{SEP}* : "For Nup153 and Pom121, these Nups are not « fully validated to be homozygously-tagged »" is misleading as it suggests that status of these lines is ambiguous, while it is not at least for Nup153. ^{SEP}It should be replaced by « * : Nup153 is heterozygously tagged and Pom121 and Nup358 are substoechiometric (see Extended figures x and x). Note that the western blots are as important as the genomic studies and should also appear as extended data and not supplemental data.

3) Line 86 "ensuring that the tagged subunit was expressed at physiological levels,". This is misleading as the authors (but not the reader at that stage) know this is not the case for 3 cell lines. This should be completed by "which was the case for most (8 out of 11) cell lines" (extended figures x and y).

4) Line 103: "We then used our [validated] cell line resource to quantitatively image the Nups during both ^{SEP}postmitotic and interphase NPC assembly.. (remove the term « validate » since 3 lines were not validated).

5) I had asked for this in one of my previous review: a scheme of the NPC, positioning the various Nups used in this study seems mandatory in the main figure 1 for non NPC experts (which will be the case for most Nature readers)(If space is needed, the quantification of Fig 1c could be moved to extended data). ONCE CORRECTED (see below), the model used in Extended Figure 6 could be used for this purpose (with the indications about recent studies mentioned currently at the end of figure 5).

6) This model in Extended Figure 6 has to be corrected as it does not take in account the additional position of NUP62, NUP93 (that were already established) and NUP205 that is currently indicated in the discussion but in a fuzzy manner. These 3 NUPS should be positioned on the cytoplasmic filament (for NUP62) and on the outer rings (NU93 and 205), possibly in black with a star, with a sentence in the legend indicating that these positions were not taken in account in this study.

7) In their rebuttal, the authors mention that they now cite "a reference paper that showed the Nup214-Nup62 interaction in human cells using proximity-dependent biotin identification (Kim D. I. et al. 2016)". However, interaction between Nup214 and Nup62 is not demonstrated in Kim et al., but seems to be proposed based on indirect evidences and data from 4 cited references. "As discussed in Kim et al" would be more appropriate. At that time, it was hypothesized based on studies in yeast Reviewed in Lin DH and Hoelz A,2019). To my knowledge, this fact is now fully demonstrated in mammalian cells in several studies (see amongst other the fig 2 from Bley et al., Science 2022).

MINOR POINTS THAT COULD ALSO EASILY BE CORRECTED :

- Line 43 : specify that you talk about cells with open mitosis (adding in mammalian cells would be

sufficient) as this is not obvious for all Nature readers.

- line 69 « postmitotic and interphase ^[1]_{ISEP} assemblies, whose co-occurrence in different regions of the nucleus was only discovered later »; ^[1]_{ISEP} the reference is missing (Maeshima, K. et al. 2006, ref 27).

- line 178 Integrative modelling of the postmitotic assembly ^[1]_{ISEP}(add postmitotic)

Author Rebuttals to Third Revision:

Point-by-point response (referee comments in *blue italics*, our response in black)

Referee #4 (Remarks to the Author):

In this revised version, the authors have performed western blots to better characterize the NUP358 and NUP205 cell lines they used, which revealed substoechiometric levels of Nup358 but not Nup205. Some aspects are also clarified in the discussion.

At that stage, no more experiments are required, but I consider that several points should still be modified in the text or figures. This should take no more than a few hours to the authors.

1) In figures 1 and 2, a star should be added on Nup358 as on Nup153 and Pom121 in figure 2 as the authors have now established that NUP358 is also substoechiometric (or Nup358 should be removed from figure 1 as done for Pom121 and Nup153.

Following the reviewer's suggestions here and in comment 2) below, we have now highlighted Nup358 with asterisk in Figure 2, and have clarified that the GFP-tagged Nup358 is subphysiologically expressed in the legend of Figure 2.

*2) A clearer explanation should be provided in the figure legend : Currently Line 406 : * : "For Nup153 and Pom121, these Nups are not « fully validated to be homozygously-tagged »" is misleading as it suggests that status of these lines is ambiguous, while it is not at least for Nup153. It should be replaced by « * : Nup153 is heterozygously tagged and Pom121 and Nup358 are substoechiometric (see Extended figures x and y). Note that the western blots are as important as the genomic studies and should also appear as extended data and not supplemental data.*

As the reviewer recommended, we have clarified that Nup153 is heterozygously tagged and Nup358 and Pom121 are subphysiologically expressed in the legend of Figure 2. We have also moved the Western blot data from Supplementary Figure 2 to new Extended Data Figure 2.

3) Line 86 "ensuring that the tagged subunit was expressed at physiological levels,". This is misleading as the authors (but not the reader at that stage) know this is not the case for 3 cell lines. This should be completed by "which was the case for most (8 out of 11) cell lines" (extended figures x and y).

As the reviewer recommended, we have further clarified this point (Line 87-88 on page 4).

4) Line 103: "We then used our [validated] cell line resource to quantitatively image the Nups during both postmitotic and interphase NPC assembly.. (remove the term « validate » since 3 lines were not validated).

We have removed the term as requested (Line 104 on page 5).

5) I had asked for this in one of my previous review: a scheme of the NPC, positioning the various Nups used in this study seems mandatory in the main figure 1 for non NPC experts (which will be the case for most Nature readers)(If space is needed, the quantification of Fig 1c could be moved to extended data). ONCE CORRECTED (see below), the model used in Extended Figure 6 could be used for this purpose (with the indications about recent studies mentioned currently at the end of figure 5).

In our opinion, it is important to show the quantification of Nup copy numbers in the main Figure 1. We do not think that it would help to move the cartoon scheme of the NPC from now Extended Figure 7 to main Figure 1, as this scheme is used to illustrate the overall difference in assembly kinetics

between postmitotic and interphase assembly. The scheme is not intended as a detailed representation of NPC structure. We therefore maintained showing the NPC kinetics scheme in the Extended Figure (please see also our response to comment 6) below).

6) This model in Extended Figure 6 has to be corrected as it does not take in account the additional position of NUP62, NUP93 (that were already established) and NUP205 that is currently indicated in the discussion but in a fuzzy manner. These 3 NUPS should be positioned on the cytoplasmic filament (for NUP62) and on the outer rings (NUP93 and 205), possibly in black with a star, with a sentence in the legend indicating that these positions were not taken in account in this study.

As explained above, the main point of the NPC scheme in now Extended Figure 7 is to illustrate that the Nups have different assembly kinetics between postmitotic and interphase NPC assembly pathways, rather than representing its structure in detail. We have now stated that it is indeed “a simplified scheme of the NPC” in the legend of Extended Figure 7 to avoid any confusion in this regard. To illustrate the difference in overall kinetics, we do not think that it is helpful to attempt to illustrate sub-populations of Nup62, Nup93 and Nup205 in this simplified scheme. In fact, we think that this would rather be confusing to the reader, as our computational model does not take the assembly of the potential additional copies of Nup62 and 205 into account, which is stated clearly in the Discussion of the main text.

7) In their rebuttal, the authors mention that they now cite “a reference paper that showed the Nup214-Nup62 interaction in human cells using proximity-dependent biotin identification (Kim D. I. et al. 2016)”. However, interaction between Nup214 and Nup62 is not demonstrated in Kim et al., but seems to be proposed based on indirect evidences and data from for 4 cited references. “As discussed in Kim et al” would be more appropriate. At that time, it was hypothesized based on studies in yeast Reviewed in Lin DH and Hoelz A,2019). To my knowledge, this fact is now fully demonstrated in mammalian cells in several studies (see amongst other the fig 2 from Bley et al., Science 2022).

To make it straightforward to support the Nup214-Nup62 interaction in human cells from the literature, we have now added an additional reference (von Appen, A. et al. In situ structural analysis of the human nuclear pore complex. *Nature* 526, 140–143, 2015), in which the authors affinity-purified the Nup62-Nup214 complex from human cell lysate (Line 264 on page 11). In our view, it is therefore not necessary to cite additional references that provide further evidence of this interaction in yeast, as the yeast and the human NPCs have fundamental differences in their stoichiometry and biogenesis.

MINOR POINTS THAT COULD ALSO EASILY BE CORRECTED :

- Line 43 : specify that you talk about cells with open mitosis (adding in mammalian cells would be sufficient) as this is not obvious for all Nature readers.

We have modified the text following the reviewer’s suggestion (Line 43 on page 2).

- line 69 « postmitotic and interphase assemblies, whose co-occurrence in different regions of the nucleus was only discovered later »; the reference is missing (Maeshima, K. et al. 2006, ref 27).

We respectfully disagree with the reviewer. Although the study (Maeshima K et al 2006) showed that there are pore-free islands on the nuclear envelope that originate from mitotic core-regions, the study did not show that two NPC assembly pathways co-occur in different regions of the nucleus.

- line 178 Integrative modelling of the postmitotic assembly (add postmitotic)

We have modified the text following the reviewer’s suggestion (Line 179 on page 8).